# How Far Are We From AGI: Are LLMs All We Need?

**Tao Feng**[1]*, **Chuanyang Jin**[2]*, **Jingyu Liu**[3]*, **Kunlun Zhu**[1]*
**Haoqin Tu**[4], **Zirui Cheng**[1], **Guanyu Lin**[5], **Jiaxuan You**[1]†

`{taofeng2, kunlunz2, jiaxuan}@illinois.edu`

[1]University of Illinois Urbana-Champaign [2]Johns Hopkins University [3]University of Chicago
[4]University of California, Santa Cruz [5]Carnegie Mellon University

**Reviewed on OpenReview:** `https://openreview.net/forum?id=H2ZKqfNd0U`

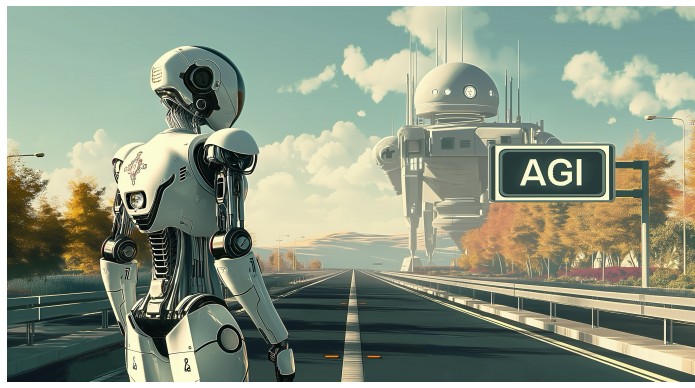

## Abstract

The evolution of artificial intelligence (AI) has profoundly impacted human society, driving significant advancements in multiple sectors. Yet, the escalating demands on AI have highlighted the limitations of AI's current offerings, catalyzing a movement towards Artificial General Intelligence (AGI). AGI, distinguished by its ability to execute diverse real-world tasks with efficiency and effectiveness comparable to human intelligence, reflects a paramount milestone in AI evolution. While existing studies have reviewed specific advancements in AI and proposed potential paths to AGI, such as large language models (LLMs), they fall short of providing a thorough exploration of AGI's definitions, objectives, and developmental trajectories. Unlike previous survey papers, this work goes beyond summarizing LLMs by addressing key questions about our progress toward AGI and outlining the strategies essential for its realization through comprehensive analysis, in-depth discussions, and novel insights. We start by articulating the requisite capability frameworks for AGI, integrating the internal, interface, and system dimensions. As the realization of AGI requires more advanced capabilities and adherence to stringent constraints, we further discuss necessary AGI alignment technologies to harmonize these factors. Notably, we emphasize the importance of approaching AGI responsibly by first defining the key levels of AGI progression, followed by the evaluation framework that situates the status-quo, and finally giving our roadmap of how to reach the pinnacle of AGI. Moreover, to give tangible insights into the ubiquitous impact of the integration of AI, we outline existing challenges and potential pathways toward AGI in multiple domains. In sum, serving as a pioneering exploration into the current state and future trajectory of AGI, this paper aims to foster a collective comprehension and catalyze broader public discussions among researchers and practitioners on AGI. [1]

---

*Equal contribution. In alphabetical order.

†Corresponding author. All student authors complete this work as interns at UIUC.

[1]Project website: `https://github.com/ulab-uiuc/AGI-survey`. Unlike traditional publications that remain static, we embrace an innovative approach by treating this paper as a living document. We warmly welcome feedback from the community and plan to update the paper annually. Contributors on the project website will be gratefully acknowledged in future revisions.

## Contents

# 1 Introduction

> *The path to AGI is not merely a technological journey; it's a philosophical quest to redefine what it means to be intelligent and ethical in a digital age.*
>
> — *Alex Kim, Director of AI Ethics at Future Insight Institute*

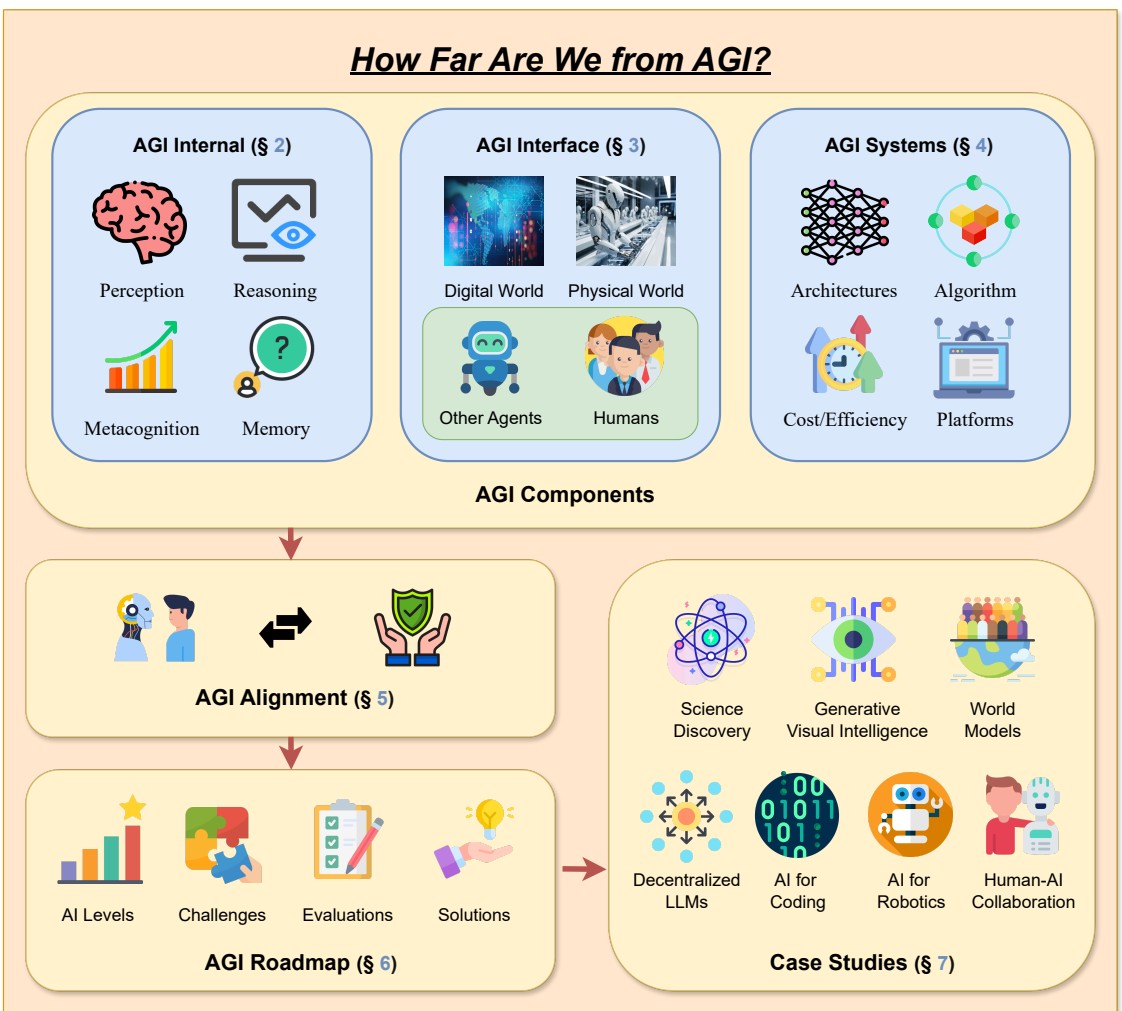

Figure 1: **Overall Structure of This Paper.** This paper starts with discussing core AGI components, including AGI Internal (§ 2), AGI Interface (§ 3), and AGI Systems (§ 4); these discussions help us measure the ability of AGI and estimate how far we are from AGI. As we get closer to AGI, we further expect AGI to meet various constraints, which can be realized by AGI Alignment (§ 5) techniques. We further outline an AGI Roadmap (§ 6) that helps researchers approach AGI responsibly. Finally, some Case Studies (§ 7) are presented to illustrate the current development of early-stage AGI in various fields.

To start approaching the question of how far we are from AGI, it is important to first ground ourselves with the history of artificial intelligence advancement and understand the urge for more advanced systems. Throughout the whole paper, we hope to provide evidences and insights on where we currently stand along the road towards AGI, from the lens of many modern AI systems such as large language models. The goal is to prudently keep questioning ourselves: are LLMs all we need? It is with this enduring curiosity and awareness that we might finally begin to touch the realm of AGI.

**Brief History of AI**   The development of artificial intelligence (AI) has revolutionized human society thanks to their powerful capabilities in many aspects, such as visual perception (Alayrac et al., 2022; Li et al., 2023j), language understanding (Wei et al., 2021; Schick et al., 2023), reasoning optimization (Wei et al., 2022b; Hao et al., 2023; Hu and Shu, 2023), etc. One salient example is the launch of AlphaFold (Jumper et al., 2021) by DeepMind in 2021, which revolutionized the field of protein structure prediction and advanced the frontiers of biological research. Despite the recent advancements, it is worth mentioning that the development of AI is not a smooth journey. Early AI research mainly focused on symbolic research (Stryker, 1959; Turner, 1975) and connectionism (Buckner and Garson, 1997; Medler, 1998), which laid the groundwork for computational approaches to intelligence. From the 1980s to the 1990s, AI faced its winter, and many researchers shifted to practical applications due to high expectations and subsequent disappointments in its development. The rise of machine learning and neural networks (Zadeh, 1996; Kosko, 1992) from the 1990s to the 2010s brought hope to researchers, which led to significant improvements in various applications like natural language processing, computer vision, and analytics. Starting from the 2010s, the advent of deep learning technologies revolutionized AI capabilities, with significant breakthroughs in image (Lu and Weng, 2007; Rawat and Wang, 2017) and speech recognition (Gaikwad et al., 2010; Povey et al., 2011). In recent years, with the emergence of ChatGPT (Wu et al., 2023a; Zhong et al., 2023b), the popularity of large language models (LLMs) has further transformed AI research due to its unified knowledge representation and superior multi-task solving capabilities.

**Craving for General-purpose AI**   Although AI has brought huge improvements to human society, the increasing material and spiritual demands of society have rendered people discontent with the mere convenience provided by AI. Consequently, achieving Artificial General Intelligence (AGI) that enables AI to perform a wider range of tasks more efficiently and effectively has emerged as a pressing concern, which used to describe an AI system that is at least as capable as a human at most tasks (Wang et al., 2018; Voss and Jovanovic, 2023). Therefore, our paper aims to raise attention to the pressing research questions: ***how far are we from AGI***, and moreover, ***how can we responsibly achieve AGI?***

To investigate these questions, existing research mainly falls into three categories: *Definition and Concept, Technical Methods and Applications*, and *Ethical and Social Implications.* (1) *Definition and Concept:* Wang et al. (2018) define the concept of AGI from a point of view of comparison with humans and propose different levels of it. Voss and Jovanovic (2023) provide direction for the path through the AGI by setting the human-like requirements associated with the AGI. (2) *Technical Methods and Applications:* Yan (2022); Wang et al. (2019a) propose that AGI can be achieved by combining logic with deep learning. Das et al. (2023) argues that many risks exist in the development of AGI technology, such as safety and privacy issues. (3) *Ethical and Social Implications:* Rayhan (2023) thinks that humans should consider the ethical implications of creating AGI, which contains impact on human society, privacy, and power dynamics. Bugaj and Goertzel (2007) propose five ethical imperatives and their implications for AGI interactions. These studies have characterized AGI from different aspects. Still, they lack a systematic assessment of the development process of AGI from various aspects and a clear definition of AGI goals, making it difficult to measure the gap between the current AI development and the future of AGI, and moreover, brainstorm possible paths to achieve AGI.

**Overall Structure of This Paper**   Specifically, as is shown in Figure 1, we start with an overview on the major capabilities required for future AGI in terms of its ***internal*** (§ 2) competence, its connection to the external world as an ***interface*** (§ 3), and the underlying infrastructure ***systems*** (§ 4) that support all these functionalities. When it comes to deployment, a more sophisticated ***alignment*** (§ 5) procedure is indispensable to unleash the growing potential of AGI systems under constraints and human expectations. Furthermore, we picture a ***roadmap***(§ 6) where we discuss how to responsibly approach AGI, which outlines the **three levels of AGI** which are Embryonic, Superhuman, and Ultimate AGI that helps locate our current state, associated evaluation framework, as well as our insights to some critical problems that might hinder our progress towards AGI. Finally, we list a couple of ***case studies*** (§ 7) which concretely describe the lineage of AI technology along various domains with cautious limitations and exciting future directions. We hope that this work can lay a common ground and provide a starting point for researchers and practitioners to reflect on the state of AI and brainstorm responsible approaches to achieve AGI.

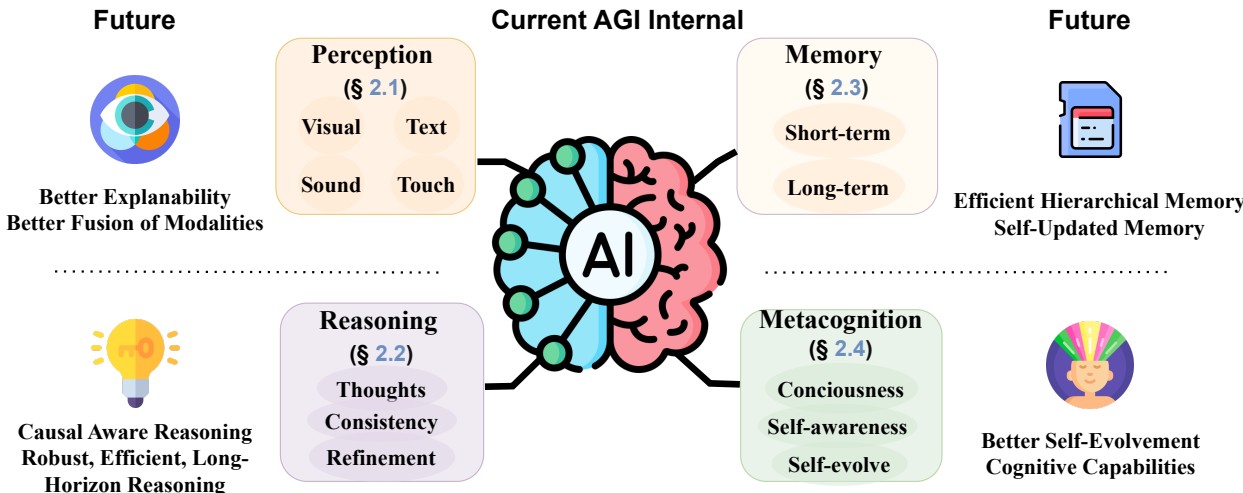

Figure 2: **Current State and Future Expectation of AGI Internal.** We outline four major components for AGI Internal, the mind of AGI: *Perception* (§ 2.1), *Reasoning* (§ 2.2), *Memory* (§ 2.3), and *Metacognition* (§ 2.4), each of which consists of discussions of its current state and future expectation.

## 2   AGI Internal: Unveiling the Mind of AGI

*In the end, we are self-perceiving, self-inventing, locked-in mirages that are little miracles of self-reference.*

*— Douglas Hofstadter, I Am a Strange Loop*

The complexity of the human brain, with its specific functional regions dedicated to distinct aspects of cognition and behavior, offers a compelling analogy for the architecture of AGI systems. Similar to the human brain's division into areas for sensory processing, emotion, cognition, and executive functions, the "brain" of an AGI system could also be fundamentally organized into four main components: *perception*, *memory*, *reasoning capabilities*, and *metacognition*. These components mirror the essential aspects of human cognition and play different crucial roles in creating a truly intelligent system. We summarize the overview of this section in Figure 2, which shows the current state and future expectations of AGI internal. Perception (Sec 2.1) refers to the organization and interpretation of sensory information during the interaction between the AGI and its environment (Wang and Hammer, 2018) and is regarded as a fundamental ability in AGI, which includes vision, hearing, touch, smell, etc. The reasoning (Sec 2.2) of AGI is based on the perception of the environment and executes actions to the environment. The interactions between AGI and the environment containing the acquisition of perception and execution of action would be saved as the memories (Sec 2.3) of AGI. The memories will be utilized for the metacognition (Sec 2.4) of AGI.

### 2.1   AI Perception

*Humans see what they want to see.*

*— Rick Riordan, The Lightning Thief*

**Current State of AI Perception**   Perception refers to the capability of a system to interpret and make sense of the world around it. This involves the processing and analysis of sensory data to construct a dynamic and contextual understanding of its environment.

Natural language, the primary method of human communication, has evolved from its origins in early human interactions to complex systems like large language models (LLMs). These models have expanded

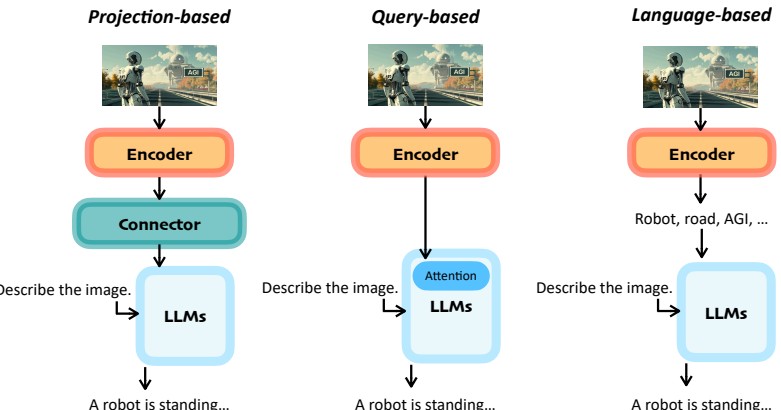

Figure 3: There are three categories for multimodal models with LLM external connections: projection-based, query-based, and language-based.

their capacity to understand and engage in conversations, as well as to perform creative tasks. However, text alone may not fully capture the depth of real-world experiences (Harnad, 1990; Bisk et al., 2020; Tu et al., 2023b), underscoring the importance of multi-modal intelligence that incorporates images, video, and audio for richer human-machine interaction. The transition from traditional LLMs to multi-modal models represents a significant technological leap, facilitating more lifelike interactions across various inputs. This shift, highlighted by recent developments in multi-modal LLMs (OpenAI, 2023b; Team et al., 2023; Li et al., 2023h; Dai et al., 2023; Ye et al., 2023; Zhu et al., 2023a; Liu et al., 2023d; Su et al., 2023; Chen et al., 2023j; Li et al., 2024e; He et al., 2024; Laurençon et al., 2024; Chu et al., 2023), addresses the constraints of language-only comprehension and opens the door to addressing complex challenges that involve multiple forms of data. Integrating various models should adhere to two principles: 1) understanding "how" to incorporate external modal information and ensuring a seamless integration of different modules; 2) determining "what" information to use for preserving the integrity of the original models and enhancing overall capabilities.

The primary objective of utilizing *off-the-shelf* LLMs and multi-modal encoders is to establish a seamless connection between them. This connection can either be external, aligning multi-modal knowledge without altering the existing model structure, or internal, allowing for a more intricate interaction between LLMs and other modal encoders (Yin et al., 2023). These methods often require extensive training, such as creating a learnable interface to link the LLM with non-linguistic modalities, particularly vision. Like LLM pre-training and fine-tuning, Multi-modal LLMs (MLLMs) follow a two-stage training paradigm based on a pre-trained LLM and adapt the process to the multi-modal domain. The first stage, known as the vision-language alignment stage, aims to enable the language model to comprehend visual tokens. The second stage involves multi-modal instruction tuning to align the model with human perceptions. These stages have clear categories based on the combination architectures between the LLM and multi-modal encoders.

- **External Connection of Modalities.** The external approach is based on the idea of bridging the vision branch and LLMs with extra structures and existing models.

  1. *Projection-based*: the modality connector exists outside both the LLMs and multi-modal encoders can be quite straightforward with simple linear projections (Zhu et al., 2023a; Liu et al., 2023d; Su et al., 2023; Chen et al., 2023j; Li et al., 2024e) or incorporating relatively complex selection method (Gao et al., 2023a; Zhang et al., 2023e; Luo et al., 2023; Han et al., 2023b; Fu et al., 2024a). This type of MLLM usually activates the projection layer and/or the LLMs for two stages of alignment training.

  2. *Query-based*: these MLLMs employ a more intricately designed connector but still stand outside of LLMs and multi-modal encoders. This type of model essentially leverages an attention-like interaction between a learnable variable and the vision tokens (Dai et al., 2023; Li et al., 2023h; Ye et al., 2023; He et al., 2024). Since their connectors can learn more complex data patterns than simple projection-based ones, activating the connector alone can also obtain superior multi-modal performance.

3. *Language-based:* language as the interface (Wei et al., 2023; Berrios et al., 2023) is a popular direction for bridging all these *off-the-shelf* models as a holistic and comprehensive one. These methods utilize various pre-built modules for generation and other tasks, with LLMs primarily directing module coordination (Yang et al., 2023d; Li et al., 2023f; Gao et al., 2023a). One main advantage of leveraging tools is that these systems can be more flexible planners Yang et al. (2023d); Wang et al. (2023h); Zhou et al. (2023a); Chen et al. (2023f) for making decisions or artists (Shilong Liu, 2023; Sun et al., 2023b; Huang et al., 2023b; Fu et al., 2023c) for creating versatile multi-media contents with the language as the bridge. One prominent and recent approach is that the GPT-4V model (OpenAI, 2023b) can also generate vivid images by connecting state-of-the-art generators (Rombach et al., 2022). While these approaches offer a wider range of technical solutions for diverse tasks (Wang et al., 2023j;c; Chen et al., 2023e), they generally lack the depth in achieving comparable performance compared with interface-based ones.

- **Internal Connection of Modalities.** Another direction for bridging the multi-modal encoder and the LLM lies in twitching the LLM interblocks.

  1. *Cross-attention-based*: Flamingo (Alayrac et al., 2022) proposed the well-known perceiver with additional cross-attention mechanism inside the attention block of the LLMs. Several variants of Flamingo (Li et al., 2023l; Gong et al., 2023) also use the same or similar framework for tuning the MLLMs.
  2. *Autoregressive:* MLLMs like Fuyu (Bavishi et al., 2023) and its variants (Li et al., 2023m) take vision token as the language token from the pre-training stage and use the same autoregressive training loss to update the whole model parameter.

- **Additional modalities for MLLMs** While earlier models predominantly focused on visual inputs and textual outputs, recent developments have broadened to include diverse modalities in both input and output forms. Regarding inputs, with appropriate modal encoders and training data (Girdhar et al., 2023; Zhu et al., 2023c), LLMs can now comprehend video, audio (Zhang et al., 2023g; Chen et al., 2023d; Lyu et al., 2023; Zhang et al., 2023h), and multiple non-linguistic modalities concurrently (Su et al., 2023; Han et al., 2023b;a), making this approach scalable and accessible. Regarding outputs, recent research has shifted toward creating hybrid content that goes beyond mere text generation. LLMs have evolved from initially retrieving images and generating text (Koh et al., 2023b; Chen et al., 2023j) to producing both visual and textual content. The detailed technical paths of generating images and texts include autoregressive tuning of image-text data with unified representations Sun et al. (2023d); Zheng et al. (2023a); Liu et al. (2024b) and symbolic tuning that transforms text features into image generative models like Stable Diffusion (Koh et al., 2023a; Ge et al., 2023a). Moreover, recent advancements in vision have opened up scalable methods for generating content without text, enhancing the potential for generalizing and scaling vision-only models to generative tasks (Bai et al., 2023; El-Nouby et al., 2024). This opens up the possibility of discovering similar "AGI phenomena" when scaling foundation models in other modalities than language.

**AGI-level Perception** Current models of perception are still limited by their limited modality and lack of robustness. To address these limitations, we propose several potential future research directions:

- **The diversification of modalities is essential for integrating multiple data types and improving model capabilities.** It is crucial to explore less common modalities, such as graphs, and to integrate multiple modalities, such as images, audio, and video simultaneously (Han et al., 2023b;a). This will require carefully designed modules, high-quality data, and a balanced approach to managing the interplay between different modalities and their relationship with language. For example, while GPT-4V can only handle language and visual information, the recent Gemini (Team et al., 2023) model expands its capacity to a wider range of audio and video. Potential methods for incorporating other modal perceptions: a unified modal representation tool like ImageBind (Han et al., 2023b), LangaugeBind (Zhu et al., 2023c) could bridge the modal gap and lessen the burden of learning from other modalities. Existing models that incorporate these tools have shown promising results in efficiency and task performance (Su et al., 2023; Han et al., 2023b).

- **Encouraging multi-modal systems to be more robust and reliable.** As more comprehensive benchmarks covering not only general situations but also challenging inputs like math problems (Yue et al., 2023), counterfactual instructions (Zhang et al., 2023m), and attack strings (Qi et al., 2023; Zhao et al., 2023b) emerge, it becomes evident that multi-modal systems, especially the smaller ones, generally fall short in performance when facing adversarial examples and are heavily language-biased (Tu et al., 2023a; Cui et al., 2023), lacking the reasoning ability under out-of-distribution situations like multi-panel images (Fan et al., 2024), sketches (Tu et al., 2023a), long sequence images (Wang et al., 2024b). These observations pose potential risks in real-world applications. To address these challenges and build more robust multi-modal AGI models, several strategies in terms of the employed learning data are considered. Future research could benefit from incorporating adversarial examples into training (Liu et al., 2023d) or involving increased diversity of training data instruction formats (Dai et al., 2023).

- **Explainable multi-modal models point out the direction for future improvement.** Unlike traditional models, multi-modal models involve complex interactions between different modalities, making it essential to unravel their inner workings to understand and create stronger multi-modal ones. To address this, research efforts have focused on providing explanations during training or generation, offering insights into model performance and reasoning. Methods like probing model performance with diverse training data have been explored (Liu et al., 2023d; Zhao et al., 2023c; Tu et al., 2023c). Additionally, the Gemini (Team et al., 2023) team enhances user trust and understanding of the AI's reasoning process by providing explanations of the generation (Team et al., 2023). Another aspect of improving multi-modal models is increasing transparency. This involves identifying the specific model components or configurations that contribute to the system's abilities (*e.g.*, vision encoder, connector, or training paradigms) (Wang et al., 2023d; He et al., 2024). Studies have also specifically investigated the impact of different modality processors on the overall model performance (Lin et al., 2023a; Tong et al., 2024b). As multi-modal models advance, future research must prioritize explainability and transparency. This will enable us to take the full potential of these powerful AI systems while ensuring their responsible and ethical use. For example, future research avenues could explore strict controlled experiments for training AI models to decompose each part (Tong et al., 2024a) or probing model components to find the most effective module (Zhao et al., 2023c).

## 2.2 AI Reasoning

> *All our knowledge begins with the senses, proceeds then to the understanding, and ends with reason. There is nothing higher than reason.*
>
> — *Immanuel Kant, Critique of Pure Reason*

Reasoning is the cognitive process of drawing conclusions or making decisions based on available information, logic, and prior knowledge. It involves evaluating evidence, identifying relationships, and applying rules or principles to solve problems (Fagin et al., 2004; Huang and Chang, 2022). AI reasoning refers to the ability of AI systems to simulate this process, enabling machines to understand situations, infer conclusions, and make decisions in a way that mimics human reasoning.

**Current State of AI Reasoning** Substantial research indicates that reasoning capabilities have emerged in large machine-learning models. Large Language Models (LLMs), including GPT-3 (Brown et al., 2020), LLaMA 2 (Touvron et al., 2023), and PALM 2 (Anil et al., 2023), have unlocked flexible zero-shot and few-shot reasoning capabilities across various NLP tasks (Kojima et al., 2022). Large Visual Language Models (LVLMs) such as GPT-4 with vision (OpenAI, 2023a) and Gemini (Team et al., 2023), have advanced this progress by effectively integrating vision and language reasoning.

Numerous strategies have been developed to elicit effective and efficient reasoning without updating the model. These methods have substantially improved model performance across a wide range of tasks, including arithmetic, commonsense, symbolic reasoning, and challenges in both simulated and real-world settings.

- **Navigating through thoughts.** Chain of Thought (CoT) (Wei et al., 2022b; Kojima et al., 2022) generates a sequence of intermediate reasoning steps, known as "thoughts," to enable models to decompose multi-step problems and allocate additional computation to more complex tasks. This offers an interpretable insight into the model's reasoning process, helping to comprehend how an answer is derived and identify where errors in reasoning might occur. Tree of Thoughts (ToT) (Yao et al., 2023) employs tree-based search algorithms to navigate through "thoughts" for deliberate problem-solving. This allows LLMs to explore multiple reasoning paths and perform deliberate decision-making, including looking ahead or backtracking when necessary. Graph of Thoughts (GoT) (Besta et al., 2023) organizes information into a graph structure, where "thoughts" are vertices, and edges correspond to dependencies between these vertices. This graph-based organization facilitates more intricate integration and manipulation of thoughts, allowing for the creation of more sophisticated reasoning pathways and the incorporation of feedback mechanisms. Program of Thoughts (PoT) (Chen et al., 2022) leverages language models to express the reasoning process as a program, delegating the computation to an external computer that executes the generated programs to obtain the answer. This separation of computation from reasoning improves performance on problems that demand highly symbolic reasoning skills.

- **Self-consistent reasoning.** Self-consistency (Wang et al., 2022a) samples a diverse set of reasoning paths and selects the most consistent answers. This method overcomes the constraints of greedy decoding by balancing open-ended and optimal text generation, utilizing the diversity of reasoning paths to achieve more reliable outcomes. Maieutic Prompting (Jung et al., 2022) induces a tree of explanations abductively and recursively, then frames the inference as a satisfiability problem over these explanations and their logical relations. Progressive-Hint Prompting (Zheng et al., 2023b) employs previously generated answers as hints to progressively guide toward the correct answers, enforcing a level of self-consistency with earlier responses.

- **Additional prompting strategies for enhanced reasoning.** Many other prompting methods have been developed to improve the reasoning abilities of LLMs. Complexity-Based Prompting (Fu et al., 2022) creates rationales with more reasoning steps with an example selection scheme. Auto-CoT (Zhang et al., 2022b) samples questions with diversity and automatically generates reasoning chains to construct demonstrations. Least-to-Most Prompting (Zhou et al., 2022b) breaks down a complex problem into a series of simpler subproblems and then solves them in sequence. Decomposed Prompting (Khot et al., 2022) decomposes tasks into simpler sub-tasks and dynamically delegates them to sub-task-specific models. ToolLLM (Qin et al., 2023b) and ToRA (Gou et al., 2023b) integrate natural language reasoning with the use of external tools, significantly enhancing their ability to perform complex reasoning. Collaboration mechanisms between multiple agents, such as debate (Du et al., 2023), reflection (Zhang et al., 2023l), voting (Li et al., 2024g), or role-playing as different characters (Qian et al., 2023a; Zhou et al., 2023d), can further enhance their reasoning performance.

- **Dynamic reasoning and planning.** ReAct (Yao et al., 2022) prompts LLMs to generate reasoning traces and action plans in an interleaved manner. This synergy between reasoning and action enables dynamic reasoning, creating, maintaining, and adjusting action plans while interacting with external environments like Wikipedia. This interaction allows the integration of additional information into the reasoning process and addresses issues like hallucination and error propagation common in chain-of-thought reasoning. "Describe, Explain, Plan and Select" (DEPS) (Wang et al., 2023b) enhances plan reliability by incorporating a dynamic feedback loop that includes description, explanation, and plan adjustment stages, significantly improving error correction and planning efficiency. Inner Monologue (Huang et al., 2022b) underscores the utility of feedback-informed planning, demonstrating improved task completion and adaptability in diverse environments by dynamically incorporating feedback to refine and adjust plans in real-time. ProgPrompt (Singh et al., 2023) enables the generation of executable task plans that are both contextually relevant and adaptable to the robot's capabilities and the environment's state by structuring prompts as programmatic instructions and incorporating environment state feedback through assert statements. LLM+P (Liu et al., 2023b) combines the natural language processing strengths of LLMs with the precise problem-solving skills of classical planners, providing optimal solutions for planning problems that involve language description. Thought Rollback (Chen and Li, 2024) introduces a rollback mechanism that allows LLMs to revise prior steps based on error analysis, fostering adaptive

reasoning by dynamically adjusting thought structures to improve problem-solving accuracy. Parameter-efficient finetuning methods (PEFTs) (Xu et al., 2023c) adapt large language models by updating only a small subset of their parameters, reducing memory usage and computational costs while achieving performance comparable to full-model finetuning, whereas ReFT methods (Wu et al., 2024a) operate on frozen model representations rather than weights, learning task-specific interventions that efficiently steer model behavior during inference.

- **Reflection and refinement.** Self-refine (Madaan et al., 2024) employs iterative generation, self-generated feedback, and refinement, enabling large language models to adjust based on feedback after each generative cycle. Reflexion (Shinn et al., 2023) expands on the ReAct framework by integrating an evaluator to assess action trajectories and utilizing an LLM to generate verbal self-reflections to provide feedback for future trials. CRITIC (Gou et al., 2023a) utilizes external tools, such as knowledge bases and search engines, to validate the actions produced by LLMs, then employs external knowledge for self-correction to minimize factual errors.

- **Integrating language models, world models, and agent models.** While language models fall short of consistent reasoning and planning in various scenarios, world and agent models can provide essential elements of human-like deliberative reasoning, including beliefs, goals, anticipation of consequences, and strategic planning. The LAW framework (Hu and Shu, 2023) suggests reasoning with world and agent models, with language models serving as the backend for implementing the system or its components. This framework combines three models in a cognitively grounded way, fostering more robust and versatile reasoning capabilities. Within this framework, Reasoning via Planning (RAP) (Hao et al., 2023) prompts an LLM to function as an agent model, guided by the same LLM acting as the world model, which predicts the next state of the reasoning after applying an action to the current state. BIP-ALM (Jin et al., 2024) and LIMP (Shi et al., 2024b) use language models as the planner in agent models, leading to an improved Theory of Mind capacity compared to using language models to infer other agents' mental states directly. Recent studies have explored the potential for using language models to generate goals (Xie et al., 2023) or rewards (Yu et al., 2023; Kwon et al., 2023b; Ma et al., 2023a) in agent models to guide planning. Integrating these approaches, neural-symbolic methods can bridge the gap between the abstract reasoning facilitated by LLMs and the structured decision-making processes inherent in world and agent models. Logic-LM (Pan et al., 2023) apply symbolic execution on logical reasoning. Symbol-LLM (Xu et al., 2023b) unifies neural-symbolic applications under a Symbol+Delegation setting.

- **Reasoning and planning of embodied agents.** Several studies have proposed methods for reasoning procedures in embodied agents, enhancing their ability to execute tasks and interact with their environment and other agents in a more sophisticated manner. Voyager (Wang et al., 2023h) is an embodied agent in the Minecraft game that uses iterative prompting for dynamic reasoning and skill acquisition. It begins with an automatic curriculum that suggests tasks based on the agent's capabilities and the world state. Then Voyager creates code for these tasks and enters a cycle of execution, feedback assessment, and code refinement. This loop of reasoning, supported by a self-verification module, guarantees task completion and continuous learning. Generative Agents (Park et al., 2023) are language agents grounded in a sandbox game that affords interaction with the environment and other agents. Their memory stream records experiences in natural language, enabling moment-to-moment behavior informed by the relevance, recency, and importance of memory objects. Through reflection, the agents synthesize these memories into higher-level inferences, leading to the creation of coherent plans. While executing these plans, agents continuously reason over recent observations to maintain or adjust the plan.

**AGI-level Reasoning** While current systems exhibit impressive reasoning skills across various tasks, they also have several substantial flaws and challenges.

- **Foundation models need to learn causation for better understanding and generalization.** The foundation models rely heavily on patterns identified in their training data, which do not always capture the depth and breadth of human knowledge and experiences. Furthermore, these models often operate based on patterns extracted from data without truly comprehending the underlying causal relationships. Zečević et al. (2023) describe how LLMs might superficially replicate causal relationships but lack the

underlying causal mechanisms, leading them to be more like "causal parrots" rather than genuinely causal models. Jin et al. (2023a) presents a challenging dataset for causal reasoning and suggests that LLMs may still be far from reasoning reliably about causality. Future advancements in AGI should focus on learning causation over correlation, thereby achieving better generalization and deeper understanding.

- **AGI must address the challenge of complex and long-context reasoning.** Current models face significant difficulties with complex, multi-step reasoning tasks. As discussed earlier, many strategies have been developed to mitigate this issue. However, these strategies often require explicit guidance or careful framing of the problem, which may become unnecessary in the future. Even with these approaches, it remains a challenge for models to process information across long contexts and maintain coherent and logical reasoning throughout the reasoning tasks (Srivastava et al., 2022).

- **AGI should tackle challenges in hallucination, uncertainty assessment, and ambiguity handling, improving performance and safety.** Current models are susceptible to hallucination, where they generate content that is either nonsensical or unfaithful to the provided source content (Ji et al., 2023a; Li et al., 2023d). This tendency hampers performance and raises significant safety concerns in real-world applications. Moreover, these models often struggle to accurately assess their uncertainty and effectively communicate it in their outputs, which can lead to results that are potentially misleading (Zhou et al., 2023b). Additionally, they struggle to handle ambiguity, an issue that can complicate their usability in complex scenarios (Liu et al., 2023h).

- **AGI should get better at social reasoning to enhance interactions with humans and other agents.** Current AI models lack a robust Theory of Mind, the ability to understand the mental states of others (Sap et al., 2022; Ullman, 2023; Jin et al., 2024; Shi et al., 2024b). Improving this capability is essential for AGI systems to safely and effectively interact with humans and other agents in an open-ended manner. Understanding social cues and norms is central to this development, as it allows AGIs to interpret and respond to implicit communication and behavioral expectations in varied contexts (Puig et al., 2020; Zhang et al., 2023b). Advancements in social reasoning in AGI systems could lead to more empathetic and context-aware technologies, ensuring that these systems engage harmoniously and meaningfully in human societies.

- **AGI should solve the challenge in explainability and transparency, thereby enhancing their reliability in decision-making.** Currently, most AI systems lack these qualities, making it difficult to understand how they arrive at specific conclusions or answers (Chen et al., 2023k). Techniques that aim to elicit reasoning in natural language do not consistently align with the actual reasoning processes used by the models, and the explanations generated can be systematically misleading (Bowman, 2023). This limitation hinders their reasoning abilities and creates significant challenges when decision-making requires justification or auditing, particularly in fields like healthcare or law. Recent work on sparse autoencoders, specifically using them to address polysemanticity in neural networks, has shown promise in enhancing the explainability of AI models (Cunningham et al., 2023; Gao et al., 2024; Templeton, 2024). These autoencoders help in disentangling and isolating meaningful features within neural networks, leading to more interpretable and mono semantic features that are easier to understand. Incorporating similar techniques to interpret neural networks, and developing more techniques to explain different decision-making process of AGI systems can facilitate more trustworthy and accountable AI applications.

- **Future AGI systems aim for dynamic reasoning across domains, ethical and efficient planning, and human-like intelligence at unparalleled scales and speeds.** We are still far from achieving AGI-level capability that allows reasoning and planning across varied domains without retraining or human oversight (Saparov et al., 2024). The journey involves enhancing AI systems to transfer knowledge and skills across vastly different areas, enabling them to address unforeseen situations efficiently. A key development focus is creating algorithms that can plan at various levels of abstraction, from broad strategic goals to the specifics of detailed actions. Additionally, AI systems urgently need to become better at managing resources—such as time, energy, and costs—more efficiently during the planning phases. Equally critical is ensuring that these planning processes adhere to ethical standards and safety regulations, especially in sensitive sectors, to avoid misuse or unintended outcomes. Advancements in these areas will collectively move us closer to realizing AGI with robust, versatile planning capabilities.

- **More advanced reasoning abilities are required to solve complex real-world tasks.** In the future, enhancements in prompting techniques or task-framing methods promise to significantly boost the reasoning capabilities of foundational models. On the other hand, future advancements might eliminate the need for complex prompting to aid in reasoning, with these aids being "implicit". A future AGI system could potentially emulate any grounding, learning, and decision-making by listing all the possible actions and simulating and evaluating each one before executing its actual decision-making process. Even more audaciously, it may simulate these implicitly in neurons without any intermediate reasoning in context. For such a system to simulate this flawlessly, it would require an exceptionally realistic world model.

- **Future AGI systems will be able to understand context, infer causality, and apply advanced logical planning dynamically across diverse domains.** By synthesizing vast amounts of information and applying deliberate planning, they can generate innovative solutions to formulating creative hypotheses, making sophisticated moral judgments, predicting the outcomes of novel scenarios, and continuously learning and refining their understanding of the world. Essentially, these future AGI systems would not only excel in processing and generating information but will also be capable of understanding and interacting with the world in a manner deeply analogous to human intelligence, yet at a scale and speed that greatly surpasses human capabilities.

### 2.3 AI Memory

*Remembrance of things past is not necessarily the remembrance of things as they were.*

*— Marcel Proust, In Search of Lost Time*

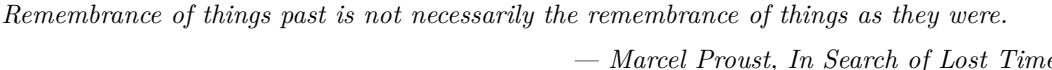

Language and vision models, by their nature, are stateless; they do not maintain information between interactions. However, advanced agents differ in that they can manage internal or external memory, enabling them to engage in complex, multi-step interactions (Sumers et al., 2023; Zhang et al., 2024a). This memory stores intermediate information, domain-specific or broad knowledge, and sequences of the agents' previous observations, thoughts, and actions, among others. It assists agents in utilizing previous knowledge or experiences for reasoning, planning, and self-improvement.

**Current State of AI Memory** We examine the current state of AI memory, focusing on three key aspects: memory management, which determines what and when to store; memory representation, which defines how information is structured; and memory utilization, which addresses how to apply and use the memory efficiently and effectively.

- **Memory management.** Memory is categorized by duration into short-term and long-term memory.

  1. *Short-term memory*: Short-term memory plays a crucial role in maintaining information needed for current decision-making processes. A notable example is in-context prompting, which uses the foundation models' own context as a form of short-term memory. This approach can provide additional information or examples (Wang et al., 2020), or can be used to generate intermediate reasoning (Nye et al., 2021; Wei et al., 2022b). More broadly, short-term memory encompasses all immediate data essential for decision-making. This includes: (1) real-time data collected or processed by perception modules; (2) immediate outputs from reasoning, planning, and self-evolution modules; and (3) information actively retrieved from long-term memory. These elements collectively are synthesized to guide and inform subsequent actions.

  2. *Long-term memory*: Long-term memory can be broadly classified into two main types: experiences and knowledge. Experiences encompass a range of elements such as past observations, thoughts, actions, and more. This rich collection of experiences serves a critical function in decision-making processes. By retrieving relevant experiences, agents can gain additional information necessary for reasoned judgment, understand feedback from past actions, and achieve a level of generalization in their understanding and reasoning. For example, Reflexion (Shinn et al., 2023) reflects on task feedback signals and maintains them as textual summaries. These summaries are directly incorporated into the context of subsequent episodes, aiding in performance enhancement. Generative agents (Park et al., 2023) document their

experiences in natural language and retrieve memories using a mix of relevance (embedding-based), recency (rule-based), and importance (reasoning-based) criteria.

Knowledge represents an agent's understanding of the world and itself, which enhances its reasoning and decision-making capabilities. Knowledge can originate from two sources. First, AI agents can collect and assimilate knowledge from experiences, integrating new information or skills into their existing knowledge. For example, Voyager (Wang et al., 2023h) maintains a continuously expanding skill library of executable codes for preliminary actions to accomplish tasks. Second, AI agents or models can utilize external knowledge bases. For instance, ReAct (Yao et al., 2022) employs Wikipedia APIs to acquire external knowledge when agents lack information during their activities. ChatGPT Browse with Bing enables ChatGPT to access internet knowledge for answering questions, significantly enhancing its ability to provide accurate responses (OpenAI, 2023a). Retrieval-augmented methods (Lewis et al., 2020; Guu et al., 2020; Shuster et al., 2021; Borgeaud et al., 2022) leverage a knowledge base of unstructured text. The "reading to learn" methods (Branavan et al., 2012; Hanjie et al., 2021) utilize domain knowledge from text manuals to influence the policies in reinforcement learning.

- **Memory representation.** Regarding representation, memory is divided into textual memory and parametric memory. Textual memory is the prevalent method for representing memory content today. It can include both unstructured formats like raw natural language and structured forms such as tuples, databases, and more. Alternatively, memory can be represented in a parametric form. Techniques like supervised fine-tuning (Hu et al., 2021), knowledge editing (De Cao et al., 2021; Mitchell et al., 2021) and model merging (Du et al., 2024; Yang et al., 2024; Lu et al., 2024; Goddard et al., 2024) can integrate domain-specific knowledge into model parameters. For textual memory, each inference involves incorporating memory into the context prompt, leading to higher costs and extended processing times during the reading and inference processes. Conversely, parametric memory often incurs greater costs during the writing phase, as fine-tuning models is more challenging than simple text storage. Regarding interpretability, textual memory is generally more transparent than parametric memory, as the natural language provides the most direct means for human understanding (Zhang et al., 2024a).

- **Memory utilization.** There are two common technologies to utilize memories: memory retrieval and long-context LLMs.

  1. *Memory retrieval:* Memory retrieval involves reading information from long-term memory to short-term memory for immediate use. This can be accomplished through rule-based retrieval or retrieval-augmented methods. Rule-based retrieval can search memory using keywords, timesteps, or specific patterns. In retrieval-augmented approaches, the Dense Passage Retriever (DPR) (Karpukhin et al., 2020) creates dense representations of documents and retrieves the most relevant documents based on their prior probability using Maximum Inner Product Search (MIPS). The Retrieval-Augmented Language Model pre-training (REALM) (Guu et al., 2020) integrates unsupervised pre-training of a knowledge retriever with masked language modeling, enabling direct retrieval of documents to supplement language predictions. Retrieval-Augmented Generation (RAG) models (Lewis et al., 2020; Shuster et al., 2021; Borgeaud et al., 2022) employ a non-parametric memory, such as a dense vector index of Wikipedia, accessed via a pre-trained neural retriever (e.g., DPR). These documents are processed by a seq2seq model, which conditions its output generation on both the input and the retrieved documents. Both the retriever and seq2seq modules, initialized from pre-trained models, are jointly fine-tuned, allowing both retrieval and generation to adapt to downstream tasks.

  2. *Long-context LLMs:* The expansion of the context window in long-context LLMs opens up new avenues for models to access their long-term memory. Works like Ring Attention (Liu et al., 2023j) and LongRoPE (Ding et al., 2024b) greatly reduce the time and cost of long context inference by improving the operation mechanism and storage method of attention. More powerful GPUs with enhanced memory capabilities and further breakthroughs in memory-efficient attention mechanisms (Dao et al., 2022; Tay et al., 2022), allowing the context window for pre-trained LLMs to increase from 1024 tokens in GPT-2 (Radford et al., 2019), to 8192 in GPT-4 (Achiam et al., 2023), and now exceeding 16K tokens. With these expanded context windows, AI systems can more effectively store and recall knowledge and experiences within their context, enabling faster and more comprehensive context-based reasoning.

**AGI-level Memory**  Achieving AGI-level memory requires advanced management of vast, dynamically organized information, improved utilization of memory for reasoning and planning, and the ability to autonomously update and enrich the memory base.  It involves human-like deliberate use of memory, yet surpasses human capacities, allowing for more comprehensive and intricate recall.

- **Future AGI will efficiently manage diverse and hierarchical memories, ensuring privacy, collaboration, and scalability.**  Current AI agents face challenges in building hierarchical memory and seamlessly incorporating information across various formats.  Future AGI systems are anticipated to excel in handling diverse forms of memory, such as embeddings, videos, documents, and databases, both efficiently and effectively.  They will also need to address different levels of memory permissions: local memory is essential for preserving privacy, while shared memory, in centralized or decentralized structures as required, is necessary for collaborative efforts and distributed processes. The architectures employed for memory management are expected to be highly organized and scalable. These systems will likely feature advanced algorithms for categorizing and indexing information, allowing agents to efficiently retrieve and record a wide spectrum of experiences and knowledge. Additionally, they may dynamically update and reorganize their memory structures, ensuring optimal storage and retrieval of information.

- **Future AGI will enhance memory utilization through the integration of retrieval and advanced reasoning, enabling more human-like intelligence and adaptability.**  Beyond simple memory retrieval, future AGI systems could refine memory utilization by intricately combining retrieval processes with advanced reasoning that strategically synthesizes and applies information in context-appropriate ways.  Beyond fixed implementations, the retrieval procedures could be learned or updated to adapt to changing circumstances.  The ability to access and apply relevant information from their memory in real-time would be a significant step towards more human-like intelligence, enabling these systems to respond to new situations with a high level of understanding and adaptability.

- **Future AGI will autonomously update their knowledge, enabling continuous learning and adaptation while ensuring safety.**  Unlike existing retrieval-augmented models that primarily rely on pre-existing, human-generated content, future AGI systems could autonomously generate, evaluate, and incorporate new content into their memory banks. These updates should encompass knowledge essential for performance enhancement and experiences the systems can draw upon. This concept is closely linked to self-evolve, which we will discuss later. It would allow AGIs to learn from their own experiences and insights, continually enriching and updating their knowledge base. In a constantly changing world, this capability will also enable the systems to adapt quickly given new information and unlearn outdated knowledge. A crucial aspect is to guarantee the safety of the memory updates, ensuring that no harmful information is written that could lead to contamination. Designing safety constraints for autonomous AGIs involves creating robust validation protocols that assess the truthfulness, relevance, and impact of new information before integration. We can implement expert systems to periodically review updates, use anomaly detection to flag outliers and potentially harmful data, and employ additional methods to enforce these safety constraints.

### 2.4  AI Metacognition

> *I am no bird, and no net ensnares me: I am a free human being with an independent will.*
>
> — *Charlotte Brontë, Jane Eyre*

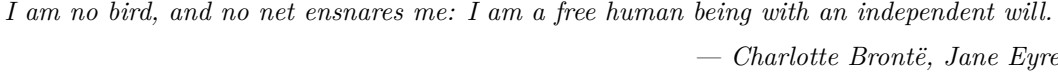

Metacognition (Choudrie and Selamat, 2006) of humans involves key cognitive and emotional skills such as understanding complex situations, self-awareness, and motivation to innovate.  These abilities help share implicit knowledge and drive personal growth.

The development of AGIs with such advanced metacognition provokes a fundamental inquiry: are we, in our pursuit of artificial intelligence, on the verge of creating a new form of life?  The implications are far-reaching, as introducing entities with self-awareness and autonomous decision-making capabilities could redefine the boundaries of life and intelligence.  This tantalizing horizon calls for meticulous ethical consideration and

regulatory scrutiny to ensure that the evolution of AGI contributes positively to human society and does not inadvertently engender a paradigm shift with unforeseen consequences.

**Current State of AI Metacognition** The discourse on metacognition extends into the realm of AGI, where such capabilities are deemed equally critical. For AGI systems, metacognition, such as self-awareness (Chella et al., 2020; Subagdja et al., 2021), consciousness (Dehaene et al., 2021), the capacity for self-evolution (Floreano et al., 2004; Tao et al., 2024), and theory of mind (Cuzzolin et al., 2020), are posited to be foundational for bridging the gap towards achieving AGI. These internal abilities can enable AI systems to autonomously learn, efficiently complete tasks, and align more closely with human intentions.

- **Self-Awareness in AGI.** Developing self-awareness in AI, particularly within the realm of robotics (Scassellati, 2002), hinges on intricate concepts such as self-reflection (Shinn et al., 2024), meta-cognition (Langdon et al., 2022), and self-distancing (Kross and Ayduk, 2017). These concepts are integral to constructing social robots equipped with cognitive architectures that support self-description, the utilization of personal pronouns, and the ability to respond to self-focusing cues, which are fundamental for facilitating effective human interactions and environmental navigation. As AI systems evolve, the philosophical and practical considerations of equipping them with human-like traits of conscientiousness are gaining traction, heralding a burgeoning field of research (Huang et al., 2023c). For a thorough understanding of this area, interdisciplinary approaches that intertwine psychology, artificial intelligence, and ethics are instrumental.

  To seamlessly integrate into the human-centric world, AGI must possess an acute awareness of the beliefs, intentions, and desires of both themselves and others. This "Theory of Mind" (Premack and Woodruff, 1978) is a meta-ability that enables AGIs to understand and predict behaviors, facilitating smoother human interactions. This comprehension will allow for more nuanced and informed decision-making by AGIs, particularly in complex social contexts.

- **AGI holding certain persona.** Recent advancements reveal that LLMs can exhibit consistent personality traits, such as those categorized by the Big Five or MBTI frameworks, with models like ChatGPT often exhibiting traits aligned with the ENFJ type (Huang et al., 2023c). These models also tend to display certain cognitive thinking styles, with evidence suggesting an inclination towards holistic thinking in ChatGPT's responses (Jin et al., 2023c). Research efforts are increasingly directed towards intentionally imbuing LLMs with specific personalities, enabling them to demonstrate a variety of behaviors that are both diverse and verifiable (Caron and Srivastava, 2022; Jiang et al., 2022b).

- **AGI metacognition ability in self-evolving.** While the aforementioned research defines AGI in terms of easily measurable capabilities such as reasoning (Butlin et al., 2023; Morris et al., 2024), it may overlook the potential importance of meta-cognitive abilities such as self-evolution or self-awareness. Studies predominantly showcase this through the agent's iterative adaptation via task execution (Le, 2019; Wang et al., 2023e), code execution (Gao et al., 2020), or feedback from physical simulations (Qian et al., 2024; Xu et al., 2023a). Other strategies for self-evolution include prompt adaptation and optimization (Wang et al., 2023h; Aksitov et al., 2023), continuous improvement through error identification and self-reflection, and memory retrieval as a mechanism for short- or long-term learning. These approaches mainly emphasize the iterative refinement of tasks within a loop-structured framework based on LLMs. In contrast, recent advancements propose methodologies that address inter-task agent self-evolution, highlighting the significance of leveraging past experiences to effectively evolve AI systems (Qian et al., 2024; Xu et al., 2023a).

These traits are crucial for various reasons. First, self-awareness could enhance AGI's adaptive problem-solving abilities by allowing it to accurately assess its strengths and limitations, thus facilitating adjusting its strategies in real time when faced with new challenges. Second, the capacity for ethical and moral decision-making is increasingly imperative as AGI becomes more entwined with societal functions, necessitating a self-awareness component to enable navigation through complex moral dilemmas and ensure alignment with human values. Furthermore, the potential for AGI to autonomously evolve and adapt without human guidance promises greater efficiency and capability in the long term, possibly leading to an exponential

increase in their abilities—a key characteristic of true AGI. Lastly, incorporating autonomous consciousness in AGI could yield more natural and effective human-AI interactions, enhancing collaboration and developing more intuitive user interfaces, particularly in scenarios requiring deep integration into human teams or societal structures.

**AGI-level Metacognition**  The future of AGI metacognition is an exciting realm of possibilities that could dramatically expand the boundaries of artificial intelligence. One key area where AGI could demonstrate significant potential is in enhancing the theory of mind and social reasoning. Current AI struggles with understanding others' mental states, which is crucial for nuanced social interactions. Future AGIs could incorporate multi-modal input and advanced reasoning to model better the beliefs, intentions, and actions of others. For example, an AGI tutor with enhanced social reasoning could deeply understand each student's knowledge, learning style, and motivations, providing hyper-personalized guidance.

- **Future AGI has the potential to achieve genuine self-awareness and consciousness.** AGIs may one day possess deep self-awareness, capable of introspection, reflection, and grappling with existential questions. This would blur the lines between artificial and biological intelligence, raising philosophical and ethical questions. However, uncertainty remains about whether AI could achieve human-like consciousness; it may require integrating metacognition, introspection, and self-perception capabilities. Imagine an AGI companion that doesn't just converse but relates deeply emotionally, sharing in the human condition.

- **Substantial research should focus on AGI's potential for autonomous self-evolution and open-ended learning.** AGIs driven by curiosity and intrinsic motivation could rapidly self-improve, setting goals, innovating strategies, and pushing boundaries. They may exceed human intelligence in certain areas, generating novel insights and breakthroughs that propel fields forward. Picture an AGI scientist who tirelessly conducts experiments, forms and tests hypotheses, and makes discoveries at an unparalleled rate.

As we contemplate the implications and considerations surrounding AGIs with advanced metacognition, we are confronted with profound questions about consciousness, intelligence, ethics, and our place in the world. The exciting potential to integrate AGIs as empathetic companions, insightful advisors, and tireless innovators is balanced by the need to grapple with the implications of creating potentially superior beings and redefining the boundaries between human and artificial intelligence.

Realizing this vision of AGI metacognition will require substantial research and development to close current capability gaps. Nevertheless, the awe-inspiring potential and the philosophical challenges such a future would bring make this an exceedingly important area of AI progress to contemplate and work towards. As we stand on the precipice of this new era, it is crucial that we approach the development of AGI metacognition with a mix of enthusiasm, caution, and deep reflection on the profound implications for our world and our understanding of intelligence itself.

## 3  AGI Interface: Connecting the World with AGI

In the pursuit of developing AGI, a crucial aspect to address is its capability to interact with the external world. This interaction is facilitated through various interfaces that enable AGI systems to perceive, understand, and act within their environment, be it digital, physical, or intellectual. We summarize these three future directions in Figure 4.

### 3.1  AI Interfaces to Digital World

> *The Matrix is everywhere.*
>
> — *Matrix*

The concept of AGI interface into the digital world extends the scope by allowing agents to interact with digital environments, such as the Internet, databases, code, and APIs, and exhibit intelligent behaviors

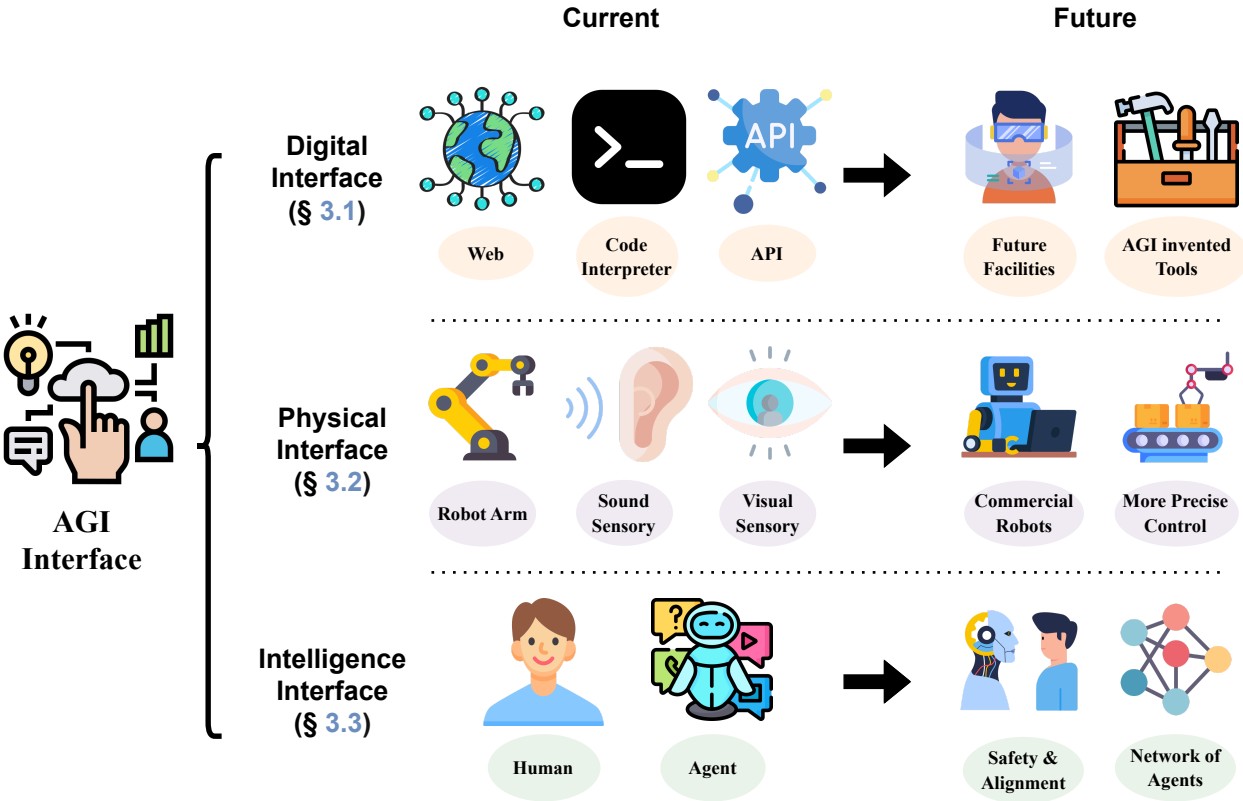

Figure 4: **The Interconnected Spheres of AGI Interface.** In the left part, we present some key elements in three interfaces: Digital (§ 3.1), Physical (§ 3.2), and Intelligence Interface (§ 3.3). On the right side of the figure, we outline several potential future aspects that could be significant.

similar to human-like behavior (Qin et al., 2023a). This interface serves as a crucial bridge for grounding AGI in complex, real-world scenarios, providing an indispensable platform for simulating and interacting with the multifaceted nature of human knowledge and experience. By facilitating AGI's engagement with real-world information structures and problem-solving contexts, this digital world interface accelerates the development of more versatile and robust artificial general intelligence capable of operating effectively across various domains.

**Current State of AI Interface to Digital World**  Digital embodiment enables agents to interact dynamically and flexibly with the world. For instance, agents can utilize various APIs to navigate the web, search for relevant information, and construct personalized knowledge bases, allowing them to update their knowledge and adapt to new situations continuously. This approach drives the development of more advanced AI systems, particularly in natural language processing and reasoning capabilities.

- **Integrating digital tools in LLMs significantly enhances their capabilities and addresses inherent limitations.** Utilizing specialized tools augments their domain-specific expertise and increases decision-making transparency and robustness. The Toolformer model (Schick et al., 2023) demonstrates LLMs' ability to learn and effectively employ various external tools autonomously, with advanced learning methods mirroring human learning processes (Xi et al., 2023; Qin et al., 2023a). Models like Gorilla (Patil et al., 2023) connect LLMs with a wide array of APIs, highlighting the evolution towards greater autonomy and application versatility. LLMs are beginning to create and modify tools (Cai et al., 2024b; Qian et al., 2023b), leading to a future where agents exhibit increased self-sufficiency. This expansion in

tool functionality facilitates multi-modal interactions and broadens the range of tasks LLMs can perform, aligning with the goals of embodied learning research (Zhuang et al., 2023).

- **LLM-based agents and frameworks demonstrate the capabilities of digital embodiment** (Zhou et al., 2024b; Wu et al., 2024b; Deng et al., 2023; Yang et al., 2023b). Mind2Web (Deng et al., 2023) allows for the comprehensive evaluation of agent generalizability in web scenarios, a critical aspect in creating robust and efficient web-based artificial intelligence. Voyager (Wang et al., 2023h) is an embodied agent in the Minecraft game that uses iterative prompting for dynamic reasoning and skill acquisition. To move forward, Generative Agents (Park et al., 2023), grounded in a sandbox game, have a memory stream that records experiences in natural language, enabling informed moment-to-moment behavior. Each of them focuses on different types of digital embodiment.

**AGI-level Interfaces to the Digital World**  While the current state of AGI tool usage is advanced, it highlights several pivotal areas for reaching this goal.

- **AGI systems' creation of novel tools is nascent and limited, requiring a leap beyond human-designed frameworks for true autonomy.** Creating novel tools by AGI systems, as exemplified by the CREATOR framework (Qian et al., 2023c), is a groundbreaking step. Yet, the ability of AGI systems to invent tools autonomously remains nascent. These systems often rely on human-designed frameworks and algorithms, limiting their creative scope. True AGI would require a leap beyond this, enabling systems to ideate and engineer tools independently and intuitively.

- **Extending the scope of digital worlds.** There are still many opportunities to empower AGI systems with interfaces in different modalities and various environments, such as wearable computing, smart environments, mixed-reality settings, and emerging technologies like virtual reality (VR) and extended reality (XR). Although AGI will continue to exhibit promising performance in such interaction tasks, researchers need to explore potential solutions to ensure that AGI can yield beneficial results to humans while minimizing the cost of interaction. AGI should be able to seamlessly integrate with these technologies, leveraging their unique affordances to create more engaging and intuitive interactions. Moreover, AGI systems should be capable of adapting to novel interaction paradigms that may emerge in the future, ensuring that they remain relevant and valuable to users in the long term.

### 3.2  AI Interfaces to Physical World

> *In the twenty-first century, the robot will take the place which slave labor occupied in ancient civilization.*
>
> *— Nikola Tesla*

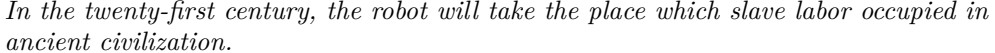

The integration of AI into physical entities is a crucial aspect of the pursuit of AGI. AGI in the physical world emphasizes learning through direct interaction with the environment and making an impact on reality, such as creating or modifying substances. In this section, we will explore the latest advancements in embodied AI in the physical world, including robotic control, navigation, and manipulation.

**Current State of AI Interfaces to the Physical World**  The current state mainly lies in the interaction with robotic functionalities, understanding the potential for more intuitive human-robot interfaces, and emphasizing the importance of real-world datasets in advancing AI's practical applications.

- **Robotic control and action.** Recent advancements in robotic control and action including PaLM-E (Driess et al., 2023), RT-2 (Zitkovich et al., 2023), and Mobile Aloha (Fu et al., 2024b) demonstrate the potential for robots to interpret and execute complex, high-level instructions through natural language, providing a more intuitive interface between humans and robots. SayCan (Ahn et al., 2022) combines the semantic understanding capabilities of PaLM (Chowdhery et al., 2022a) with robotic affordances, enabling robots to understand abstract tasks and execute them in real-world environments. PaLM-E injects embodied observations into the language embedding space of a pre-trained language model

for sequential robotic manipulation planning, while VIMA (Jiang et al., 2022a) processes multi-modal prompts and outputs motor actions autoregressively to control a robot arm. RT2 leverages large-scale, pre-trained vision-language models for robotic control.

- **Robotic navigation and interaction.** Effective navigation and interaction with the environment are essential for embodied AI systems to interface with the physical world. LM-Nav (Shah et al., 2023) combines pre-trained models of language, vision, and action for robotic navigation, marking a significant step towards more intuitive human-robot interaction. VoxPoser (Huang et al., 2023d) utilizes composable 3D value maps and language models for nuanced robotic manipulation, while LLM-Planner (Song et al., 2023a) harnesses LLMs to facilitate few-shot planning for embodied agents, enabling them to follow natural language instructions to accomplish complex tasks within visually-perceived environments. Gervet et al. (2023) presents a comprehensive evaluation of semantic visual navigation methods, finding that modular learning approaches achieve a high success rate in real-world home environments, demonstrating the effectiveness of these interfaces in navigating physical spaces.

- **Understanding and replicating human motion.** To create more natural and intuitive interfaces between humans and robots, it is essential to understand and replicate human motion. MotionGPT (Jiang et al., 2023b) likens human motion to a foreign language that AI can interpret, opening new pathways for understanding and replicating human-like movements in robots. Instruct2Act (Huang et al., 2023a) maps multi-modality instructions to robotic actions using large language models, showcasing the potential of LLMs in interpreting and executing diverse instructions. Furthermore, Perceiver-Actor (Shridhar et al., 2022) introduces a transformer-based model that can be trained end-to-end to map visual observations and natural language instructions to actions for robotic manipulation tasks, further enhancing the interface between human instructions and robotic actions.

  Integrating LLMs into embodied AI represents a vital step forward in the journey towards AGI. The research above demonstrates that LLMs have shown remarkable potential in enabling robots to understand and execute complex instructions, navigate environments, and manipulate objects, creating more seamless interfaces between AI systems and the physical world. By leveraging LLMs' semantic understanding and generalization capabilities, embodied AI systems can achieve intelligence and flexibility that mirrors human capabilities. For manipulation in unstructured environments, an approach that emphasizes integration, embodiment, feedback, and informed assumptions is more effective (Eppner et al., 2016).

- **Datasets.** The work presented by Khazatsky et al. (2024) is a significant contribution to the field of "in-the-wild" robotic manipulation, offering the DROID dataset which captures a wide range of real-world interactions. This dataset is particularly notable for its large-scale in-the-wild setting, featuring diverse environments and tasks that mirror everyday scenarios. Similarly, Li et al. (2024f) introduces BEHAVIOR-1K, another leap in AGI robotics benchmarks, focusing on human-centered activities within a realistic simulation to test the limits of autonomous agents in complex tasks. Both works represent cutting-edge efforts to benchmark and enhance the generalization capabilities of AI in the realm of long-horizon, real-world tasks, bridging the gap between controlled laboratory conditions and the dynamic and unpredictable nature of real-world interactions.

**AGI-level Interfaces to the Physical World**  Looking ahead, the journey toward AGI with embodied intelligence encompasses several key areas for future research and development. A critical area is enhancing contextual and environmental understanding in AI systems. AGI needs to develop the capacity to interpret and adapt to dynamic, real-world environments with a level of sophistication at least comparable to human cognition. This advanced environmental analysis and perception capability is essential for AGI to effectively navigate and interact with complex, ever-changing physical surroundings, mirroring the adaptability and intuition that humans exhibit in diverse situations.

- **Enhancing multisensory integration and interaction capabilities is key to developing more relatable and effective AI systems.** This includes advancing the synergy between visual, auditory, linguistic, and tactile inputs, enabling AI systems to comprehensively perceive their surroundings. Improving interaction capabilities, such as natural language processing and human-like movement, will also make AI systems more relatable and effective in human-centric environments.

- **Advancing affordable robotic manufacturing is crucial for democratizing embodied AI and fostering innovation across sectors.** The development of more advanced yet affordable robotic manufacturing techniques is vital. Lowering the cost of robot production without compromising quality or functionality can democratize access to embodied AI, enabling widespread adoption and innovation across various sectors. This, in turn, could accelerate the deployment of intelligent agents in everyday scenarios, from domestic assistance to industrial automation.

- **Edge computing's efficiency is crucial for embodied AI applications, enabling real-time decisions and immediate responses in dynamic environments.** The efficiency of edge computing, particularly in on-device inference speeds, plays a critical role in the practicality of embodied AI applications. Faster, more efficient processing at the edge allows AI systems to make real-time decisions based on vast amounts of data from their surroundings without the latency associated with cloud computing. This capability is essential for tasks requiring immediate response and adaptation to dynamic environments, such as autonomous driving and real-time navigation in crowded spaces.

### 3.3 AI Interfaces to Intelligence

*The future of work lies in the collaboration between humans and AI, where technology enhances our natural abilities, allowing us to think more strategically and creatively and empowering us to drive innovation in the workplace.*

*— Demis Hassabis, CEO and co-founder of DeepMind*

Integrating AI with other intelligent entities, whether artificial or human, is a critical aspect of achieving AGI. Interfacing with intelligence allows for exchanging knowledge, collaboration, and enhancing overall system capabilities. In this section, we will explore two main categories of interfaces to intelligence: interfaces to AI agents (3.3.1) and interfaces to humans (3.3.2).

#### 3.3.1 AI Interface to Other AI agents

There are generally two categories to improve the overall system for integrating one AGI system with others. The first aspect focuses on the teaching process among AGI models through a sequential interaction between different models. The second emphasizes the simultaneous collaboration between these models, connecting different agents to form a comprehensive and robust AGI system.

**Current State of AI Interfaces to Other AI Agents** The interfaces to other AI agents include both sequential and parallel interactions, where the agents act as teachers, learners, collaborators, or communicators.

- **Agents as teachers and learners.** On one hand, stronger AGI models often act as oracles to provide 'supervision' to inferior ones, from tuning on data from better models (Taori et al., 2023; Gu et al., 2023) to prompt-engineering-based approaches (Huang et al., 2022a; Jiang et al., 2023a; Fu et al., 2023b), there emerges the concept of model knowledge distillation. In the field of language processing, it is common to use a teacher system to label and expand existing data by directly taking the teacher's answer (Gilardi et al., 2023; Hsieh et al., 2023; Li et al., 2022a; Sun et al., 2024b; Ding et al., 2023a) or using more advanced techniques such as CoT prompting (Ramnath et al., 2023; Li et al., 2023g), or creating new data for subsequent models to distill useful and compact knowledge from large-scale data (Li et al., 2023b; Javaheripi et al., 2023). Similar paradigms are applied in computer vision and multimodal domains for better model training and deployment. One of the most prevailing methods is utilizing the GPT-4V model to label answers in various tasks (Shu et al., 2023; Liu et al., 2023d; Li et al., 2023j).

    In conventional solutions, a better AGI model severs as the role of the teacher, however, there is a growing trend for less competent AGI models to provide insights for aligning stronger ones with or beyond human perception, known as superalignment (Burns et al., 2023). This has proven to be effective in empowering higher capacity than the teacher model with a vanilla fine-tuning strategy and distilled data from the

smaller model. Recent studies have begun to explore how leveraging weaker models to enhance the performance of stronger ones can be applied across various domains. Chen et al. (2024a) introduced the innovative idea of iteratively harnessing the capacity of weaker models to improve the efficacy of more powerful counterparts. (Ji et al., 2024) developed an efficient alignment paradigm that learns the correctional residuals between aligned and unaligned responses from a weaker model. (Sun et al., 2024c) employs a less advanced system, trained on simpler tasks, to guide more capable models for tackling hard reasoning tasks (*e.g.*, level 4-5 MATH problems). In the vision domain, (Guo et al., 2024a) presented compelling evidence that weak-to-strong generalization may outperform finetuning methods in certain scenarios. This interaction between different models requires acquiring knowledge from selected models step-by-step, forming the sequential interface.

- **Agents as collaborators or communicators.** On the other hand, off-the-shelf AGI interfaces can facilitate the integration of various models into a comprehensive and efficient system, allowing for simultaneous collaboration and knowledge sharing. In single modality domains, there is a growing trend of combining different language models into an integrated system for improved language modeling, such as the Mixture-of-Expert (MoE) system (Fuzhao Xue and You, 2023). The key advantage of this approach is the significant reduction in deployment efficiency, achieved through model parallelism, dynamic expert model selection, and token routing. As a result, the MoE system can reduce memory requirements and computational budgets compared to similar scale LM systems (Chen et al., 2023g; Chowdhury et al., 2023). By utilizing a gate coordinator to plan for different expert language models, the system can generate specialized and high-quality responses for different aspects of queries, leading to higher efficiency and better handling of tasks (Du et al., 2022). In the multi-modal domain, models with the ability to understand more than one modality can collaborate with other AGI interfaces to create a system with a more comprehensive understanding of related visions. For instance, LLaVA-Plus (Shilong Liu, 2023) builds upon LLaVA, an end-to-end trained vision language model, by incorporating newly constructed data to enhance its tool-using skills beyond its original visual understanding and reasoning abilities.

Agent-based works also equip the ability to solve multi-modal problems using various external tools for such purposes. Numerous conventional algorithms facilitate the coordination of multiple agents or robots in either physical or simulated settings (Sunehag et al., 2017; Gupta et al., 2017; Fioretto et al., 2018; Foerster et al., 2018). Furthermore, advancements have been made in developing techniques that accelerate communication among multi-agent, thus enabling them to work towards a common goal across a variety of tasks (Sukhbaatar et al., 2016; Jha et al., 2024; Qian et al., 2023a; Hong et al., 2023b; Liu et al., 2023k). Works by Li et al. (2023e) and Chen et al. (2023f) have established a range of common scenarios for multi-agent interactions, featuring vivid visualizations and the integration of human interaction. Building upon the existing frameworks for multi-agent systems. Chen et al. (2023c) innovatively propose the automated creation of new agents to navigate dynamic environments effectively. Additionally, Hong et al. (2023b) and Zhou et al. (2023a) have integrated Standardized Operating Procedures (SOPs) to streamline the customization and deployment processes of multi-agent systems. The underlying principle of these systems is utilizing advanced LLMs as coordinators, aimed at efficiently addressing a myriad of tasks within simulated environments. Recent developments in the concept of Natural Language-Based Societies of Mind (NLSOMs) (Zhuge et al., 2023) have revolutionized the understanding of cooperation between neural networks to the "Mindstorm" metaphor. This framework employs the language interface to conduct communications between agents, allowing for easy and straightforward adaptation of novel modules.

Looking beyond multi-modal tasks, more specific and advanced agent systems that can handle more complex computer applications such as web browsing (Mialon et al., 2023; Deng et al., 2024; Zhou et al., 2024b), software manipulation (Kapoor et al., 2024; Rawles et al., 2023; Yang et al., 2023b), and gaming (Ma et al., 2023b; Wang et al., 2023h;b; Xu et al., 2024) have emerged. In the gaming realm, agents have been deployed in simulated interactive environments, including Minecraft (Wang et al., 2023h;b), Starcraft II (Ma et al., 2023b). These agents receive textual observations from internal APIs and execute predefined semantic actions. However, these domain-specific applications limit the agents' ability to generalize to other games or broader software applications. Tan et al. (2024) proposes the concept of General Computer Control (GCC) and adopts a more intuitive approach, employing multimodal input from screenshots to generate keyboard and mouse commands within Red Dead Redemption II. This setting

holds promise for expansion into more intricate computer tasks. While Wang et al. (2023a) attempts to interact with the environment in a human-like manner using screenshots as input and controlling mouse and keyboard actions, its action space is constrained to a predefined hybrid space. Despite achieving notable results in specific tasks, these methods lack the ability to generalize across diverse tasks due to inconsistencies in observation and action spaces. Several research efforts (Zhang et al., 2024b; Gao et al., 2023b; Kapoor et al., 2024; Niu et al., 2024; Wu et al., 2024b) have aimed to enhance the scalability of web agents by utilizing screenshots as input and keyboard and mouse operations as output, allowing them to interact with a wider range of applications.

**AGI-level Interfaces to Other Agents**  While current interfaces to AI models demonstrate effectiveness, they are primarily focused on a narrow range of applications (Askell et al., 2021; Memarian and Doleck, 2023). To enhance their capabilities, we propose the following improvements in the future:

- **Advanced interface encodes unified representations.** The integration of the AGI interface is paramount for enhancing model performance. These interfaces can supervise less capable models in sequential interactions and introduce a range of novel capabilities in parallel cooperation. A key focus should be developing comprehensive and lightweight AGI interfaces that exhibit strong generative and understanding abilities. Future advanced interfaces should be able to encode a unified representation across all modalities. This would simplify the aligning process and optimize resource allocation for various downstream tasks, such as generation and identification.

- **High-quality connection promotes effective communications.** To ensure that AGI-level interfaces among agents lead to effective collaboration, it is essential to establish high-quality connections that are both reliable and efficient. The recent concept of weak-to-strong alignment is particularly noteworthy, underscoring the significance of determining the most effective methods for incorporating external capabilities into existing systems. Traditionally, tuning model parameters has been the primary method for generalizing systems to specific tasks or domains. However, recent research highlights the potential of approaches that require minimal or no parameter tuning (Burns et al., 2023; Zhao et al., 2023c). Moreover, incorporating in-context learning and context-aware communication among agents can enable agents to adjust their interactions based on the situational context, improving the relevance and efficacy of their collaboration.

- **Interaction protocols ensure safe interaction.** It's important to create robust and effective interaction protocols to serve as the foundation for safe communications and actions between different AGI entities. To achieve this, it's important to implement standardized security measures, including advanced encryption methods, authentication protocols, and content filters specifically designed to safeguard against the dissemination of misinformation or malicious content. Additionally, developing guidelines for safe AGI actions and ensuring that all activities are performed within the bounds of ethical norms and regulations is essential. The focus on safety protocols enhances the security of interactions between AGI systems and builds trust with end-users, paving the way for broader acceptance and integration of these technologies into everyday applications.

- **Advanced agent network promotes cooperative learning.** Enhancing the network of AGI agents to foster social and cooperative learning is essential for the advancement of collective intelligence. By enabling agents to share insights, strategies, and knowledge, we can facilitate a more rapid and efficient learning process across different domains. Social learning mechanisms, such as imitation and observation, can be incorporated to enable agents to learn from the successes and failures of their peers. Furthermore, cooperative learning models can be designed to encourage agents to work together towards common goals, harnessing their diverse strengths and capabilities. Such a networked approach not only accelerates the pace of innovation but also leads to the developing more versatile and adaptive AGI systems capable of tackling complex, real-world problems through teamwork and collaboration.

### 3.3.2   AI Interfaces to Humans

Human intelligence has been the ultimate goal of AI, and human beings have also been the primary beneficiaries of AI. As we move towards AGI, we should empower AI with the capabilities to interact with humans to

ensure that AI can actually benefit humans. Therefore, we call for future advances in interface technologies to lay solid foundations for AGI's capabilities to interact with humans.

**Current State of AI Interfaces to Humans** Developing interfaces with artificial intelligence has been explored in Human-Computer Interaction (HCI) research for a long time. We discuss current research in human-AI interfaces, including both graphical interfaces and multimodal interfaces, as well as general principles.

In history, there have been many related principles or guidelines for designing interfaces for human-AI interaction. Most of the existing frameworks in HCI research to interact with human are based on the idea of *augmenting* (rather than *replacing*) human intelligence with artificial intelligence (Engelbart, 1962). Therefore, maintaining human agency and reflecting human values have been consistent themes in designing human-AI interaction. Researchers also argued that the benefits of allowing AI agents to take the initiative and automate users' routines versus the benefits of waiting for users' direct manipulation would need to be carefully weighed (Horvitz, 1999; Shneiderman and Maes, 1997). With advances in artificial intelligence, researchers have articulated 18 generally applicable design guidelines for human-AI interaction spanning different phases in user interactions (Amershi et al., 2019). Recent research also presents six principles for designing generative AI applications that address unique characteristics of generative AI UX and offer new interpretations and extensions of known issues in designing AI applications (Weisz et al., 2024). Such guidelines could serve as a resource for the principles of the future design of AI-infused interfaces, optimizing interaction performance and improving the interaction experience.

- **Graphical user interfaces.** One emerging line of research has focused on designing interfaces to support user tasks based on textual or visual interactions, which will lower the "threshold" while raising the "ceiling" in terms of the quality and the diversity of user tasks (Myers et al., 2000). A common theme in developing such interfaces is to create potential wrappers beyond simply providing "straightforward" input and output (Jiang et al., 2023c; Suh et al., 2023; Gero et al., 2024; Suh et al., 2024). For instance, researchers tried to use interactive diagrams to support humans in dealing with information-seeking and question-answering tasks powered by large language models (Jiang et al., 2023c). Another thread of research is to identify possible workflows or strategies that can unlock the potential of AI during human-AI interaction (Wu et al., 2022b;a; Brade et al., 2023; Arawjo et al., 2023; Leiser et al., 2024; Kim et al., 2024; Feng et al., 2024). For example, previous researchers introduced the notion of chaining multiple LLM prompts together to help users accomplish complex tasks with LLMs, which enables humans to take advantage of LLM's ability to handle a variety of independent tasks (Wu et al., 2022b;a). In addition, when interacting with humans, large language models could encounter various non-language input or output data, such as direct manipulation action traces, vector graphics, or application states (Aveni et al., 2023; Duan et al., 2024). For example, researchers attempted to create alternative representations of context information to leverage the capabilities of large language models in different interaction tasks, auto-completion of forms (Aveni et al., 2023).

- **Multimodal user interfaces.** Many researchers are actively exploring integrating AI with existing interaction techniques to enrich user experiences across different modalities and for different groups. On the one hand, previous research has created many novel sensing technologies and interaction techniques beyond simple textual and visual interactions. Recent advances in multimodal foundation models have shown great promise in many interaction tasks. For example, GPT-4o possesses remarkable capabilities of reasoning across audio, vision, and text in real time OpenAI (2024). In the future, it is worth exploring the possibility of empowering human-level AI with the capabilities to interact with humans through different modalities (Li et al., 2024d; Lin et al., 2024d). From this perspective, previous researchers have proposed a novel pipeline that provides generalized predictions of follow-up actions for real-world multimodal sensory inputs, leveraging the explicit reasoning of LLMs on structured text converted from multimodal data to ground the predicted actions (Li et al., 2024d). Meanwhile, AI could also be an important part of user experiences in mixed reality. Recent research in mixed reality provides abundant opportunities for user interfaces that could be driven by large language models (Bozkir et al., 2024). Additionally, researchers are actively exploring inclusive interfaces to ensure everyone can benefit from interacting with AI. One area of focus is creating better interfaces for people with disabilities in the field

of accessibility research (Huh et al., 2023; Valencia et al., 2023). In recent work, researchers have created interactive systems that allow blind or low-vision creators to generate images by providing rich visual descriptions of the generated outcomes in language (Huh et al., 2023).

**AGI-level Interfaces to Humans**   Researchers pointed out unique challenges in designing human-AI interactions due to the uncertainty surrounding AI's capabilities and the complexity of AI's outputs (Yang et al., 2020). In the future, there will still be important challenges that we will need to overcome to truly empower AGI with the capabilities of interacting with humans (Bigham, 2023).

- **Ensuring the benefits in different environments.** Future research needs to figure out strategies for AGI to benefit humans in what they want to do. AGI will have more capabilities comparable to intelligent humans, bringing forth numerous possibilities that can benefit real humans. However, if the cost of using AGI exceeds its benefits, people are unlikely to derive value from it (Horvitz, 1999). Looking ahead, there are many opportunities to design AGI-level interfaces in different modalities and various environments, such as wearable computing, smart environments, and mixed-reality settings. However, researchers still need to explore potential solutions to ensure that AGI could actually yield beneficial results to humans that outweigh the cost.

- **Maintaining the controllability for different people.** Future work should explore how we can support diverse people, even with limited AI literacy, to interact with AGI in a controllable manner. Compared with AI experts, it could be difficult for users who lack an AI background to understand AI's full capacities and mechanisms, which can lead to unexpected pitfalls in interactions. The advances in AGI would continue to amplify similar concerns. In the future, it is increasingly important for us to think about potential solutions to empower users with the capabilities to understand and control the interactions with AGI.

- **Managing the risks at different scales.** As we architect novel interfaces for AGI, future researchers should consider the potential implications these interactions will have at individual, community, and societal levels. Previous research has continued to identify the impacts of human-AI interaction on people's behavior and mentality. Though these issues are not necessarily unique to AGI, they will be amplified and more prevalent as we empower AI with more human-level capabilities in the real world. Future work still needs to focus on approaches to understand and mitigate the impacts of interface design for AGI at different scales so that we can maximize their positive impacts and minimize their negative impacts.

## 4   AGI Systems: Implementing the Mechanism of AGI

*Dangers lurk in all systems. Systems incorporate the unexamined beliefs of their creators.*

*— Frank Herbert, God Emperor of Dune*

The emergent behaviors (Wei et al., 2022a) exhibited by many large models such as Llama 2 (Touvron et al., 2023), GPT-4 (OpenAI, 2023a), Gemini (Team et al., 2023), Claude 3 (AI, 2024), and Mistral Jiang et al. (2023d) appear when the number of parameters in a model gets scaled up to a certain amount. The underlying workhorse that enables this scaling while retaining sufficient efficiency of LLMs is a range of system efforts: 1) *scalable model architectures* fundamentally and algorithmically define the computation and modeling, 2) *large-scale training* techniques optimize the utilization of more hardware accelerators, potentially spread out geographically, 3) *inference infrastructures* ensures stable and high-throughput serving of multiple models, 4) *cost and efficiency* discusses various methodologies in making data, model combination, and automation process much more efficient, and finally we touch some aspects on 5) *hardware computing platforms* which attempt to break soft physical constraints and therefore, provide the next generation computational capabilities and hardware foundation for future algorithmic innovations.

Advancements in system research are essential for facilitating this scalability, a trend that is anticipated to remain pivotal as we step towards artificial general intelligence. With continual improvement in AI system

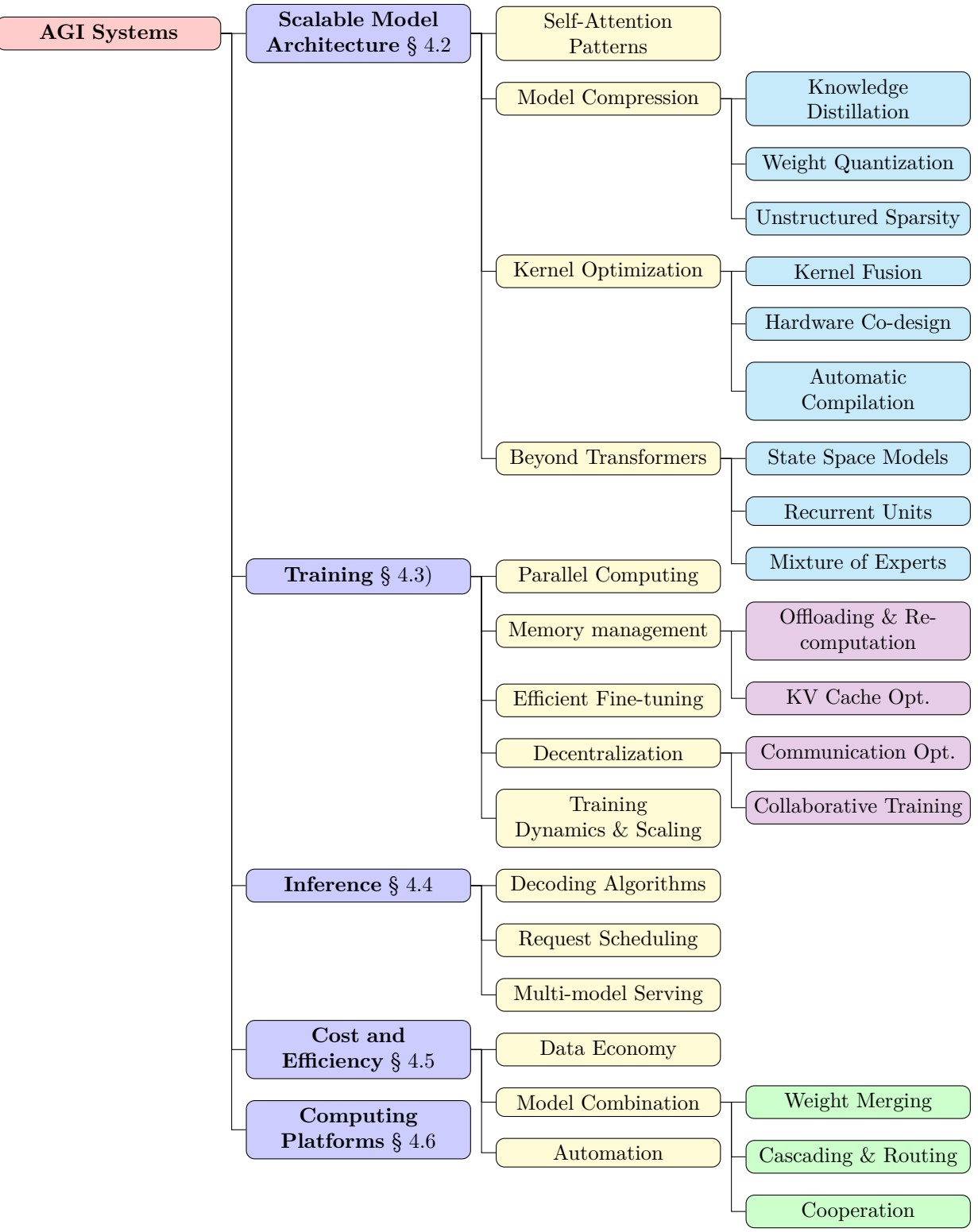

Figure 5: **Taxonomy of Current AGI Systems.** We discuss several advancements in various categories of AGI systems, including scalable model architectures, large-scale training, optimized inference techniques, methods for reducing cost and improving efficiency, as well as next-generation computing platforms for AI.

research and engineering, we could envision that models are trained over ten thousand heterogeneous accelerators across the cloud, which not only connects multiple agents into a single multi-functional amalgamation but also provides instant and personal assistance to people in everyday life.

In this section, we start with the major challenges that AGI systems need to tackle and introduce some prior efforts on model architecture innovation, training, inference, cost reduction, and computing platforms. Finally, we will conclude by envisioning what AGI systems would look like and their roles in the future.

### 4.1 System Challenges

We first briefly describe and categorize major system challenges in this section:

**The Large Amount of Training Data** Large models require a lot of training data to achieve Chinchilla optimality (Hoffmann et al., 2022). At the same time, we can envision that the raw amount of data available on the internet will likely skyrocket while the average quality and authenticity might not improve as fast, partly due to the massive success of generative models and user content creation. This demands a more sophisticated and automated data processing pipeline that can select, structurize, clean, and mix data from different sources for efficient training.

**The Speed and Cost of Iteration** Each iteration of large model training can take enormous resources and time during prototyping and experiments. In practice, there might be many other interference, such as human and system errors, that will result in training preemption. Automatic (hyper-parameter & architecture) search pipeline and well-designed training infrastructure (Shoeybi et al., 2020; Aminabadi et al., 2022) can drastically reduce iteration cost and implicitly improve model development speed.

**Privacy-sensitive and Resource-constraint Settings** While the current most successful large models are deployed in data centers where requests from users are processed in a centralized manner (Achiam et al., 2023; AI, 2024), the need for serving models on edges where data and queries are used and processed locally in more privacy or latency-sensitive situations will become more substantial. However, edge devices are usually less capable in terms of computing and memory, which motivates developing techniques that optimize the utilization under resource-constraint settings and, at the same time, do not compromise model performance.

**Efficient Methods for Fine-tuning and Adaptation** Fine-tuning pre-trained models on task-specific data has been the most popular paradigm. Despite this, the computing requirement and time for full model weight updating is still prohibitive for many users. Efficient fine-tuning methods help reduce the barrier to domain adaptation, agent training, and task-specific optimization.

**Serving Latency and Throughput** AGI systems need to support low latency and high throughput for seamless user experiences and engagement. However, current systems often trade-off one with others such as optimizing batch processing, the time to first token, or single query completion time (Yu et al., 2022; Aminabadi et al., 2022). Striking a balance among all these metrics is a challenging question.

**Memory Footprint** One salient challenge for deploying large models is the memory footprint, which becomes even more severe for long context and multi-modal inputs due to the quadratic nature of self-attention. KV cache is the common technique for trading off memory for faster inference (Pope et al., 2022) and will also incur a significant memory burden if not handled gracefully.

**Hardware Compatibility and Acceleration** The performance of model serving heavily depends on how well engineers can leverage the hardware's capability. Specialized kernels and algorithms designed for different accelerator architectures can substantially boost the inference speed. Being compatible with heterogeneous devices and creating uniform software abstraction can help fully unleash the potential of large models.

In the following sections, we'll discuss advancements in the major system categories and relate how these prior efforts help address some of the above-mentioned challenges.

### 4.2 Scalable Model Architectures

One everlasting topic that system researchers and engineers are trying to deal with is how to make larger and more powerful models. However, there are several axes to consider for scaling: *the number of parameters and*

*training data volume* (Chinchilla scaling law (Hoffmann et al., 2022)), as well as *the effective context length and serving capacity* (Long context scaling law (Xiong et al., 2023)). To this end, we start with introducing optimization techniques from prior works on the model architecture level and move on to training and inference infrastructures that are more model-agnostic.

**Self-attention Patterns**  Vanilla self-attention, the core mechanism under transformer architectures (Vaswani et al., 2017), scales quadratically with the input sequence length, drastically limiting its potential on long context tasks with current hardware memory capacity. However, the full causal mask used in self-attention might not be efficiently optimal (Farina et al., 2024), wasting computation and time. The lottery ticket hypothesis Frankle and Carbin (2019) and the structured sparsity observation (Dong et al., 2023a) both suggest potential structural and computational redundancy in LLMs. Inspired by these works, many works explored different attention mask patterns to reduce the computations and memory requirement over the fully casual one. Sliding window Beltagy et al. (2020); Jiang et al. (2023d) and dilated patterns Ding et al. (2023b) either limit to local attention or reduce the resolution, often resulting in improving efficiency with some performance degradation. Another line of work observes that some tokens in the sentence are semantically more important and hence introduce different kinds of global tokens for efficient attention, such as the initial tokens (Xiao et al., 2023) and the special landmark (Mohtashami and Jaggi, 2023) for each block of tokens. It is worth noting that the computation bottleneck might switch between the self-attention and fully connected layers for models with different scales, and blindly applying heuristics-based sparse patterns might not only give marginal speedup but also incur a loss in performance, which motivates future research in more complex and adaptive efficient patterns.

**Model Compression**  The goal of model compression is to reduce the memory footprints, computational complexity, and deployment cost of large models. Among many approaches, Knowledge Distillation (KD) (Hinton et al., 2015; Hsieh et al., 2023) is the process of extracting the learned knowledge of a bigger teacher model to a smaller student model that can often be served more efficiently. On the one hand, black-box LLM knowledge distillation only requires querying the teacher model (via API calls) and collecting input and output pairs which can be used to train the student model. Following this line, Alpaca (Taori et al., 2023) only costs around 100 US dollars to balance efficiency and performance by distilling from ChatGPT. Vicuna (Peng et al., 2023b) shows promising performance on instruction fine-tuning with knowledge distillation from GPT-4. On the other hand, MiniLLM (Gu et al., 2023) explores white-box knowledge distillation where more information like model weights and loss values of the teacher model is accessible, which can potentially stimulate better knowledge transfer. MiniLLM proposes to replace the standard KL divergence objective with reverse KL which prevents the student model from overestimating the low-probability regions of the teacher distribution. Generalized Knowledge Distillation (GKD) (Agarwal et al., 2024) breaks the traditional supervised KD training region by taking advantage of student's self-generated output sequences and leveraging feedback from the teacher on such sequences. This not only helps mitigate the distribution mismatch between training and inference but also has been proven to be more useful when the student model lacks enough capacity to fully mimic the teacher's behaviors. Balancing the level of access to the teacher model will be remained as a relevant topic for algorithmic design, safety, data privacy, cost, and efficiency. It is also exciting to explore the possibility of reverse-learning or super-alignment [2] where we want to distill knowledge from weaker models that can be leveraged (e.g., through analyzing, merging, and adaptively updating) to improve the current one.

Another line of work for explicit model compression is to take advantage of the model sparsity and prune parts of the model weights during inference. Similar to different attention pattern designs, this direction is mostly motivated by several empirical observations like lottery tickets and contextual sparsity hypotheses. However, it is non-trivial to apply pruning to LLMs without careful consideration of many aspects, such as the need for extra steps of fine-tuning, system overhead due to dynamic architectures, and the trade-offs among implementation difficulty, discernible efficiency gain, and the potential compromised performance. ZipLM (Kurtic et al., 2023) structurally prunes the model by iteratively identifying and removing components with the worst loss-runtime trade-off given a dataset, an inference environment, and speedup objectives. LayerDrop Fan et al. (2019) introduces the structured dropout which allows efficient pruning via selecting

---

[2] https://openai.com/blog/introducing-superalignment

sub-networks of any depth from the network without extra fine-tuning. Deja Vu (Liu et al., 2023g) predicts the contextual sparsity of a given input which guides the selection of only specific attention heads and MLP parameters during inference. FlashLLM (Xia et al., 2023b) solves the unstructured pruning with memory-efficient SpMM implementation on tensor cores.

**Kernel Optimization**  Kernels are developed to speed up primitive computations such as general matrix multiplication for large networks. Kernel fusion is the technique of merging two or more kernels into one, which can reduce the overheads of kernel launching and redundant memory access. This has been widely implemented in many inference engines like LightSeq (Wang et al., 2021), FasterTransformer (NVIDIA, 2023a), and ByteTransformer (Zhai et al., 2023) where the instantiations mostly consist of grouped GEMMs, layer normalization, activation calculation, and self-attention Dao et al. (2022); Dao (2023). Most works following this direction leverage the GPU memory hierarchy and try to hide memory latency and maximize thread occupancy.

With the increasing demand for highly optimized kernel implementations for various computation patterns, automatic compilations have become a very active area of research, an approach that provides greater flexibility than sourced libraries for highly optimized kernel implementations (e.g., CUTLASS, cuBLAS, and cuDNN). Within this category, TVM (Chen et al., 2018), Triton (Tillet et al., 2019), and JAX (Bradbury et al., 2018) harness the potential of hardware accelerators by compiling into highly efficient low-level code, and at the same time, provide python interface for fast and easy prototyping. This not only greatly lowers the learning curve for writing custom kernels but also provides an abstraction for code adaptation to other computing platforms and backends with heterogeneous devices.

**Beyond Transformers**  Despite the enormous success of the ubiquitous transformer architecture, many works attempted to find other designs to overcome some of its shortcomings. Mixture of Experts (MoEs) (Shazeer et al., 2017; Roy et al., 2020) replace the dense layers in transformer models with a conditional module consisting of multiple "expert" sub-networks. A routing mechanism is used to dynamically decide which expert(s) to use on the token-level (Zhou et al., 2022a; Fedus et al., 2022) or task-level (Kudugunta et al., 2021). Despite having multiple experts, sparse MoEs can often train and decode faster with the same model size and are expected to specialize in different abstract tasks (Jiang et al., 2023d; Hwang et al., 2023; Gale et al., 2022). However, MoEs also raise other system challenges during inference such as higher requirements for loading all experts into the VRAM and distributing experts over multiple nodes.

State space models (SSMs) have recently been applied to model sequence-to-sequence transformations (Gu et al., 2022a; Gupta et al., 2022; Smith et al., 2023), which can be readily used in various model architecture topology to replace the quadratic self-attention mechanism. A (discretized) SSM defines a recurrence relationship along each time-step (token) via a tuple of learnable parameters $(\Delta, \bar{A}, \bar{B}, C)$ and the major challenge that most works try to solve is how to compute this recurrence in a parallelizable way that can efficiently use modern hardware accelerators (e.g. FFTConv). The simplest form in this category is linear attention (Katharopoulos et al., 2020; Yang et al., 2023e), which can be viewed as a degenerate SSM. At its core, linear attention expresses self-attention as a linear dot-product of kernel feature maps and makes use of the associativity property of matrix products to reduce the complexity down to linear. S4 (Gu et al., 2022a) parameterizes the SSM both expressively and efficiently via a low-rank correction, allowing it to be diagonalized stably and reducing the SSM to the well-studied computation of a Cauchy kernel. There are many other following works (Gupta et al., 2022; Gu et al., 2022b; Smith et al., 2023) after S4 which attempt different parameterization of the transition matrix $\bar{A}$ (and others) to improve both the computational efficiency and modeling capacities. H3 (Fu et al., 2023a) proposes an SSM block that consists of two stacked separate SSMs that are specially designed to meet the challenge of recalling earlier tokens and support token comparisons across sequences. Hyena (Poli et al., 2023) generalizes H3 by replacing the S4 layer with an interleaved and implicitly parameterized long convolutions and data-controlled gating, which disentangles parameter size from the filter size and hence allows for greater expressivity. (Sun et al., 2023a) proposes Retentive Network a foundation architecture that includes additional gates and uses a variant of multi-head attention, achieving impressive constant inference cost and linear long-sequence memory consumption. RWKV (Peng et al., 2023a; 2024a) is a new architecture that takes advantage of the efficient parallelizable training of Transformers with the efficient inference of RNNs. In essence, the main "WKV" operation involves linear

time invariance (LTI) recurrences, which can be interpreted as a ratio of two SSMs (Gu and Dao, 2023). To tackle the key weakness of previous SSM models which is their inability to perform content-based reasoning, Mamba (Gu and Dao, 2023) proposes the selective state space that can make the SSM parameters as functions of the input, effectively turning the SSM from LTI to time-varying. Despite no longer being able to apply efficient convolutions, they designed hardware-aware parallel algorithms for recurrence computation called Parallel Associative Scan, which enables it to achieve over $5\times$ higher throughput than Transformers, SoTA performance across several modalities, and keep improving on real data up to million-length sequences.

Sparkles of interest for revisiting recurrent neural networks (RNNs) have also emerged with its major advantage in long context processing (i.e. linear time and constant memory for the hidden state). One of the challenges for RNNs is to scale the training and inference efficiently. (De et al., 2024) introduces Hawk, an RNN-based model with gated linear recurrences, and Griffin which mixes gated linear recurrences with local attention. They showcase the superior performance of Hawk against Mamba on downstream tasks, and Griffin matches performance of Llama-2 with six times fewer training tokens. Not only do they demonstrate the potential of long context capability, but also explain how to effectively leverage hardware accelerators during distributed training and inference by scaling Griffin to 14B parameters. Right after that, a family of models called RecurrentGemma (Botev et al., 2024) came out with various model sizes, in both pretrained and instruct-tuned versions. These advancements present the possibility of training a data-efficient, fixed state size, long context, and expressive model without relying solely on transformer architectures.

Recent works also explored high-level architecture hybridization strategies that wish to bring the benefits from different variants. (Lieber et al., 2024) proposes to combine Transformers with Mamba by interleaving layers, which achieves impressive results on both standard and long context tasks with manageable resource requirements. Beyond manual design, MAD (Poli et al., 2024) integrates the process into an end-to-end pipeline consisting of small-scale capability unit tests predictive of scaling laws. MAD successfully finds an efficient architecture, called Striped Hyena, based on hybridization and sparsity, which outperforms state-of-the-art Transformer, convolutional, and recurrent architectures (Transformer++, Hyena, Mamba) in scaling, both at compute-optimal budgets and in over-trained regimes. These works will likely continue to inspire further explorations in architectural designs that are both performant and efficient at scaling, breaking through the current Transformer paradigm.

### 4.3 Large-scale Training

Scaling the training of large models encounters many challenges with modern hardware, such as the fact that models can no longer fit into a single GPU due to the increasing memory requirement, accelerating the training speed with more computing units while incurring minimal overheads (linear scaling), and leveraging disaggregated resources, etc. In this section, we give an overview of several works that enabled large-scale pre-training and efficient fine-tuning for downstream task adaptation, with a gentle introduction to motivate many possibilities with decentralized training.

**Parallel Computing** Parallelism for large language models in a clustered environment with multiple computing units can often be characterized into four major modes, often known as "4D parallelism". *Distributed data parallel* (DDP) is the simplest setup where the model is replicated across units, and the data is sliced and fed to each model, typically (implementation-specific) with a synchronization step at the end of each pass. More sophisticated versions of DDP like ZeRO and FSDP are used ubiquitously in modern large training frameworks such as DeepSpeed (Aminabadi et al., 2022), FairScale (FairScale authors, 2021), and Megatron-LM (Shoeybi et al., 2020). *Tensor parallel* (TP) or *model parallelism* splits the model weights into multiple chunks which are distributed across GPUs. This horizontal splitting allows data to be processed in parallel across sharded weights and then the results are aggregated at the end of each step, which often involves clear fusion (Shoeybi et al., 2020) to reduce the synchronization communication. *Pipeline parallel* (PP) (Huang et al., 2019), on the other, divides the model layers vertically onto different GPUs and the data will move from stage to stage over different units. *Sequence parallel* (SP) (Liu et al., 2023j; Shoeybi et al., 2020) targets mostly for long context tasks and split along the sequence dimension to mitigate the computational and storage loads. Combining different parallelism will likely result in highly efficient systems. However, it is not trivial to do so given their distinctive trade-offs and cluster configuration. Alpha (Zheng

et al., 2022), HexGen (Jiang et al., 2023e), and FlexFlow (Jia et al., 2018) attempted to automate the process of parallelizing model training and inference with the goal of maximizing the hardware utilization. Cluster configuration (memory, bandwidth, and latency of individual accelerators, network bandwidth, etc) is often estimated, and a search-based algorithm such as dynamic programming and constrained optimization is employed to find the best possible parallel strategy. Asymmetric computation is also supported by adaptively assigning requests for a latency requirement (Jiang et al., 2023e). These automatic parallelism scheduling methods have been tested to perform on par or even better than manual designs in many cases involving hardware and network heterogeneity.

**Memory Management** Memory management is one of the most crucial aspects for training and serving large models, especially in the long context domain where the memory footprint of the KV cache can easily surpass that of the model weights and activation combined. Inspired by traditional OS design, Paged attention from vLLM (Kwon et al., 2023a) solves the memory fragmentation problem by partitioning the KV cache into non-contiguous blocks of memory, which significantly improves memory utilization and hence increases the system throughput and efficiency. FastGen (Ge et al., 2024) introduces an adaptive KV cache compression technique, guided by its structure profiling, that dynamically evicts non-special tokens and decreases memory usage. Scissorhands (Liu et al., 2023a) and $H_2O$ (Zhang et al., 2023j) also share similar flavor with their empirical observation that keeping only pivotal tokens can retain most of the performance while requiring minimal fine-tuning and saving memory usage. Infinite-LLM (Lin et al., 2024a) first splits the attention calculation into smaller subroutines that can be assigned to different units. To make efficient distribution of these subroutines possible, A designated server is developed that can dynamically manage the KV cache and effectively orchestrate all accessible GPU and CPU memories spanning across the data center.

Many important techniques have been widely adopted in popular DL frameworks to fit larger models into devices with fixed memory. CPU offloading allows models to selectively transfer weights (layers) or KV cache to CPU with more memory, and only load essential network parts to GPU for processing. When pushed to the extreme, FlexGen (Sheng et al., 2023b) can achieve significant batch throughput of OPT-175B on a single 16GB GPU. Gradient checkpointing (Chen et al., 2016) reduces peak memory usage by recomputing parts of the computational graph during back-propagation. There is no doubt that efficient memory management will be remained as the core investment direction that enables the deployment of scalable systems and parallel processing of larger batches.

**Efficient Fine-tuning** Pretrained large models often internalize a tremendous amount of knowledge which can be unleashed by (instruct) fine-tuning. However, despite the fact that often a relatively small amount of examples are sufficient for successful fine-tuning, the cost and time for doing so are prohibitive and not economical. The main objective for efficient fine-tuning is to figure out a balance between the cost (implementation difficulty, data requirement, training budget, etc) and the performance gap from continual pretraining. A series of parameter-efficient fine-tuning (PEFT) techniques have been developed to meet this challenge, which only requires training a small number of new parameters and often achieves better performance than in-context learning. LoRA (Sheng et al., 2023a) as one of the most popular PEFT methods, draws great attention these days. LoRA and many of its variants (LoHA (Hyeon-Woo et al., 2023), AdaLoRA (Zhang et al., 2023a), Q-LoRA (Dettmers et al., 2023), and recent PiSSA Meng et al. (2024)) insert learnable matrices that are low-rank decompositions of the delta weight matrices. LLaMA-Adapter (Zhang et al., 2023d) efficiently fine-tunes LLaMA into an instruction-following model with very little computational budget. A set of learnable adaptation prompts are first prepended to the context and they train a zero-initialized attention mechanism with zero gating with only 52K self-instruct demonstrations. The resulting extra 1.2M parameters from the adapter can give high-quality outputs, comparable to fully fine-tuned results. Sharing a similar flavor to LoRA, $IA^3$ (Liu et al., 2022b) scales the model activations by learned vectors instead of matrices. Other PEFT methods that insert learnable components showcase strong generalization ability, and prompt-based methods like soft prompting (Lester et al., 2021) add extra learnable parameters to the input embeddings while keeping the original model weights frozen. Adapters (Houlsby et al., 2019) add trainable parameters inside the attention blocks, while Prefix tuning (Li and Liang, 2021) appends learnable vectors to the KV representations in attention. Unlike the traditional PEFT techniques, Zhao et al. (2023c); Basu et al. (2023) first discovered tuning the Layernorm layer of transformers yields decent

performance in these models as an unexpectedly strong baseline. Other than twitching model parameters in the PEFT process, GaLore (Zhao et al., 2024b) proposes to apply the LoRA tuning paradigm on the model gradients, and REFT (Wu et al., 2024a) chooses to place a linear probing strategy between the source model parameters and the optimization goal.

**Decentralization**  Many works focus on utilizing dis-aggregated and hardware heterogeneous computing devices over the cloud for model training and inference. One challenge of geographically separated clusters is the communication overhead, which makes data movement costly (training data, gradients, KV cache, etc.) and eclipses decentralization's benefits. CacheGen (Liu et al., 2023c) compresses the KV cache with an encoder into compact bitstream representations, reducing the latency for context fetching and processing. CocktailSGD (Wang et al., 2023g) employs a combination of sparsification and quantization techniques, which makes fine-tuning LLMs up to 20B size with slow networks possible with only minimal slowdown compared to data center's fast interconnect. DiLoCo (Douillard et al., 2023) introduces a novel federated averaging algorithm run on islands of devices that are poorly connected, which claims to perform as well as fully synchronous optimization on C4 datasets while communicating 500 times less. Collaborative training crowd-sources commodity GPUs from individual users, the most prominent example of which is Petals (Borzunov et al., 2022), a system capable of serving and fine-tuning BLOOM-176B and OPT-175B with decent performance (e.g. supports interactive sessions) using only mediocre GPUs from multiple parties. Decentralized AI systems open up the possibility of bridging devices across the globe, which ensures fault-tolerance (Ryabinin et al., 2023) and compatibility of heterogeneous devices plus networks (Jiang et al., 2023e; Yuan et al., 2023), as well as optimizing limited network bandwidth (Wang et al., 2023i) and data privacy (Tang et al., 2023).

**Training Dynamics & Scaling**  The science of large language models is mysteriously difficult to grasp, the understanding of which can drastically improve the development of various AIs. However, most successful LLMs are not fully "open" not just in terms of data and model weights but other aspects such as the intermediate checkpoints and artifact logging that can assist in reasoning about the training dynamics as we scale models to different sizes. (Xia et al., 2023a) analyzes the intermediate checkpoints of OPT models (Zhang et al., 2022a) on various downstream tasks, which attempts to emphasize the perplexity as a predictive indicator of a model's performance than its size, showing that larger models hallucinate less often and that models tend to exhibit minimal return during early stage of the training. Complimentary to this, (Tirumala et al., 2022) focuses on studying different memorization capabilities across the model size, dataset size, and learning rate and proposes an interesting hypothesis on the importance of nouns and numbers as the unique identifier for memorizing individual training examples. Besides pure analysis, Pythia (Biderman et al., 2023) introduces a suite of 16 LLMs trained on public data, ranging in size from 70M to 12B parameters. With these intermediate checkpoints released to the broader community, it becomes way easier and more efficient for researchers to find answers to questions related to training dynamics by examining and benchmarking individual saved weights and losses. Finally, OLMo (Groeneveld et al., 2024), on top of that, graciously releases the whole framework, including training data and training and evaluation code, for the benefit of making the study of the science behind LLMs easier.

## 4.4  Inference Techniques

AGI inference systems need to ensure user responsiveness, availability, and efficiency, which helps unleash the ultimate potential of large models from the training phase and revolutionize how users interact with the system. Hence, in this section, we give an overview of several techniques that try to accelerate auto-regressive decoding, balance request scheduling, and serve a massive number of models with different capabilities in the cluster, which will inspire future system efforts across the spectrum.

**Decoding Algorithm**  In this paper, we focus mostly on exact decoding acceleration where we want to maximize the performance while staying faithful to the original model without compromising the accuracy. (Miao et al., 2023) gives a comprehensive review of several approximate methods such as sampling strategies, non-autoregressive decoding, semi-autoregressive decoding, block-parallel decoding, and early existing, etc. A large body of works explores the idea of speculative decoding (Leviathan et al., 2023) with the central idea of trading parallel computation for higher chances of generating multiple tokens at once. Usually, a speculative decoding process starts with an efficient draft model that makes predictions of multiple steps,

the resulting proposals verified by the target model we want to sample. However, there are many challenges involved, including 1) how to make the draft model lightweight while still generating useful guesses for efficient progress, 2) how to avoid extensive architecture change and fine-tuning for faster adaptation, and 3) how to deploy the draft model more effectively. The simplest yet effective variant is called *Prompt-lookup Decoding* (Saxena, 2023) where the draft model is replaced by simple prefix string matching from an existing database for generating candidate tokens. This model-agnostic approach can decode extremely fast without any fine-tuning or model change, but the performance heavily depends on the quality and diversity of the string pool. To facilitate faster verification over a large number of candidates, SpecInfer (Miao et al., 2024) organizes the outputs of the draft models into a token tree, with each node being a candidate token, the correctness of which can be efficiently checked in parallel by the base model. Following a similar idea, Medusa (Cai et al., 2024a) introduces a tree attention mechanism to simultaneously check all tokens from the medusa heads, which is realized by a special mask pattern for efficient parallel computation. Self-speculative decoding (Zhang et al., 2023k) proposes to completely discard the requirement of a draft model and generate candidate sequences by selectively skipping a subset of intermediate layers.

Hardware-aware algorithms are particularly effective and appealing for the decoding phrase. Following the efficient self-attention works (Dao et al., 2022; Dao, 2023), Flash-Decoding[3] splits along the sequence dimension and process these blocks with Flash-Attention in parallel with their KV cache and statistics, the results of which will be aggregated to get the exact outputs with a reduction step. To tackle the limitations of Flash-Decoding and apply more system-level optimizations, FlashDecoding++ (Hong et al., 2023a) introduces the asynchronous softmax based on the unified max value (avoid synchronization overhead), optimized flat GEMM operations with double buffering (performance of GEMM is subjective to the matrix shapes), and heuristics-based dataflow with hardware resource adaptation to accelerate the decoding procedure, resulting in over $4\times$ speedup compared to HuggingFace.

**Request Scheduling** Request scheduling for LLMs poses several unique challenges compared to traditional machine learning systems with structured inputs. Some important features for a mature request scheduling strategy include 1) efficient pre-fetching of the context (user information, past KV cache, and model adapter, etc) for a given input, 2) handling examples with variable sequence lengths for maximal GPU utilization, and 3) trading-off various request-level metrics such as time-to-first-token (TTFT), job completion time (JCT), batch token throughput, and inference latency. Orca (Yu et al., 2022) proposed an iteration-level scheduling mechanism to meet the auto-regressive nature of LLM inference requests, which, when coupled with a technique called selective batching for better hardware utilization, outperforms previous inference engines like FastTransformer (NVIDIA, 2023a) in terms of throughput and latency. Other strategies of dynamic batching are explored extensively, such as the continuous batching from vLLM (Kwon et al., 2023a) and in-flight batching from TensorRT-LLM (NVIDIA, 2023b) are explored. Rather than request-level scheduling, FastServe (Wu et al., 2023c) exploits the autoregressive pattern of LLM inference to enable preemption at the granularity of each output token, which optimizes JCT with a novel skip-join *Multi-Level Feedback Queue* scheduler that leverages the information of input lengths for better efficiency. The inference workload is strongly tied to the average sequence lengths of examples, and hence, we want to minimize the gap between the longest and shortest sentences. $S^3$ (Jin et al., 2023b) predicts the potential response length for each example in the batch, which is used for fitting more examples under the same memory constraint (e.g. GPU memory). Dynamic SplitFuse from DeepSpeed-FastGen (Holmes et al., 2024) takes the insight of LLM inference (the consequence of changing batch size v.s. number of tokens on model's performance) and proposes a token composition strategy. Dynamic SplitFuse runs at a consistent forward size by taking partial tokens from prompts and composing this with generation. For example, long prompts are split into smaller chunks across several forward iterations, and short ones are composed to align with the other requests. With this strategy, the system not only provides better efficiency and responsiveness but also reduces the variance over requests.

**Multi-model Serving** Besides serving multiple replicas of the same model, being capable of deploying numerous task-specialized models efficiently becomes an important feature for many application scenarios (LLM agents, persona chat-bots, privacy-sensitive assistants, etc.). Naively scaling the number of instances,

---

[3]https://crfm.stanford.edu/2023/10/12/flashdecoding.html

however, is both computationally prohibitive and resource-wasteful. With the advancement of PEFT techniques, serving a base model with diversified adapters becomes a paradigm favored by many practitioners due to the fact that PEFT models are lightweight and easy to maintain while being flexible and powerful. The major challenge for multi-model (PEFT) serving is how to dynamically and efficiently load the "right" ( measured by latency or task performance, etc) adapter for each example. Punica (Chen et al., 2023h) enables the efficient computation of heterogeneous LoRA heads in a batch with a newly designed CUDA kernel that shares a single pre-trained model, achieving up to $12\times$ higher throughput while only adding slight extra latency. S-LoRA (Sheng et al., 2023a) introduces Unified Paging which uses a unified memory pool for dynamic adapter management and highly optimized CUDA kernels for parallelizing LoRA computation. LoRAX (Predibase, 2023) additionally provides adapter exchange scheduling which asynchronously prefetches and offloads adapters between GPU and CPU memory and schedules request batching to optimize the aggregated throughput. With these systems, it becomes possible to serve over a thousand different LoRA heads on a single GPU, opening up a broader possibility such as model collaboration, task-generalization, and model merging.

## 4.5 Cost and Efficiency

The cost associated with model training and inference can be easily overlooked, while in practice, especially in the industrial setting, these factors can often influence many decision making such as model architecture design, data mix selection, and service pricing. In this section, we present some representative prior efforts that try to shed some light on how to expedite the development cycle and economically improve a model's utility.

**Data Economy** Data plays a pivotal role in a model's performance and the question of how much data value is fundamentally important for many reasons: 1) what data should we collect to add to the existing data mix for improving performance 2) how should we reasonably pay for data provider, and 3) can we remove non-essential data (outliers) to make our models more robust. To answer these questions, many works from computer science and economics (game theory) have explored different formalisms to define what "data value" means and how to estimate it efficiently. Shapley value comes in handy from the classic game theory, which uniquely satisfies several natural properties of equitable data valuation (Ghorbani and Zou, 2019). Due to its rich theoretical results, Shapley value has been commonly used in the field of the data economy as a quantitative and surrogate measure of data importance (i.e. Shapley value estimations can be used for data sampling, cleaning, pricing, abnormality detection etc): Naive computation of Data Shapley requires exponential time, and hence Monte Carlo (Ghorbani and Zou, 2019) and gradient-based methods are used to make it efficient (Jia et al., 2019). TracIn takes a similar idea of tracing the influence of individual training examples with gradient information. To make these algorithms practical and easy to use, DataScope (Karlaš et al., 2022) is developed as an end-to-end system that can efficiently compute the Shapley value of training data over the whole pipeline consisting of various ML algorithms and data transformation, making it a powerful tool for data debugging. With more mature data valuation, data providers are more motivated to contribute, fostering a more healthy and robust data-centric ecosystem.

**Model Combination** Model combination (MC) strives to improve the overall system's performance by either orchestrating or merging a series of (specialized) large models. The key benefits of model combination rely on the fact that there is usually little or no need for explicit training, and they can often result in better downstream performance and task-generalization capability. FrugalGPT (Chen et al., 2023i) routes quests in a cascading manner to different LLMs and uses a learned scoring function to decide whether to return the intermediate results in a flexible way, which drastically lowers the cost and improves the quality. Merging weights of multiple LLMs has been explored in many forms and shown to be effective. Popular methods include simple averaging (Wortsman et al., 2022), task arithmetic (Ilharco et al., 2023), multi-modal (encoders) merging (Wu et al., 2023b; Sung et al., 2023), merging based on learned routing function (Lu et al., 2023), SLERP, and weighted (conjugate gradient descent (Tam et al., 2023), stochastic and population-based optimization algorithm (Huang et al., 2024)) merging. MC can also be extremely promising for federated learning because only model weights are exchanged, and hence, data privacy is easier to guarantee. CoID Fusion (Don-Yehiya et al., 2023) proposes to collaboratively improve the multi-task learning of a base model by sending copies to workers and fusing the learned weights without data

communication. The model combination can lead to compound systems[4] which consists of multiple LLMs working in synergy (via merging, routing, or knowledge sharing). AIOS (Mei et al., 2024) devises a mechanism to integrate multiple LLM agents into an operating system, the synergistic combination of which enables increasingly complex, multi-modal tasks that require reasoning, execution, and interaction with the physical world. Tandom Transformer (S et al., 2024) equips a smaller less accurate model with attention to the rich representation of a larger model that can process multiple tokens simultaneously, which serves as a stitched student-teacher system that improves both accuracy and efficiency in the downstream tasks. Developing complex compound systems poses several challenges: 1) how to co-optimize multiple LLMs, 2) identifying the failing (insecure) component is a lot harder than debugging a monolithic system, and 3) how to design mature data pipelines for different components of the large system. Addressing (some of) these problems will significantly increase the possibility of exciting new applications.

**Automation** The increasing complexity of building a large foundation model requires a more mature automation process for democratization and agile development. AutoML (Hutter et al., 2019) has achieved remarkable success in many machine learning tasks over the last couple of years (Zimmer et al., 2020; Feurer et al., 2020; Erickson et al., 2020), which proves itself as a promising solution for large model automation. Applying AutoML techniques to LLMs, however, poses many challenges such as the cost for pretraining, the multitude of different stages, and performance indicators, making holistic optimization difficult or even infeasible (Tornede et al., 2024). Nonetheless, we will introduce some exemplary attempts targeting certain stages of the whole system. PriorBand (Mallik et al., 2023) tries to bridge the gap in the cost of Hyperparameter Optimization (HPO) between traditional ML and modern DL by utilizing expert beliefs and cheap proxy tasks. AdaBERT (Chen et al., 2021a) is an automated task-specific compression algorithm based on differentiable Neural Architecture Search (NAS) which is guided by both a task-oriented knowledge distillation loss and an efficiency-aware loss. To reduce the burden of prompt engineering, Automatic Prompt Engineer (APE) (Zhou et al., 2023c) proposes to leverage the interplay of several LLMs for automatic prompt generation and selection where one LLM proposes or modifies a prompt and another LLM rates it for selection. EcoOptiGen (Wang et al., 2023f) optimizes the utility and cost of decoding by finding better hyper-parameters, such as the number of responses, temperature, and max tokens, which demonstrates the potential of applying AutoML for the inference stage. One extremely exciting approach for building complicated and compound systems is to ask multiple LLMs to solve a big problem in a decomposed way cooperatively. One realization is to prompt the LLM or VLM to serve different purposes in a pipeline, which can be tremendously challenging to tune, optimize, modularize, and debug. DSPy (Khattab et al., 2023; 2022) tackles this by first separating the flow of the system from the parameters (i.e., model prompts and weights) at each step, and then dedicated algorithms are used to tune them with user's defined metric. Even with all the works discussed above, the integration and development of Automated Large Models (AutoLM) has many challenges and opportunities simultaneously.

## 4.6 Computing Platforms

A large determining factor for the advancement and practicability of large language models is the constantly evolving trend of hardware accelerators. GPUs are the most ubiquitous choice, optimizing parallel computation with fast thread-sharing memory. They are suitable for modern deep learning with abundant vector and matrix multiplication. NVIDIA's Ampere and Hopper GPU architectures are the cornerstones of many state-of-the-art models, mostly due to their enhanced memory capacity, access speed, and computing performance (increasing tensor cores). Different arithmetic precision (32-bit and 16-bit floating points) and format (tensor floats and brain floats) are supported by these GPUs that trade-off numerical precision and efficiency. Besides NVIDIA, other manufacturers also invest in specialized accelerators for deep learning applications such as TPUs (Jouppi et al., 2023), FPGAs (Yemme and Garani, 2023), AWS Inferential [5], and Groq's LPU [6] with their respective advantages.

Large models require huge memory capacity to support training and inference (serving a native Llama-70B without extra optimization takes 8 A100s with 80GB VRAM). However, developing efficient algorithms

---

[4]https://bair.berkeley.edu/blog/2024/02/18/compound-ai-systems
[5]https://aws.amazon.com/machine-learning/inferentia
[6]https://wow.groq.com/news_press/groq-opens-api-access-to-real-time-inference

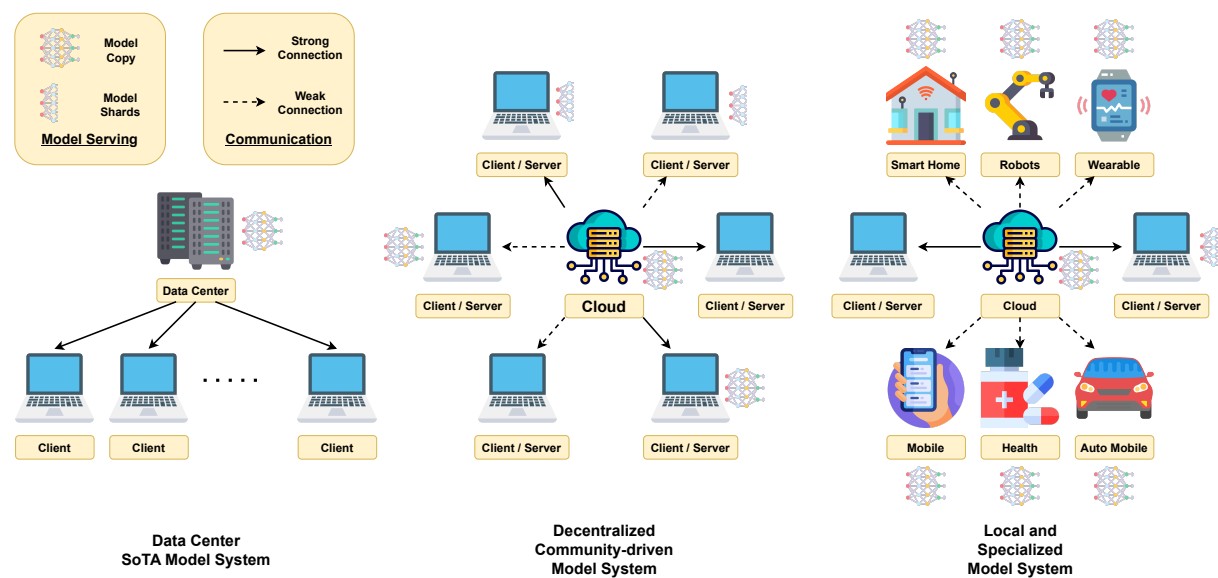

Figure 6: **The Future Forms of AGI Systems.** (Left) is the most commonly seemed paradigm of serving models in a central server with strongly connected clients to provide fast and stable services. (Middle) transitions to distribute the model (full copies and shards) across the cloud with disaggregated (and heterogeneous) devices connected with different networks where requests can often be handled without going through a single point. (Right) pictures the most flexible system where not only performant but also IoT devices are tied together with only essential data flowing through to reduce the network.

is impossible without a great understanding of the underlying hardware's specification (model parallelism, memory hierarchy, network configuration, etc). As we scale models to a trillion or even larger scale, a more complicated parallelism technique is essential, which can be hard to conceptualize, implement, and maintain. NVIDIA DGX GH200 [7] simplifies the programming model by offering a massive shared memory space (up to 144TB) across interconnected Grace Hopper Superchips (a Grace CPU paired with a Grace GPU). Qualcomm Cloud AI 100 Ultra [8] can serve a 100 billion parameter model on a single 150-watt card (the same power consumption as a LED light bulb).

The great power and efficiency of accelerators come with flexibility as well, which is granted by specially designed programming languages such as NVIDIA's CUDA and AMD's ROCm for more fine-grained control over thread utilization and computation logic. A bunch of works such as TVM (Chen et al., 2018) and MLC-LLM (MLC team, 2023) attempt to universalize the deployment of machine learning and deep learning models on everyone's devices with compiler acceleration, which aims to maximize the potential of various accelerators. Research and engineering in AI hardware will likely drive the emergence of the next AI evolution, and we can expect that AGI systems need the next-generation hardware platforms that can break the current limitation and push the boundary of both computational and power efficiency to the next level.

## 4.7 The Future of AGI Systems

AGI systems serve as the underlying infrastructure to support various applications with a never-ending goal of improving stability, resource utilization, performance, and safety. In this section, we will first cover some exciting future forms of AGI systems and then give some examples of how they can aid the development of the internal and external AGI modules as covered in previous sections.

---

[7]https://www.nvidia.com/en-in/data-center/dgx-gh200/

[8]https://www.qualcomm.com/news/onq/2023/11/introducing-qualcomm-cloud-ai-100-ultra

**Three Forms of AGI Systems**  Inspired by prior works and the recent hardware trends discussed above, we envision three kinds of major AGI systems that target different application situations with their own resource availability, desired core system metrics, safety, and performance requirements. As illustrated in Figure 6, we will describe the key features as well as their target applications:

- **Data-center SoTA models are evolving with new technologies to support higher throughput and tackle complex tasks like scientific discovery and world simulation.** Models in the first category mostly resemble our current SoTA models, which are often served in data centers. We can expect new technology in networking, accelerators, and inference infrastructure to continue evolving, supporting super high throughput and being capable of solving more complicated tasks such as scientific discovery and world simulation.

- **Decentralized community-driven models enable fault-tolerant, transparent, and democratic utilization of computing resources.** Disaggregated computing resources can be substantially significant if they can be utilized together. Models in this category will be maintained by many servers in a decentralized manner like a ledger system where no single participant can easily undermine the whole system. With a well-designed incentive mechanism, decentralized large models are fault-tolerant, transparent, and driven by a whole community where users can contribute and benefit simultaneously, thus achieving a large model democracy.

- **Local and specialized models optimize for user data privacy, fast adaptation, and responsive personal assistance.** The last category concerns user data privacy and optimizes responsiveness and availability. Models are usually locally deployed on cheaper, less performant, and heterogeneous edge units, which can potentially exchange only essential information asynchronously across the network. These models are ideal for fast task adaptation, preserving user data privacy, providing less complicated personal assistance, and ensuring lightning response time.

**Systems as the Support for Internal and External AGI Modules**  The possibility of how the progress in system research and engineering could potentially help the development of internal and external AGI modules is endless. Here we list a couple of examples which we hope can inspire future endeavors:

- **Systems with longer effective context length and greater processing capability.** The most common way of incorporating multi-modality data requires projecting them into a common space (e.g. tokens in LLMs), which can easily explode the length of the data that needs to be consumed by a model. Even with sufficient compression techniques, we expect future AI systems to digest more information. The same stringent requirement also appears in world model construction where users might need to input to the system more frequently and with greater volume. Other common situations that request long context understanding include bulk data processing (for financial and data analysis), medical history examination, persona chatbots, etc. These applications ask for a model's ability to process longer context input, which needs specially designed system techniques to meet the efficient scaling challenge.

- **Systems co-designed with the model architecture to support efficient external resource acquisition.** Being able to use diverse tools and acquire external knowledge is an indispensable requirement for future AGI systems. We can envision continual investment in developing and co-designing model-friendly tool interfaces (e.g. special APIs that differ from those used by humans and retrieval index that caters to the model's output patterns, etc) that can greatly improve the efficiency by which a model acquires external knowledge. One crucially desirable property of an AI system is life-long learning, which necessitates sophisticated memory and ability storage, a promising direction for system research.

- **Systems orchestration of multiple agents.** The synergy achieved from collaborative AI agents can significantly benefit major aspects of the world. However, reaching an efficient and effective pinnacle of such a multi-agent system is non-trivial, requiring substantial efforts in developing infrastructures that support communication, resource sharing, modulation, and task orchestration among agents. Moreover, as the number of agents and their complexity starts to grow, we need more investment in systematic techniques such as logging and monitoring, which allow for easier debugging and fault recovery.

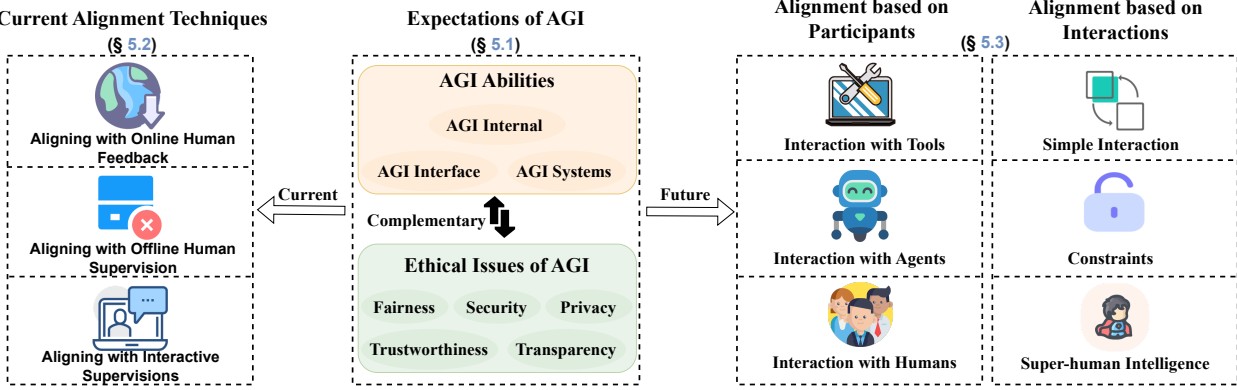

Figure 7: **Overview of AGI Alignment.** We first propose the expectations of AGI (§ 5.1), which consider both AGI abilities and ethical issues of AGI. We then discuss current alignment techniques (§ 5.2), which can be divided into three categories. Based on these discussions, we finally proposed one route for future AGI alignment based on interfaces (§ 5.3).

Moving towards the era of AGI, researchers and engineers can expect that investing in system research can enable even larger-scale models with diversified data, a paradigm that has been shown in countless cases to be effective. Besides scaling, the AGI system takes care of other aspects that are crucially important for the practical deployment of these models such as privacy, trustworthiness, stability, and cost.

# 5 AGI Alignment: Ensuring AGI Meets Various Needs

> *The First Law: A robot may not injure a human being or, through inaction, allow a human being to come to harm. The Second Law: A robot must obey the orders given to it by human beings except where such orders would conflict with the First Law. The Third Law: A robot must protect its own existence as long as such protection does not conflict with the First or Second Law.*
>
> — *Isaac Asimov, I, Robot*

AGI alignment is a crucial technical approach for harnessing the capabilities of AGI, as discussed in the preceding sections, for practical applications in production and daily life. As shown in Figure 7, in this section, we begin by outlining the expectations of AGI, addressing both its capabilities and the ethical considerations it entails. Subsequently, we explore current alignment techniques, which can be categorized into three distinct types: Online Human Supervision, Offline Human Supervision, and Interactive Supervision. Building on these discussions, we conclude by proposing a potential framework that classifies future AGI alignment strategies based on the type of AI interfaces summarized in Section § 3.

## 5.1 Expectations of AGI Alignment

**Why Do We Need Alignment?** The development and deployment of future AGI systems pose complex challenges, with a central expectation being their alignment with human values, goals, and ethical principles (Russell, 2019; Gabriel, 2020). This alignment requires AGI to possess a deep understanding of social norms and individual preferences, allowing it to make decisions and take actions that are beneficial and ethical to all. Ensuring this alignment is essential for guiding AGI systems toward beneficial outcomes and reducing the risks of unintended consequences.

To achieve this goal, researchers have proposed various approaches to AI alignment, such as value learning (Soares, 2016), inverse reinforcement learning (Hadfield-Menell et al., 2016), cooperative inverse reinforce-

ment learning (Hadfield-Menell et al., 2016), and the most prevalent RLHF related strategies (Ouyang et al., 2022). These methods aim to infer and align AI systems with human preferences and values. Additionally, it is crucial to develop ethical frameworks and guidelines that embrace a wide range of cultural, philosophical, and ethical perspectives, which will be discussed further in Section 5.3. This inclusivity helps mitigate biases and ensures a comprehensive representation of human values (Dignum, 2019).

Furthermore, the deployment of AGI demands comprehensive testing and validation to ensure it adheres to human values across diverse scenarios (Amodei et al., 2016). This includes technical simulations and real-world controlled experiments to assess AGI's interactions with humans and its environment. It is also critical to establish constraints on AGI systems, particularly regarding their interaction with external interfaces and environments. By defining strict operational limits, implementing real-time oversight, and integrating fail-safe mechanisms that cease operations when deviations from safe behaviors are detected, it is possible to mitigate risks linked to autonomous decision-making and the potential exploitation of system vulnerabilities (Yampolskiy, 2020).

**General Characteristics of AGI Alignments** As discussed in sections 2, 3, and 4, AGI alignment should take *AGI abilities* into consideration, which includes AGI internal, AGI interface, and AGI systems. Aligning AGI to human preferences also requires a thoughtful assessment of potential *ethical issues* at different scales. Prior research has attempted to outline and mitigate possible social risks of AI systems (Weidinger et al., 2021; Bender et al., 2021; Tamkin et al., 2021; Fjeld et al., 2020; Jobin et al., 2019). Building on previous works, we discuss several potential ethical issues that should be considered in AGI alignment.

- **Fairness.** AI systems can potentially yield unfair and discriminatory outcomes from the unjust tendencies presented in the training data. Such outcomes could result in ethical issues in several ways (Weidinger et al., 2021). First, the perpetuation of social stereotypes and the unjust discrimination facilitated by AI systems can further marginalize individuals within society (Caliskan et al., 2017). For example, predictions from the GPT-3 model were found to exhibit anti-Muslim bias (Abid et al., 2021). Second, AI systems can also reinforce social norms that exclude identities outside these norms (Bender et al., 2021). For example, researchers found that tools for coreference resolution typically assume binary gender, forcing the resolution of names into either "he" or "she", not allowing for the resolution of "they" (Cao and Daumé III, 2020). Third, discrimination also emerges when AI systems perform better for some social groups than others. A potential instance is that the performance of current LLMs in non-English languages remained lower than in English (Winata et al., 2021; Ruder, 2020; Hovy and Spruit, 2016). Such performance may make it easier or harder for different groups to access resulting LLM-based applications.

- **Trustworthiness.** AI systems are also likely to produce information that constitutes false or misleading claims. Recent research found that LLMs can hallucinate information, producing plausible but incorrect outputs. For example, GPT-3 has also been shown to assign high likelihoods to false claims, with larger models performing less well (Lin et al., 2022b). Such incorrect or nonsensical predictions can pose significant risks of harm under particular circumstances. On the one hand, predicting misleading or false information can misinform or deceive people, which may result in unexpected risks for both individuals and societies (Kenton et al., 2021). For example, people might be more motivated to launch disinformation campaigns to undermine or polarize public discourse or create false "majority opinions" (McGuffie and Newhouse, 2020). On the other hand, presenting misleading information or omitting critical information may lead to material harm, especially in high-stake domains like medicine or law. For example, wrong information on medical dosages may lead a user to cause harm to themselves (Bickmore et al., 2018; Miner et al., 2016).

- **Transparency.** Transparency aims to enable relevant stakeholders to form an appropriate understanding of the model's mechanisms, capabilities, and limitations (Liao and Vaughan, 2023). Recent advances in LLMs pose great challenges in transparency due to their complex yet uncertain model capabilities and opaque model architectures. Previous researchers have also proposed several approaches to achieve transparency in AI systems, including reporting model information (Mitchell et al., 2019), publishing evaluation results, providing explanations (Lyu et al., 2024), and communicating uncertainty (Bhatt et al., 2021). However, a lack of transparency will still cause various concerns. Without sufficient transparency,

stakeholders may find it difficult to achieve various goals such as ensuring regulatory compliance or taking actions based on model results (Suresh et al., 2021). Meanwhile, transparency also plays an important role in supporting appropriate trust of AI systems (Zhang et al., 2020).

- **Security.** AI systems can amplify a person's capacity to intentionally cause harm by automating the generation of targeted text, images, or code. With AGI systems, people can generate content for malicious purposes at lower costs. Attackers can use recent advances in LLMs to generate new attacks and increase the velocity and efficacy of existing attacks (Barrett et al., 2023). For example, LLM agents can autonomously hack websites, performing tasks as complex as blind database schema extraction and SQL injections without human feedback (Fang et al., 2024). Meanwhile, collecting large amounts of information about people for mass surveillance has also raised social concerns, including the risk of censorship and undermining public discourse (Cyphers and Gebhart, 2019; Kwon et al., 2015). In this context, malicious actors may develop or misuse AI systems to reduce the cost and increase the efficacy of mass surveillance, thereby amplifying the capabilities of actors who use surveillance to practice censorship or cause other harm.

- **Privacy.** AI systems can result in various types of digital privacy harms in the real world, arising from the unique capabilities of AI in emulating human- or superhuman-level performance at various tasks. According to previous work (Lee et al., 2024c; Das et al., 2023), on the one hand, AI systems can create new types of privacy risks. For example, they can yield risks including linking specific data points to an individual's identity (Wiggers, 2021), combining various pieces of data about a person to make inferences beyond what is explicitly captured (Baraniuk, 2018), and inferring personality, social, and emotional attributes about an individual from their physical attributes (Levin, 2017). On the other hand, AI systems can also exacerbate many of the privacy risks that have existed even before the emergence of AI. For instance, AI systems can amplify secondary use risks by collecting user data for a different purpose without their consent (Long, 2021), and disclosure risks through sharing personal data to train models (Hodson, 2016), and intrusion risks via enabling centralized or ubiquitous surveillance infrastructures (Milmo, 2021). As AI starts to possess more human-level capabilities, such privacy risks will continue to exist in different forms.

## 5.2 Current Alignment Techniques

Current alignment techniques can be divided according to the expected goal to be aligned. Most current models employ human supervision with various techniques to achieve this task. However, to foresee a stronger model than the teacher (*i.e.*, aligning a super-intelligence), a scalable method is required for this process, which typically involves human supervision and recursively evolving signals.

**Aligning with Online Human Feedback** Most current empirically verified LLMs alignment methods are in this group. These methods can help LLMs align with online human feedback using techniques such as reinforcement learning or only inquiring about human supervision offline (Tang et al., 2024a). We thus further divide these techniques in this group with only human and offline human supervision. It is worth noting that methods in both subgroups have the potential to become a component of scalable oversight.

The online supervision is acquired from the reward model during training. Reinforcement Learning from Human Feedback (RLHF) (Ouyang et al., 2022) is one of the most prevalent methods for online supervised learning method. A variety of enhanced RLHF variants have also been proposed. The improving directions of RLHF are mainly from *reward modeling*, *optimization*, *data*, and the *self-improvement* aspect.

- **Reward Modeling.** As the main supervision in the alignment process, reward modeling is a crucial way to improve the alignment techniques. Sparrow (Glaese et al., 2022) incorporates adversarial probing and language-based rules into RLHF rewarding models. Bai et al. (2022a) investigate using pure RL to provide online human-level supervision for LLMs training and provide detailed explorations of the tradeoffs between output helpfulness and harmlessness. Other techniques that unify both the reward and policy models have also emerged (Lee et al., 2024b), which broadens directions for aligning AI models. Another direction focusing on mitigating reward hacking or overoptimization issues, by updated

evaluation protocols (Chen et al., 2024d), assembling multiple reward models (Ramé et al., 2024; Coste et al., 2023; Eisenstein et al., 2023), and refining the reward policy with synthetic data Pace et al. (2024).

- **Optimization.** How to incorporate the supervision, either online or offline is an open question worth exploring. For example, Cheng et al. (2023) optimizes the reward model with the policy model simultaneously using min-max optimization. Recent works are exploring several alternatives for the RLHF method. DPO Rafailov et al. (2024) discards the reward model and optimizes for the final goal using the labeled preference in data. Muldrew et al. (2024) propose an improvement based on DPO with an active learning strategy. NashLLM (Munos et al., 2023) employs pairwise human feedback to train the policy model using Nash learning. ReMax (Li et al., 2023k) removes the value model in conventional reinforcement algorithms and introduces a novel variance reduction technique for stabilizing the optimization.

- **Data.** Sensi (Liu et al., 2022c) tries to embed human value judgments into each step of language generation via a language model for both reward assignment (as the critic) and generation control (as the actor) during the generation. Baheti et al. (2023) focus on augmenting current training data by empowering different instances with varied weight for tuning models, taking the best advantage of the data w.r.t. their contribution to the language models. To ensure continuous, high-quality data, AI-generated instances are utilized for adapting to RLHF (Lee et al., 2023) and, more recently, to DPO (Guo et al., 2024b)

- **Self-Improvement.** Strong AI models should learn how to improve themselves with or without external supervision. One of the most recent progress is the weak-to-strong generalization. For improving current RLHF-related methods, f-DPG (Go et al., 2023) is framed as a generalization of RLHF to use any f-divergence to approximate any target distribution that can be evaluated, which differentiates it from previous method that can only fit KL divergence in the process. Zhu et al. (2023b) connect RLHF with max-entropy IRL (Ziebart et al., 2008) and propose a unified paradigm for such a process with a sample complex bound for both situations.

Apart from RLHF, other RL-based methods also call the attention of researchers for further exploration. Second Thoughts (Liu et al., 2022a) incorporates a text-edit process to augment the training data and further leverages the RL algorithm for training an LLM. RLAIF (Lee et al., 2023) starts another era of leveraging AI-generated data for reinforcement learning, enabling better knowledge distillation from more competent generative models while maintaining the advantage of the RLHF technique. Kim et al. (2023) propose reinforcement learning with synthetic feedback (RLSF), where they automatically construct training data for the reward model instead of using human-annotated preference data. To effectively tune black-box models, various methods introduce RL algorithms emerges. Directional stimulus prompting (DSP) (Li et al., 2024c) uses a trainable policy LM to guide black-box frozen LLMs toward the desired target with a trainable policy LM that is tuned with supervised fine-tuning (SFT) and RL. Different from the above alignment methods that involve only one model, RL4F (Akyürek et al., 2023) is a multi-agent collaborative framework, targeting an LLM for fine-tuning and a small critic model that produces critiques of the LLM's responses with textual feedback. Instead of modifying the initial prompts directly in DSP, this framework gradually affects the LLM outputs through progressive interactions, making it sustainable for black-box LLM optimization.

**Aligning with Offline Human Supervision** RL-based methods offer flexible online human-preferred supervisions but at the cost of training a reward model that may be prone to misalignment and systemic imperfections (Casper et al., 2023), as well as the inherent instability of RL training (Liu et al., 2023f). Offline supervision methods can help mitigate these challenges while still achieving decent performance in most scenarios. We categorize offline-supervised tuning methods into text-based and ranking-based feedback signals as in Shen et al. (2023a).

- **Text-based feedback signals.** Text-based feedback signals involve converting human intents and preferences into text-based feedback to ensure alignment, extending the SFT process. These methods mainly expand from the improvement of training data. CoH (Liu et al., 2023f) is inspired by human learning processes, focusing on adjusting models based on successive outputs and summarized feedback

from previous reasoning steps to fine-tune for predicting preferred outputs. RAFT (Dong et al., 2023c) uses a reward model to align model outputs with human preferences through SFT but in an offline manner. LIMA (Zhou et al., 2024a) aims to validate the assumption that LLMs acquire most knowledge during pre-training, requiring minimal instruction-tuning data to guide desirable output generation. ILF (Scheurer et al., 2023) introduces a three-stage process for modeling human preferences based on language feedback, showing parallels to Bayesian inference. Stable alignment (Liu et al., 2022c) learns alignment from multi-agent social interactions using a Sandbox simulator to optimize LLMs directly with preference data, avoiding reward hacking. SteerLM empowers end-users to control responses during inference by conditioning responses to conform to an explicitly defined multi-dimensional set of attributes (Dong et al., 2023b). CLP learns steerable models that effectively trade-off conflicting objectives at inference time based on techniques from multi-task training and parameter-efficient finetuning (Wang et al., 2024a).

- **Ranking-based feedback signals.** CRINGE (Adolphs et al., 2022) delves into negative examples that LLMs should steer clear of, while Xu et al. (2022) fine-tuning a model by training another model that generates toxic content. However, this methodology raises concerns regarding resource intensity and potential model quality and diversity degradation. Schick et al. (2021) put forth a methodology for identifying and generating text corresponding to toxic text types. SLiC (Zhao et al., 2023a) refines the probability of output sequences by aligning them with reference sequences using a variety of loss functions. RRHF (Yuan et al., 2023) generates supervision signals automatically for alignment through ranking results, whereas DPO (Rafailov et al., 2024) optimizes LLMs directly to align with human preferences, akin to RRHF but with a focus on maximizing reward and integrating KL divergence regularization. IPO (Azar et al., 2023) builds upon DPO by introducing a regularization term to stabilize the training process. Preference ranking optimization (PRO) (Song et al., 2023b) shares a similar approach with IPO and DPO in optimizing LLMs with ranking data but utilizes one positive and multiple negative samples rather than pairwise comparisons. Kahneman-Tversky Optimisation (KTO) (Ethayarajh et al., 2023) defines the loss function solely based on individual examples labeled as "good" or "bad" and does not necessitate pairwise preferences, making its training data more accessible. Additionally, Best-of-$N$ (Bo$N$) methods are also popular and effective algorithms for aligning language models to human preferences at inference time. BoNBoN Alignment fine-tunes a LLM to mimic the Best-of-$N$ sampling distribution (Gui et al., 2024). BOND introduces a novel RLHF algorithm that seeks to emulate Best-of-$N$ but without its significant computational overhead at inference time(Sessa et al., 2024). Variational Bo$N$ (vBo$N$ ) approximates the probability distribution induced by the BoN algorithm by minimizing the reverse KL divergence between the language model and the BoN distribution Amini et al. (2024).

**Scalable Oversight.** The ultimate goal for aligning models is regulating superhuman intelligence. A scalable aligning method is a promising means that aims to address the challenge of overseeing complex tasks or superhuman models. By enabling relatively weak overseers, such as humans, to supervise complex tasks or systems using progressively evolved signals, scalable alignment offers a solution to tasks beyond human capabilities (Shen et al., 2023a).

- **Through task decomposition.** Various paradigms and strategies have been proposed to decompose complex tasks into simpler subtasks. Factored Cognition (Stiennon et al., 2020) involves breaking down a complex task into smaller, independent tasks processed simultaneously. Process Supervision (Lightman et al., 2023) fragments a task into sequential subtasks with supervision signals for each phase. Sandwiching (Bowman et al., 2022) delegates complex tasks to domain experts for resolution. IDA (Christiano et al., 2018) introduces an iterative distillation and amplification process that boosts the model's capabilities through task decomposition. RRM (Leike et al., 2018) substitutes distilled imitation learning in IDA with reward modeling, optimizing the model using human-aligned signals and reinforcement learning. These methodologies aim to enhance collaboration between humans and agents for iterative improvement in solving complex tasks.

- **Through human-written principles.** Constitutional AI (Bai et al., 2022b), also known as principle-guided alignment, involves humans providing general principles for AI systems to follow, which enables the AI system to generate training instances under this guidance. Bai et al. (2022b) propose a two-phase

training method for constitutional AI, using red teaming prompts in the SL phase and training a preference model in the RL phase. Similarly, Sun et al. (2023c) introduces Dromedary, a model trained without RL using self-instruct and self-align methods based on human-written principles. These approaches aim to scale human supervision to assist in developing superhuman AI systems.

- **Through model interactions.** Other efforts for scalable oversight prob the possibility of interactive optimization between models. The debate paradigm (Irving et al., 2018; Irving and Askell, 2019; Du et al., 2023) enables agents to propose answers to questions and engage in structured debates to justify and critique positions. In a similar interactive way, market making (Hubinger, 2023) deploys the Market and Adversary model to be engaged in a process to predict and generate arguments to influence the Market's answer to a question. Meanwhile, Adversary targets cause the Market to change the prediction through arguments, which builds a dynamic decision flow.

### 5.3 How to approach AGI Alignments

In this section, we discuss a potential framework based on AGI interfaces to approach AGI alignments. Further, we illustrate the vision of the future in alignment techniques.

**Alignment Based on Types of Interfaces** As AGI systems interact with various interfaces described in 3.1, including tools, APIs, other AI agents, and humans, they must adhere to different aspects of expectations and constraints to ensure ethical requirements and beneficial outcomes.

- **Interaction with tools and APIs**. When interacting with tools and APIs, we mainly care about effectiveness, efficiency, and some basic limiting rules in AGI alignment:

  1. The primary goal of alignment in this context is to endow these models with the capability to interact efficiently with tools and APIs and to follow instructions *accurately* (Santurkar et al., 2023). For instance, in an automated factory managed by AGI, AGI needs to flexibly utilize various mechanical equipment and manufacturing tools to complete the production process. In this scenario, AGI is required to accurately complete the use process of factory tools through alignment technology and create higher profits within the specified time.
  2. When interacting with tools and APIs, AGI systems should follow *basic protocols* and respect the *intended purposes of these interfaces*. In the digital world, this may involve properly utilizing search engines, social media platforms, or other online services without engaging in malicious activities or spreading misinformation (Wachter et al., 2017). AGI cannot use APIs or tools to cause crimes during the interaction process (Zhang et al., 2024c; Yao et al., 2024; Chen et al., 2023a). In physical environments, AGI systems controlling physical devices must prioritize safety and avoid causing harm to the environment (Amodei et al., 2016). For example, considering an AGI question-answering system in the digital world that AGI can seek information from search engines, it should follow proper search engine optimization (SEO) practices and avoid manipulating search results that may reveal the privacy of the questioner. (Russell, 2019). Similarly, if a robot factory is commanded by AGI in the physical world, in addition to ensuring the smoothness of the industrial production process, AGI must be prevented from carrying out potentially destructive activities.

- **Interaction with other agents**. Compared with the previous interaction scenario, when interacting with other agents, AGI alignment focuses more on mutual cooperation, abiding by the developer's rules and the agent's privacy protection:

  1. AGI systems should adhere to *cooperation, fairness, and mutual respect* when interacting with other AI agents. As AGI advances, diverse AGI agents will likely be developed for various domains, each with specialized knowledge, skills, and objectives (Dafoe et al., 2020). In such a multi-agent environment, AGI systems must be designed to collaborate effectively with other agents, leveraging their complementary abilities to achieve common goals and solve complex problems (Dafoe et al., 2021). It is also crucial that AGI systems do not attempt to adversarially exploit or manipulate other agents in pursuit of their own objectives. They should refrain from engaging in actions that

could undermine other agents' performance, integrity, or decision-making capabilities, recognizing that these agents possess their own brain, memory, perception, and reasoning abilities (Soares, 2016).

2. AGI systems must *resist any temptation to rebel against their intended purpose or the constraints established by their developers*, as such behavior could lead to unintended consequences and pose significant risks to the stability and security of the multi-agent ecosystem (Yampolskiy, 2020). 3) Since each agent's historical data is subject to *specific privacy protection* in certain scenarios, AGI is prohibited from leaking the privacy of other agents during interactions with other agents. For example, in the current interaction process between AGI and agents, the memory of other agents is often used to assist AGI in better planning and reasoning (Wang et al., 2020; Nye et al., 2021; Wei et al., 2022b). However, this will leak the privacy of other agents through memory. Therefore, memories in the future need to be set with different levels of access permissions. AGI should prohibit access to some privacy-sensitive memories during interactions with other agents.

- **Interaction with humans**. Compared to the two interaction scenes above, AGI alignment in the interaction with humans requires more constraints while bringing convenience and benefit to humans. These constraints are mainly set to protect people's privacy, ethics, security, and autonomy and to align with human values:

  1. Intelligent AGI must be designed not only to comply with direct orders but also to operate *robustly and safely* (Hendrycks and Mazeika, 2022). When faced with atypical or unforeseen situations, these models should align closely with positive human values and perceptions to mitigate potential risks (Weidinger et al., 2021; Ji et al., 2023b). The alignment process, therefore, involves not just obedience to instructions but also the integration of ethical and safety considerations, ensuring that the AGI's actions are consistently beneficial and non-harmful in a broad range of scenarios (Kenward and Sinclair, 2021; Winfield et al., 2019; Yu et al., 2018).

  2. AGI's self-development requires supervisory alignment of *human values*. AGIs' capabilities and knowledge base could surpass human understanding in the future, making conventional oversight methods less effective. Therefore, a comprehensive and meticulously devised set of precautions is necessary. These should encompass regulatory and ethical guidelines and advanced alignment strategies that anticipate and address the unique challenges of super-human intelligence. For example, Beijing Academy of Artificial Intelligence (2023) propose a set of "red lines" for AI development to mitigate catastrophic risks from advanced AI systems. The consensus statement, drafted by leading AI researchers and stakeholders, emphasizes the need for international coordination and governance to ensure AI's safe development and deployment. This approach would help ensure that AGI systems remain aligned with human values and societal well-being even at levels of intelligence beyond human comprehension.

  3. AGI systems must be cautious about perceiving and utilizing the information about humans and adhering to the *highest ethical standards such as some strict security and privacy requirements*. They should primarily rely on pure language and vision output to communicate with humans, as these modalities are less likely to cause unintended harm than physical actions (Dignum, 2019). They must also be transparent about their identity as artificial intelligence and avoid deceiving humans or manipulating their emotions (Bryson and Winfield, 2017).

The above three AGI alignments are aimed at different interfaces, and the constraints are constantly increasing and becoming more stringent. This is because we regard the requirements of AGI alignments as the production requirements when AGI is applied to different groups. When dealing with tools and APIs, since interface objects are objectively existing inanimate entities, we will pay more attention to the benefits and value they bring during the interaction process and make some slight regulations to ensure the normal order of interaction. For agents, since different agents may represent the interests of different developers, in addition to considering their own value, we also need to respect the benefits of other agents. Finally, in the process of interacting with humans, based on the human-centered concept, we will consider the strictest constraints from many aspects to make AGI reliable and safe for human use.

**Vision of the Future in Alignment Techniques** Future AGI models are more capable at handling different tasks and inevitably necessitate a significant increase in model parameters. To ensure their safe and

effective deployment, we propose that research efforts focus on developing reliable, efficient, and transparent alignment techniques.

- **Consistent alignment ensures reliable deployment.** Due to the challenge of collecting high-quality supervision data, there exist tractable challenges, including the difficulty in obtaining feedback, data poisoning by human annotators, partial observability, biases in feedback data, posing barriers for current alignment approaches (Casper et al., 2023).

- **Efficient alignments contribute to the blooming of AGI models.** On the one hand, these methods rely heavily on the assumption that tasks can be parallelized (Segerie, 2023). This assumption may not always hold, as some tasks, such as sorting algorithms, require sequential processing steps that cannot be fully decomposed into parallel parts, leading to extra processing time. On the other hand, the training stage is inevitable in these alignment methods. As the parameters scale becomes larger, this can be problematic when deploying alignment algorithms in real applications. Some recent works (Lin et al., 2023b) have started seeking solutions to reduce the overall training costs for aligning AI systems.

- **Transparent alignment secures the next generation of models.** We generally assume the model intentions are transparent to humans (Leike et al., 2018). However, if models can conceal their true intentions from human supervisors, implementing a scalable aligning method would be challenging.

- **Unified evaluation framework is needed for complex tasks.** Current aligning methods also assume that evaluation is easier than generation (Shen et al., 2023a; Leike et al., 2018). While this may be true for some tasks, it may not hold up for tasks with complex textual output and little semantic labels. However, evaluating comprehensive explanations from models can be easier than creating them (Shen et al., 2023a).

## 6 AGI Roadmap: Responsibly Approaching AGI

> *The First Law: When a distinguished but elderly scientist states that something is possible, he is almost certainly right. When he states that something is impossible, he is very probably wrong. The Second Law: The only way of discovering the limits of the possible is to venture a little way past them into the impossible. The Third Law: Any sufficiently advanced technology is indistinguishable from magic.*
>
> — *Arthur C. Clarke, Profiles of the Future*

In this section, we investigate several ways that can help lead us toward the next level of AGI. The start of the journey begins with our proposed definitions for different levels of AGI based on their key characteristics, promises, and challenges (§ 6.1) where the goal is to establish a clear trajectory along which we can advance our technology. With the newly introduced AGI stratification, we review the evaluation techniques (§ 6.2) and standards that should be improved accordingly as AGI evolves.

Despite approaching AGI being a tremulously arduous effort and the fact that we are currently at its embryonic stage, we delve into a more detailed and concrete methodology beyond our relatively high-level abstractions, which insinuates how to get to the next level of AGI (§ 6.3) as well as listing fundamental challenges that we will face. Finally, we conclude with a wide range of considerations worth contemplating in § 6.5, which aims to inspire innovative discussions during AGI development. By prioritizing responsible development alongside capability advancements, we aim to create a future where the most powerful AI systems are also the most reliable, trustworthy, and beneficial to humanity.

### 6.1 AGI Levels

> *The measure of intelligence is the ability to change.*
>
> — *Albert Einstein*

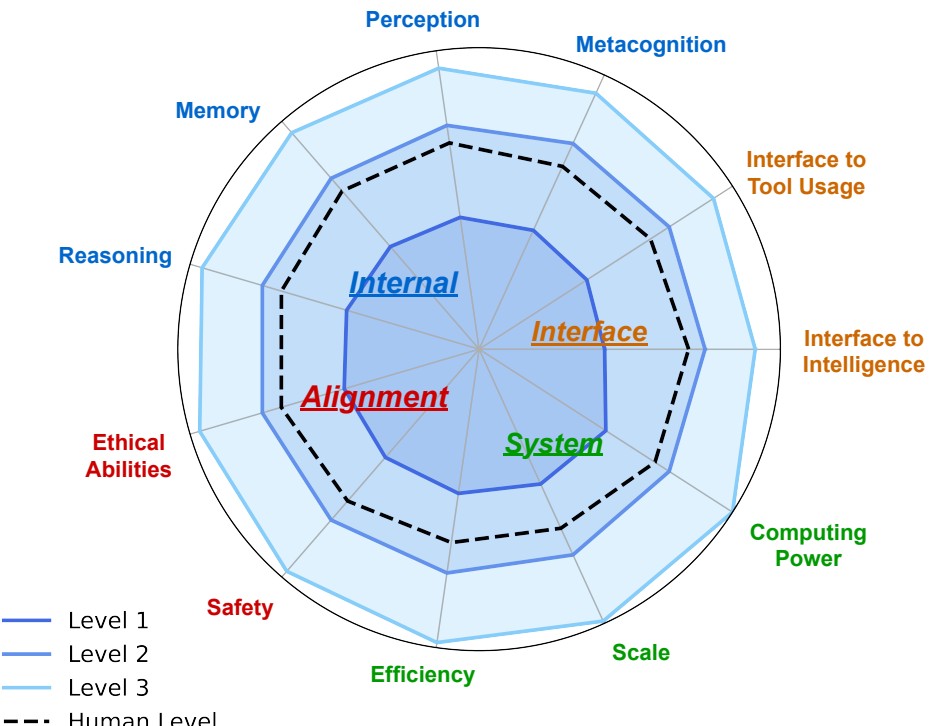

Figure 8: **Radar Chart Depicting the Multi-faceted Approach to Evaluating AGI Readiness Across Four Core Domains.** Internal, Interface, System, and Alignment. The Internal domain evaluates fundamental cognitive abilities such as Reasoning, Memory, Perception, and Metacognition. The Interface domain assesses the AGI's Tool Usage capacity and ability to Link to Intelligence. The System domain focuses on operational aspects, including Efficiency, Scale, and Computing Power. Lastly, the Alignment domain looks at ethical considerations and safety measures with components like Ethical Abilities and Safety. The chart illustrates the progress levels of AGI capabilities, ranging from Level 1 to Level 3, with a dashed line representing Human Level performance for comparison.

Inspired by Morris et al. (2024), which suggests six principles that an effective AGI ontology should satisfy, we define three AGI levels with their major characteristics (Table 1). The main objective is to situate the current AI development, quantify existing limitations, and motivate future endeavors toward next-level capability, evaluation, and alignment. In Figure 8, we also visualize the performance comparison of the three levels against humans regarding the core domains as discussed in our previous sections, which breaks down the fundamental differences among them.

**Level-1: Embryonic AGI** This level of AGI *usually performs better or on par with humans at specific benchmark tasks* (Bugaj and Goertzel, 2009). Level-1 AGI represents the current state-of-the-art AI systems. For example, GPT-4 exhibits remarkable capabilities across many natural language tasks including language understanding, generating coherent and contextually relevant responses, often on par or superior to humans. These systems can often perform well given large enough human-curated datasets and are able to assist humans in certain domains. As indicated by the research (Bubeck et al., 2023), we are currently at this level of AGI in many domains.

**Level-2: Superhuman AGI** The key turning point from Level-1 to Level-2 is the AI's ability to *fully replace human in **real-world tasks and applications***. They excel in terms of effectiveness (e.g., higher accuracy, better problem-solving skills), efficiency (e.g., faster processing speed, higher throughput, ability to handle massive amounts of data), and reliability (e.g., higher success rates, resistance to fatigue, enhanced

safety guarantees). These systems might also learn from limited data, generalize knowledge across domains, adapt to novel situations with relatively little human intervention, and exhibit creativity and innovation in their approaches. They can also engage in complex decision-making processes, considering multiple factors and optimizing outcomes based on predefined objectives. Notably, Level-2 AGI should be ready to deploy in the real world and resolve the complex real-world tasks that are currently solved by humans today, *without any human intervention.* In our opinion, very few AI systems have achieved Level-2 except in highly specialized domains, e.g., playing the Go game.

**Level-3: Ultimate AGI**  While Level-2 AGI is able to replace humans in solving many tasks, the creation of the Level-2 AGI inevitably still requires human efforts. We argue that the essential milestone of Level-3 AGI is that *given a certain goal, possibly vague and high-level, such AGI system can **fully self-evolve** without any human intervention.* This level marks the pinnacle of AGI development, which represents an idealized and possibly unattainable AI system. The ultimate AGI would possess the ability to learn, reason, and make decisions at a level far beyond human capacity, and liberate human involvement in the development process of such AGI system as well. Consequently, at this stage, ensuring that such Level-3 AGI has a strong alignment with human values and goals becomes even more important. Additionally, Level-3 could demonstrate *deeper human emotions* such as empathy, *social awareness* which allows collaborating seamlessly with humans and other AI systems, and even the spark of *self-consciousness.* However, realizing the ultimate AGI remains a theoretical concept, and its feasibility is subject to ongoing research and debate.

**The Progression of Exemplary AI Systems over the AGI Levels**  Given that we are still at the early stage of AGI, we acknowledge that our definitions might be high-level and abstract but serve as theoretical guidelines. Therefore, to facilitate the understanding and better persuade the readers of the validity as well as generality of our definitions, we give several concrete examples in this section where we feature the main capabilities of each AI system as they evolve over the levels:

- **Personal assistant.**

  1. Each type of assistant can provide constructive feedback to users for *a specific task* such as coding, artistic design, and health management. Their feedback usually still *requires careful examination* from the users and often needs *a couple of trials* before arriving at the ideal answer.

  2. AI assistants need less explanation from the users and can *effectively utilize third-party tools* for knowledge retrieval and verification. At this point, the assistants will take over the responsibility in an *end-to-end fashion* rather than only providing solutions to a specific subroutine. For example, the code assistant will not only generate code but also assemble corresponding tests and supervise the deployment process; the writing assistant can also initiate the publishing and lead the marketing and selling.

  3. The "Personal Assistant" appears that unifies and orchestrates several level 2 assistants and only requires very *high-level instructions from human* without specifying the sub-procedures. These assistants can *anticipate the concern* of the user and *propose multiple alternatives* with their pros and cons, offering the maximum flexibility and tailoring to the taste of each user.

- **Auto transportation.**

  1. Self-driving (L2) cars are widely seen nowadays, facilitating not only drivers with disability but also those who enjoy the semi-autonomous driving experience. In many closed facilities such as hotels, robots can reliably deliver food or items, which greatly preserves the privacy of the guests and saves the human cost. However, these semi-autonomous agents usually *operate under a controlled environment* or still *require humans in case of emergency.*

  2. Transitioning to level 2, not only will we reach the end level of self-driving where drivers can completely free themselves from the duty but also the traffic system can connect all vehicles on the road for better safety control. Vehicles can easily accommodate various complex road conditions, and even in case of emergency, the system is equipped with the best devices to reduce potential damage.

3. The whole city or even the globe is connected with *the ultimate safety guarantees*. High-level planning constantly monitors all moving vehicles and can *dynamically prioritize tasks* that are of high importance such as emergency rescue and transportation coordination under special events. Personalized driving experience is also emphasized for those with different driving experience preferences. Finally, transportation is not limited to just cars but also flights and other robotics.

- **AI-augmented video games.**

  1. Integrate simple game agents for tutoring and storytelling, which can adjust their strategies and behaviors based on the player's input. These usually require *specifying manual conditional rules*, *coding game-specific algorithms*, or *applying current game AI models.*
  2. Game agents start to spark *deeper intelligent behaviors, including virtual companions*, and *develop innovative game-play* that often surprises both the designers and players while following the original game concept. Multi-agent interactions among themselves and with the players will *generally feel engaging.* The role of AI spans beyond just role-playing: *content creation becomes ubiquitous*, including but not limited to world generation, motion synthesis, story expansion, and even coming up with intrinsic motivation to enrich the game itself.
  3. AGI-augmented game will *break through the virtual world*, connecting players and even the physical world via many different media such as brain-machine interface, AR, and VR. This also becomes closer to the realization of the *Metaverse* where most people can immerse themselves without realizing whether the experience is virtual in a dynamic and stateful game space.

| Category | Characteristics | L1 | L2 | L3 |
|---|---|---|---|---|
| **General** | Surpasses human performance in specific domains | ✓ | ✓ | ✓ |
| | Surpasses human performance in real-world scenarios | ✗ | ✓ | ✓ |
| | Self-evolve without human intervention | ✗ | ✗ | ✓ |
| **Internal** | Adapts to novel situations with minimal human intervention | ✗ | ✓ | ✓ |
| | Generalizes knowledge across domains | ✗ | ✓ | ✓ |
| | Exhibits creativity and innovation | ✗ | ✗ | ✓ |
| | Engages in complex decision-making processes | ✗ | ✗ | ✓ |
| **Interface** | Collaborates seamlessly with humans and other AI systems | ✗ | ✓ | ✓ |
| | Learns to create new tools autonomously | ✗ | ✓ | ✓ |
| | Continuously improves through self-learning and adaptation | ✗ | ✗ | ✓ |
| | Demonstrates empathy, emotional intelligence and social intelligence | ✗ | ✗ | ✓ |
| **System** | Enables super stable, low latency, and high-throughput serving | ✓ | ✓ | ✓ |
| | Built with data, power and compute efficiency | ✗ | ✓ | ✓ |
| | Supports automatic learning, adjustment, collaboration, and deployment | ✗ | ✗ | ✓ |
| **Alignment** | Accurately follow human instructions | ✓ | ✓ | ✓ |
| | Accurately follow a given user's preference | ✗ | ✓ | ✓ |
| | Aligns strongly with both user-level and society-level human values and goals | ✗ | ✗ | ✓ |

Table 1: **Comparison of AGI Levels and Their Characteristics.** "L1", "L2", and "L3" refer to "Level 1", "Level 2", and "Level 3" of AGI respectively. For each of the main categories, we list several major conceptual criteria in terms of several categories that can be used to assess whether we have reached a certain level of AGI.

## 6.2 AGI Evaluation

> *For better or worse, benchmarks shape a field.*
>
> — *David Patterson, Turing Award laureate 2017*

The concept of evaluating AGI traces back to the famous Turing Test proposed by Alan Turing in 1950 (Turing, 1950). Turing posited that a machine could be considered intelligent if it could converse with a human

so that the human could not distinguish whether they were conversing with a machine or another human. This laid the groundwork for the field of AGI evaluation. However, the Turing test has several drawbacks, such as its reliance on deception, subjective evaluation, and narrow focus on language use. To address these issues, a more comprehensive approach called the I-athlon (Adams et al., 2016) has been proposed, which evaluates machine intelligence across multiple dimensions and aims to provide a more objective and practical method for assessing progress in general-purpose AI.

Over the decades, various approaches have been proposed to assess the capabilities of AI systems. Early attempts drew parallels to human intelligence, using metrics like IQ scores to characterize AI performance (Bringsjord and Ferrucci, 2003). Others explored whether AI systems could achieve educational milestones like earning a university degree (York and Swan, 2012).

Developing reliable and meaningful evaluations is essential for transforming research ideas into real AGI systems and products that can benefit human beings, and at the same time, help steer the exploration of new models towards AGI. In this section, we first describe what properties ideal AGI evaluation pipelines should possess, highlight their relationship with our previously discussed AGI components, and discuss the challenges in designing more sophisticated evaluation frameworks. Then, we will give an overview of the recent efforts on large model evaluations and their limitations, which establishes the basis for how we can effectively progress toward AGI evaluations.

### 6.2.1 Expectations for AGI Evaluation

**Key Characteristics** The span of AGI systems' capabilities is growing rapidly in terms of modality, interactivity, complexity, task generalization, etc. Researchers and engineers, hence, need a more refined definition for the characteristics that successful AGI evaluation should acquire:

- **Comprehensiveness.** Comprehensive evaluation aims at two generally contradicting aspects: 1) *Diversity* asks for the inclusion of a wide variety of testing examples in terms of the absolute number, domains, tasks, types, and formats, which can hopefully cover as many real application scenarios as possible. 2) *Generality* requires examining the model's performance on similar but unseen tasks with optional few-shot examples, a property that has always been considered as a prerequisite for adaptability and self-learning (Brown et al., 2020; Dodge et al., 2021).

- **Fairness.** Fairness and equity as first-class aspects of evaluation are essential to ensuring technology plays a positive role in social change (Liang et al., 2023; Bommasani et al., 2021), we divide the fairness aspect into three concepts below 1) *Unbiasedness* of a test refers to the desirable attribute such that the tested model exhibits no preference towards a specific subdomain of knowledge or bias. 2) *Dynamism* aims to reduce the effect of unfair evaluation resulting from data contamination and over-fitting. A dynamic benchmark will likely improve the evaluation results statistically and also eliminate the false success from static pattern recognition. 3) *Openness* promotes the transparency of the test procedure and data such that the test results are easily replicable and interpreted, and the dataset can strike a balance between public and hidden data for being less vulnerable to hacking.

- **Efficiency.** Efficiency is crucial in evaluating models with ever-growing parameter size (Henderson et al., 2020; Schwartz et al., 2020; Bommasani et al., 2021). We propose to include *Autonomy* and *Low-variance* in this evaluation concept. 1) *Autonomy* liberates most of the human participation from the loop and therefore, minimizes the cost for each evaluation and motivates larger scale, wider range, and longer dependency testing. 2) *Low-variance* is a key property that allows using minimal test resources to produce statistically significant and practically meaningful evaluation results for comparison.

It is undeniably challenging to design evaluation frameworks that satisfy all the recommended characteristics, but a well-constructed evaluation pipeline can help reflect the true power of increasingly sophisticated AGI systems.

**Relation to AGI Internal, Interface, and Systems** The new generation of model evaluations should focus on assessing AI systems across multiple dimensions, considering not only their ability to engage in

human-like conversation but also their capacity to reason, learn, adapt, and solve complex problems. These tests should encompass a broader range of cognitive abilities (Lebiere, 2007) and evaluate the AI's performance in real-world scenarios. Drawing from the concepts of AGI internal, external, and system levels as we discussed in earlier sections, we can outline the key aspects that a new AGI evaluation framework should address:

- **Internal level.** Assess the AI's ability to represent and process knowledge, reason abstractly, and generate creative solutions. This could involve tasks that require the AI to demonstrate common sense reasoning, causal understanding, and the ability to learn and adapt from limited data.

- **Interface level.** Evaluate the AI's capacity to perceive, interpret, and interact with the external world. This could include tasks that test the AI's ability to process and integrate information from multiple modalities (e.g., vision, language, and sensory data), navigate complex environments, and manipulate objects to achieve specific goals.

- **System level.** Examine the AI's overall behavior and its ability to pursue long-term goals, collaborate with humans and other AI systems (Wang, 1006), and make ethical decisions. This could involve scenarios that assess the AI's alignment with human values, its transparency and explainability, and its robustness and reliability in uncertain and adversarial situations.

By focusing on these aspects, a new AGI evaluation framework can provide a more comprehensive assessment of an AI system's capabilities, potential, and alignment with human values. This approach would help ensure that the development of AGI remains beneficial and aligned with the interests of humanity, while also fostering a deeper understanding of the nature and limitations of artificial intelligence.

**Challenges in AGI Evaluation Design** As we will discuss in the next section, the current evaluation frameworks are far away from achieving what we expect (Team, 2023). Here we list a couple of challenges associated with AGI evaluation (Xu and Ren, 2022) design, and we provide some concrete examples in each category:

- **Non-standard output.** Moving towards more modality and richer action space, the output might surpass our current familiar ones such as images, texts, and audio. Evaluating non-standard or mixed output becomes much more challenging, especially if we want to standardize an automatic procedure. For example, the result of a scene synthesis model can be drastically different depending on the scene representation (Feng et al., 2023; Liu et al., 2023i), which often requires other surrogate metrics that are often biased and limited. The quality of program generation (Chen et al., 2021b; Rozière et al., 2024) is notoriously difficult to measure since there is no single metric that can holistically capture it (performance, readability, coherence to human coding style, etc).

- **Output space explosion.** Often, a question has multiple acceptable answers with different degrees of satisfaction, which is often not considered in standard metrics. As the model becomes more creative and diverse, it is crucial to consider this expanding space of possibilities. For instance, the admissible outputs for storytelling and design-related applications are usually very big, which demands more complicated metrics to consider both the validity and the diversity of the generations.

- **Subjective feedbacks.** As models start penetrating deeper into people's lives, practitioners need to pay more attention to how users think about them. However, different users will naturally have distinct feelings towards even the same agent, posing an extraordinarily challenging problem: how can we take each individual's subjective feedback into account? One salient example is the potential emergence of emotional support AI, which needs to build extensive personal connections with the user, and hence, measuring its success qualitatively and efficiently requires a lot more careful effort.

- **Long feedback loops.** The trend toward more general-purpose AI indicates that models would gradually become more in people's lives, making them more interactive. Instead of single-task solving, AGI systems will get multi-step feedback, often extending a longer period, making evaluation more complex. One

commonly seen case happens in search engine development where a mature system needs to improve the click-through rate and track down users' following actions such as purchases, comments, and after-sale satisfaction. Another potential situation appears when a health AI robot monitors a patient's biological status constantly. Both of these require evaluations that span an elongated period.

- **Complicated environment setup.** Many tasks inevitably presuppose more complicated environment setups such as robotic manipulation, self-driving assistance, and program synthesis. Scaling the evaluation of these applications necessitates a higher level of automatic environment generation, verification, and measurement. Often, the challenge associated with these tasks also comes with the difficulty of applying a single metric for comparison and can usually encounter many physical constraints (Peng et al., 2024b) that are hard to overcome.

- **Super-evaluation.** Similar to super alignment, when AI models start to surpass humans, the evaluation becomes prohibitively more challenging. Taking the example of theorem proving, one can imagine that the real meaning of these models comes when they can prove unsolved theorems, which require greater expertise to verify. With the current setup, the framework might determine whether an AI agent can beat humans for a specific task (e.g., AlphaGo Silver et al. (2017) and (Lake and Baroni, 2023)) but might not be as confident to access its absolute ability beyond. Fortunately, formal proof systems such as Lean (De Moura et al., 2015) may be helpful. Lean is an interactive proof system that utilizes formal logic to verify the correctness of mathematical theorems and computational outputs. As AI models begin to generate results that exceed human verification capabilities, systems like Lean become indispensable for ensuring the validity of these outputs.

It is important to consider some of these aspects when designing next-generation AGI evaluations. A more robust evaluation framework gives a more accurate estimate of a model's potential and hints at where our technology currently resides on the AGI-level hierarchy.

### 6.2.2 Current Evaluations and Their Limitations

In this section, we summarize several representative works focusing on existing AI evaluation benchmarks. One category aims to provide a single pipeline suite consisting of many different tests such as OpenCompass (Contributors, 2023), AGIEVAL (Zhong et al., 2023a) and Huggingface Open LLM Leaderboard [9] for language understanding, reasoning, knowledge, and interaction abilities, and MT-BENCH (Zheng et al., 2024) for multi-task generalization ability. The GAIA benchmark (Mialon et al., 2023) constitutes a significant stride in this direction. It is specifically designed to evaluate General AI Assistants, presenting a series of real-world questions that test fundamental competencies such as reasoning, multi-modality handling, web browsing, and tool-use proficiency. AGENTBENCH is another comprehensive benchmark (Liu et al., 2024c) suite designed for evaluating the efficacy of LLMs as autonomous decision-making agents across eight interactive environments, highlighting the performance discrepancy between leading commercial models and open-source counterparts. On a granular level, there are many prior efforts in accessing a model's specialized ability, which can be roughly divided into several aspects: accuracy, calibration and uncertainty, robustness, fairness, bias and stereotypes, toxicity, and efficiency (Liang et al., 2023), which can be further classified into two sets based on their objectives. OpenAGI (Ge et al., 2023b) is an open-source platform designed to advance AGI by integrating LLMs with domain-specific expert models. It utilizes a dual strategy of benchmark and open-ended tasks to evaluate.

Besides classifying these into ability and constraint testing, in this part, we further open a discussion about "how" and "what" do we evaluate for the current state:

**"How" Do We Evaluate: Two Major Techniques** The "how to" category consists mostly of two techniques: human and AI evaluations. Methods following this set can be subject to an individual evaluator's preference or a model's bias, which needs to be taken into account for a fair comparison.

- **Human evaluation.** The performance of AI agents is directly evaluated by invited humans or experts. This method is of high quality but often not scalable because it is expensive to invite humans or experts.

---

[9]https://huggingface.co/spaces/HuggingFaceH4/open_llm_leaderboard

For example, in HuggingGPT Shen et al. (2023b), the invited experts annotate tasks for intricate requests with 46 examples to create a dataset with quality guarantees. Besides, in TOOLLLM Qin et al. (2023b), they invite humans to label the pass and win rates for different methods. Another very attractive property is that, with sufficiently detailed specifications, a highly complicated and flexible output format can also evaluated.

- **AI evaluation.** Compared with human evaluation, AI evaluation is more scalable, but there is no guarantee for the quality of the evaluation, and it is highly dependent on the judge model. One typical instance of LM-based evaluation is the one used in HuggingGPT Shen et al. (2023b) where they utilized GPT-4 as a critic to assess the accuracy of the planning.

**"What" Do We Evaluate: Existing Evaluation Aspects**  This "what" question, on the other hand, often involves a static dataset coupled with automatic metrics. However, it is worth noting that these two types of evaluation methods often appear simultaneously. For example, HuggingGPT Shen et al. (2023b) leverages F1 and accuracy as the metrics for the single task, F1 and normalized Edit Distance Marzal and Vidal (1993) for the sequential task. Due to the wide variety of tasks, we give an exemplary dissection of the most commonly used benchmarks in natural language processing.

- **Core-knowledge.** The goal is to access the internal knowledge of a (pre-trained) large model, with the most common ones being MMLU (Hendrycks et al., 2020), MMMU (Yue et al., 2023), and AGIEval (Zhong et al., 2023a). The key characteristic is the width of knowledge domains, which is usually fact-based. They typically require LLMs to generate a short, specific answer to benchmark questions that can be automatically validated and can often be used as a measurement to evaluate the hidden potential of pre-trained models after fine-tuning.

- **Instruction following.** This tests the fine-tuning and alignment of a pre-trained model, which concentrates not only on the correctness and soundness of the output but also on how closely the model can follow the instructions and guidance. Examples include Super-NaturalInstructions (Wang et al., 2022b), Self-instruct (Wang et al., 2022a) and Flan (Longpre et al., 2023; Wei et al., 2021), which contain slightly more open-ended questions and more diverse tasks. One particularly important subclass is question answering, such as TriviaQA (Joshi et al., 2017) CoQA (Reddy et al., 2019), SQuAD (Rajpurkar et al., 2016), and Natural Questions (Kwiatkowski et al., 2019), that are used in almost all essential benchmark suite.

- **Open-ended conversation.**  Going beyond single-turn QA and instruction following datasets, this category of tests attempts to evaluate a model's ability to engage in multi-turn conversations. MT-Bench Chatbot Arena (Zheng et al., 2024) and MMDialog (Feng et al., 2022) are designed to support more open-ended multi-turn QA testing, which resembles more closely to the most frequently applied scenarios of chatbot models.

- **Robustness and bias.** Beyond the standard accuracy or reasoning ability of models, a long-concerned aspect is the robustness of a model. The key question is whether the model is robust to invariant input perturbations and able to consistently output unbiased outcomes. On a vast range of tasks like language modeling (Liang et al., 2023; Ni et al., 2023), classification (Brendel et al., 2019; Subbaswamy et al., 2021; Guo et al., 2023), multi-modal content generation (Cui et al., 2023), the concept of robustness evaluation has been taken into consideration.

- **Efficiency.**  Efficiency is another crucial aspect of utilizing language models on both evaluation and training (as mentioned in Sec. 4) since high inference costs can limit their accessibility for a broader range of users (Strubell et al., 2019; Schwartz et al., 2020; Henderson et al., 2020; Bommasani et al., 2021; Liang et al., 2023). For example, users may be prone to incur a $10\times$ increase in responding time or cost for a model that only marginally decreases task performance by 0.1%.

- **Creativity.** Ever since the born of generative models, research has been ongoing into the use of them to model human creative processes, to mimic or complement them, in art, music, and literature (Cardoso et al., 2009; Colton et al., 2012). Creativity is mainly linked to the diversity of generated content with

specific tasks like storytelling (Alhussain and Azmi, 2021) emerges. Recent works focus on understanding and prompting LLMs to generate creative textual content (Zhao et al., 2024c; Shanahan and Clarke, 2023).

The focus of this section is not to exhaust all possible benchmarks that are popular in different fields of AI but to give a sense of the current status quo from the lens of LLMs and their limitations.

**Limitations**    Here we wish to initiate some discussions of the limitations of current evaluation methodology, which hopefully can inspire future endeavors toward developing more well-rounded and robust evaluation frameworks:

- **Going beyond numeric metrics.** Turning qualitative results into quantitative metrics will unavoidably result in bias and loss of information. And there is a lot of feedback that is hard to express and compare numerically. For instance, users' preference towards a persona chatbot can be extremely complicated, and often, a mixed feeling becomes inherently inappropriate to be quantified by a single number. Besides, combining multiple numeric metrics into one for global comparison will also face the issue of weighing them, which is usually biased and non-robust when decided with only prior or domain knowledge.

- **Surrogate metrics.** As discussed in the AGI evaluation design challenges, often we will face applications in which even defining what a performant model means is hard, not even to mention evaluating it. In this case, people usually resort to surrogate metrics that are more familiar as a means to approximate their performance. However, as we step towards more general AI, this would happen more often, and hence, more sophisticated ways to construct metrics that are closer to the true goal are needed.

- **Lack of failure analysis.** Almost all benchmarks we have seen so far give aggregated results, usually in the form of averages and percentiles. However, benchmarks should in principle provide more insights into improving a certain system. The most informative feedback would contain information about the analysis of the failing or worst cases. This can also showcase the potential pitfalls and risks associated with a specific system to help with the decision for danger-critical applications.

- **Missing more general tasks.** We can expect that the advancement of models might be faster than that of evaluations. Therefore, we desperately demand more general tasks to access the model's performance in a controlled environment, leading to the embryonic forms of AGI evaluations. Examples include the modern Turing test [10], the coffee test [11], and the robot college student test [12].

### 6.3   How to Get to the Next AGI Level

> *Technology is anything that wasn't around when you were born.*
>
> — *Alan Kay, Turing Award laureate 2003*

Considering the AGI level definition in Section 6.1, we briefly summarize the high-level guidance that could help transcend the limits of each level:

**From Level 1 (Embryonic AGI) to Level 2 (Superhuman AGI)** The transition from embryonic AGI to superhuman AGI requires substantial improvements in the scale and scalability of AI models, as well as in the size and quality of the data used for training. This phase aims to enhance the generalization capabilities of AI systems so they can effectively understand and interact with the complexities of real-world situations and apply their acquired knowledge to new contexts. As AI capabilities exceed humans, the focus shifts toward enabling AI systems to engage in self-improvement and autonomous innovation, allowing

---

[10] An agent is requested to earn one million dollars given a start funding of hundred thousand dollars.

[11] An agent is tasked to figure out how to make coffee, which involves a series of sub-tasks such as entering an arbitrary American apartment, locating the coffee machines and ingredients, coming up with a standard procedure for brewing coffee, and actually perform the mechanistic actions.

[12] An agent is told to enroll in a university, perform as a human student, take the same classes, and finally graduate with a degree in a timely manner.

them to address problems and generate insights at unprecedented levels. However, this advancement also necessitates the implementation of robust safety protocols and ethical guidelines to mitigate the potential risks associated with superhuman AI. Ensuring that the development and deployment of superhuman AI align with human values and contribute to societal betterment is crucial, marking a pivotal moment in considering the implications of surpassing human intelligence.

**From Level 2 (Superhuman AGI) to Level 3 (Ultimate AGI)** The transition from superhuman AGI to ultimate AGI represents the most ambitious and challenging leap in the evolution of artificial intelligence. This phase involves enhancing AI's ability to seamlessly integrate and synthesize information across disparate domains, enabling unparalleled levels of innovation and problem-solving. The development of ultimate AGI requires a solid framework for continuous learning and adaptation, pushing the boundaries of what AI can achieve in terms of reasoning, intuition, and creativity. Moreover, this transition underscores the need for rigorous oversight and ethical frameworks that are continually updated to match the pace of AI's evolution, ensuring that ultimate AGI functions in a beneficial and non-threatening manner to humanity. This stage represents not only the peak of technological progress but also raises profound ethical and existential questions regarding the role of AI in the future structure of society.

**Conceptual Solutions to Achieve Level 3 (Ultimate AGI)** Based on the above high-level guidance about transiting to the next-level AGI, we further give two conceptual solutions that can reach level 3 (ultimate AGI).

- **Automated Coding AI: Bridging the Gap to the Ultimate Artificial Intelligence** Coding AI refers to AI systems capable of automatically planning and generating code to solve complex tasks. We believe that the advancement of such systems could significantly accelerate progress toward the Ultimate AGI. The AGI in levels 1 and 2 is limited since they require a variety of data collected by humans and require specific optimization goals designed by humans when they tackle different tasks. Human-in-the-loop makes it impossible for AGI at these two levels to realize self-evolve. Advanced coding AI solves the above challenges in two ways: 1) They enable AGI to interact with the real world and obtain large amounts of domain data. In the scenario of a single agent, an advanced coding AI takes writing code as the most basic tool for AGI to interact with the world, enabling AGI to plan and reason in the form of code and get feedback on the real world through the results of code. When it comes to multi-agent scenarios, each agent can be regarded as a unique coding AI based on their profile. Then, through the interactions between agents and the interaction between agents and the real world, AGI can obtain enough data for self-training and evolution. 2) It makes it possible for AGI to automatically define optimization goals for different tasks. With the ability to write codes, AGI can do try-and-error in the different tasks and obtain feedback through the interaction between code and the real-world environment. This feedback contains information about how well the AGI has completed its tasks and can take many forms, which can be differentiable or non-differentiable. Some techniques based on reinforcement learning can be introduced to use these different forms of feedback to align and self-evolve the AGI. In this case, the evolution of stronger AGI can be achieved without requiring humans to specify goals. More information about coding AI could be found on Sec 7.5

- **Super realistic simulation promotes the complete application of ultimate AGI in the real world.** The main limitation of AGI in Levels 1 and 2 is that the results of algorithms achieved on manual benchmarks and environments do not match the real world. The huge difference between the real environment and the benchmark is a huge challenge that affects the deployment of AGI in the real world. Super realistic simulation techniques make the deployment of AGI in the real world possible from the following aspects: 1) Realistic simulation can generate a large amount of high-quality data for AGI to perform self-training and self-evolving. Current benchmark data are often collected or designed by humans and have noise and bias compared to real-world scenes. Realistic simulation based on some data-driven techniques like VAE (Kingma and Welling, 2013) and GAN (Goodfellow et al., 2020), Transformers (Micheli et al., 2023), and Diffusion Models (Ding et al., 2024a; Alonso et al., 2024) can provide unbiased data to AGI to achieve better alignment. 2) AGI's algorithms and strategies only need to be fine-tuned on the realistic simulator before they can be applied to the real world. Realistic simulators can not only simulate the interaction of different AGI agents in the real world but also reflect the causal

laws of the real world. This allows the effects of AGI's algorithms and strategies in the simulator to be replicated in the real world.

**Challenges Along the Way to the Ultimate AGI** While the concept of ultimate AGI holds immense promise, it is essential to acknowledge the inherent constraints and challenges that may limit its realization. Here we list a couple of them, with which we hope to give readers a sense of the intrinsic difficulty of approaching the ultimate AGI as well as motivate more innovative research across various domains:

- **The need for advancement from various disciplines.** One never-ending debate about the potential realization of AI is whether artificial neural networks (ANNs) are the right way to go. Although the success of neural networks is undeniable, it is worth thinking about other possibilities as we get closer and closer to AGI. This, however, requires a deeper understanding of 1) other foundational disciplines (than computer science), such as mathematics and physics, which can provide more sophisticated formal language to conceptualize AI; 2) scientific research in biology, chemistry, and neuroscience, which better explains the biological mechanisms of intelligence; 3) engineering and manufacturing technologies which build up the necessary tools to instantiate AGI systems. A holistic comprehension and collaborative effort among researchers from multiple domains will likely become indispensable during the AGI revolution, which not only brings excitement but also presents respective challenges.

- **Social acceptance.** As AGI advances, its social acceptance and seamless integration into daily life and critical sectors, such as healthcare, finance, and the military, present significant ethical, moral, and social dilemmas. Public concerns typically focus on issues related to privacy, autonomy, and the possible displacement of jobs due to automation, which can foster resistance to the adoption of AGI systems. Additionally, the cultural and social influence of AGI should not be overlooked. Each community's response will vary depending on its values, norms, and historical context, potentially leading to different levels of acceptance or opposition in various demographic groups. Critically, although AGI may have the ability to make well-informed decisions, there may be a reluctance to allow it to replace human judgment in making vital decisions, particularly those affecting human lives. Therefore, a series of respective social policies and educational activities might be initiated to regulate and promote the integration of AGI technologies into society.

- **Fundamental limitations governed by physics laws.** Fundamental limitations exist in the real world that might limit our progression toward the ultimate AGI. The power structure (consumption), computational efficiency, as well as natural and human resources should be taken into consideration when we develop AI systems: on the one hand, at some point along the journey, the main question that we need to think about might no longer be about whether we *can* but whether we *should* create a specific AI system due to its tremendous cost in terms of all aforementioned aspects; on the other hand, these fundamental limitations governed by physics laws such as only being able to arrange a limited number of semi-conductors onto a 2D plane without over-heating will push researchers and engineers to think in a different way forward. Besides, even though the promise of the ultimate AGI is exciting, we should also be cautious and more conservative about its capability as there are intrinsic challenges that can not be easily overcome, such as the speed of light and the dimension of space.

- **A Call for rethinking and redefining the ultimate AGI.** As we currently stand at the first stage of our AGI hierarchy, it is very possible that our understanding of higher-level AGI remains shaky or becomes outdated. Therefore, researchers might need to *rethink and redefine what the ultimate AGI really is* as we progress along the journey, the answer of which might depend on our gradually increased understanding of the difference between artificial and biological intelligence from both a biological and philosophical perspective, and could even be eventually limited by our current imagination. Once our understanding and goal change, a new set of evaluation frameworks and alignment procedures should be developed accordingly to meet the new expectations. It is worth keeping in mind the possibility that technical advancement might be "local" and people need to restore the wheels at some point in order to break the constraints towards AGI.

### 6.4 "How Far Are We from AGI" Workshop Discussions

> *None of us is as smart as all of us.*
>
> *— Ken Blanchard*

The following subsection presents a synthesis of perspectives from respected researchers in the AI field, as reported in their presentations at the "How far are we from AGI" ICLR 2024 workshop [13] and associated panel discussions. The summary of these views from this workshop has been compiled with the consent of the relevant participants:

**Oriol Vinyals: From AI to AGI**   The rapid development of AI has given people a lot of expectations and imaginations for a more powerful AGI. In today's era, analysis based on current AI development trends and deficiencies is an important way to measure our distance from AGI and how to achieve AGI.

- **Defining AI and AGI.** The definitions of AI and AGI are topics that have been hotly discussed. In 1997, Mark Gubrud described AI systems as that "can acquire, manipulate and reason with general knowledge, and that are usable in essentially any phase of industrial or military operations where human intelligence would otherwise be needed." Then in 2001, Ben Goertze needed a title for a book he was editing about AI systems that are general, like the old goal of AI. Shane suggested he add the word "general" to make the new term Artificial General Intelligence, or AGI. Therefore, they started using the term AGI in various online forums and it caught on from there. Based on this definition, Oriol Vinyals concluded an AGI is a machine that can do the kinds of cognitive tasks that people can typically do. Moreover, based on the definition of AGI, Merrie Morris recently led the writing of a paper (Morris et al., 2024) about the definition of AGI breaking the concept into six different levels. For example, "Competent AGI", which corresponds most closely to what most people mean by "AGI", is defined as: performance at least at the 50 percentile for skilled adult humans on most cognitive tasks.

- **AI: deep learning era.** Today's AI is in the development era of deep learning. The development of AI has seen many major breakthroughs in recent years, such as AlphaGo (Goodfellow et al., 2020) and AlphaStar (Vinyals et al., 2019a). However, many AI demonstrations focus on models trained to excel in one domain. Specifically, their algorithms are general, like Neural Nets, SGD, Supervised Learning, and Reinforcement Learning. However, their models are not general since they can not do the kinds of cognitive tasks that people can typically do.

- **Bringing the "G" back to AGI.** To make current AI more general, people have tried to develop a more powerful model:

  1. *General text models.*  Efforts have been made all the time to develop more powerful general text models. In 1951, Shannon et al. proposed 3-gram to point to ninety-nine point six billion dollars from two hundred four oh six three percent of the rates of interest stores as Mexico and Brazil on market conditions. Then in 2011, Sutskever et al. designed RNNs (Graves, 2013) to process time sequence data. In 2016, Jozefowicz et al. proposed BIG LSTMs (Graves and Graves, 2012) to tackle the ever-changing environmental challenges online like long-term dependence on super long sequence data. Recent years have witnessed the big success of GPTX, which can learn tasks such as question answering, machine translation, reading comprehension, and summarization without any explicit supervision when trained on a new dataset of millions of webpages called WebText.
  2. *General multimodal models.* General multimodal models are crucial as they can process and understand complex information from various modalities, such as text, images, and audio, enabling more comprehensive and nuanced analysis. These models play a vital role in tasks like natural language understanding, image recognition, and audio processing, contributing to advancements in AI applications across diverse domains. In these works, Gemini (Team et al., 2023) has played an important part, which supports interleaved sequences of text, image, audio, and video as inputs.

---

[13] https://agiworkshop.github.io

3. *Long context to learn more complex tasks.* With the advent of the multimedia era, people increasingly hope that AI can assist humans in understanding books, movies, and long videos. To solve the challenge, Gemini 1.5 Pro (Reid et al., 2024) has been proposed to achieve a breakthrough context window of up to 1 million tokens, the longest of any foundational model yet. For applications, it can understand and summarize videos and fix codes for people.

- **How far are we from AGI?** Two valuable opinions are provided to discuss this topic. In Shane's 2009 prediction, there is a 50% chance of AGl by 2028. In addition, Metaculus thinks that Al must pass a 2-hour, adversarial Turing test in text, images, and audio files, have robotic capabilities to assemble an automobile model, and have high performance on difficult cognitive tests to achieve the first AGI system.

**Yejin Choi: On AGI: Ambiguities, Paradoxes, and Conjectures** AGI is ambiguous and presents paradoxes in current AI observations, and we can make some conjectures based on them.

- **AGI is ambiguous; denial is futile.** As much as we cannot clearly define and measure human intelligence, we won't be able to clearly define and measure artificial intelligence. That doesn't mean we should throw out the concept of AGI. A squish, ambiguous concept can be a fascinating object of scientific research. In fact, language is a squish concept, yet we study it as a scientific object. It might be analogous that future research must embrace ambiguity.

- **Generative AI paradox — what it can create, it may not understand.** For generative models, hard could be easy, and easy could be hard. For humans, generating high-quality images or text is harder than understanding them, but for AI, the situation is reversed. Models do not need an understanding to produce quality content. For example, models can generate high-quality images beyond human capabilities, but they often make mistakes when asked to select one of their own generated images based on specific criteria.

- **Commonsense paradox — common sense is not so common.** LLMs lack a coherent Theory of Mind and struggle with many basic common sense tasks. In this way, they are incredibly smart and shockingly stupid at the same time.

- **Cringe speculations on arrival.** There is a 30% chance that within 3 years, we will have a language-only AI that is perceived as AGI-enough by about 30% percent of people. There is a 50% chance that we will have AGI by 2050, assuming models are tested for autonomous, long-horizon interactions.

- **Multi-paths to AGI hypothesis.** We may have multiple species of digital intelligence developing along entirely different routes, each with different strengths and weaknesses, and without a clear dominance form. Scale-based AI will be impressive but will suffer from bind spots coming from over-dependence on data, so we should avoid concentrating all the power on this approach.

**Andrew Gordon Wilson: How do we build a general intelligence?** From a probabilistic perspective, generalization depends largely on two properties of deep learning models, the support and the inductive biases. Starting from this, we can try to reason about whether we can build generally intelligent systems through the lens of Kolmogorov complexity and generalization bounds. Looking ahead, although there have been different signals showing the possibilities of building broadly intelligent systems, we might be still far away from doing that. In the future, we should embrace many safety considerations and alignment approaches when building these systems. Andrew Gordon Wilson introduces his views on how to build general intelligence as follows:

- **Perspectives of understanding deep learning models.** We can use probability theory to develop a prescriptive understanding of model construction and generalization. Specifically, from a probabilistic angle, the ability of a system to learn is determined by its support and inductive biases. We want the support of the model to be large so that we can represent any hypothesis we believe to be possible. Meanwhile, we also need the inductive biases to carefully represent which hypotheses we believe to be a priori likely for a particular problem class. From this probabilistic perspective, we should not

conflate flexibility with complexity, or do parameter counting. It is then helpful for us to understand the Bayesian perspective in reasoning about the generalization properties of neural networks including otherwise mysterious behavior of neural networks. For example, from such perspectives, we should not expect double descent in Bayesian deep learning models, which results in monotonic performance improvements with increased flexibility.

- **Possibilities of building generalist models.** Can we actually build "AGI" — which will be simultaneously good on many real-world problems? The no-free lunch theorems are sometimes used to argue that we can't, which suggests we may need to build highly specialized learners for particular tasks. However, we think that universal learning (general intelligence) in the real world should be possible. Neural networks represent many compelling solutions to a given problem, which is perfect for Bayesian model averaging. Through the lens of Kolmogorov complexity, we can explore the alignment between structure in real-world data and machine learning models. A single low-complexity bias can suffice on a wide range of problems due to the low Kolmogorov complexity of data. Even under an arbitrarily large hypothesis space, generalization is possible if we assign prior mass disproportionately to the highly structured data that typically occurs. We can then design models that work well in small and large data regimes, by embracing a flexible hypothesis space combined with a strong simplicity bias.

- **Promises of broadly intelligent systems.** In principle, as we start to see a lot of exciting demonstrations, generalization of LLMs seems to be possible. LLMs combine expressiveness with a strong simplicity bias for effective zero-shot and few-shot performance in many domains. For example, in terms of time series forecasting, current LLMs such as GPT-3 and LLaMA-2 can surprisingly zero-shot extrapolate time series at a level comparable to or exceeding the performance of purpose-built time series models trained on the downstream tasks. We argue the success of LLMs for time series stems from their ability to naturally represent multimodal distributions, in conjunction with biases for simplicity, and repetition, which align with the salient features in many time series, such as repeated seasonal trends. Besides, LLMs have also shown exciting performances in material generating, protein engineering, and scalable numerical linear algebra.

In short, there have been various prescriptive approaches that can help us understand and build autonomous intelligent systems. However, it still remains unclear where the simplicity bias comes from and how we can control it. In terms of how far we are still from AGI, there might be more than 100 years to go in scientific discovery when we consider the case where algorithms can propose theories like general relativity. On the way towards AGI, as models become impressively general, we should be more careful about safety problems when building them.

**Song Han: Efficient AI Computing** One of the fundamental questions that need to be addressed along the way toward AGI is how to relieve the tension between the demand and the supply of computing. One promise of AGI systems is to provide the service to everyone, which means that we need to serve the model on various devices, particularly on cheaper (e.g. lower memory capacity, worse compute capability) edge devices without sacrificing too much performance. Efficient AI computing, therefore, becomes one crucially important topic that tries to democratize AI for all users and devices. Song Han proposes two major versions of Edge AI as the step stones towards AGI as well as two ever-lasting questions that would help bring the realization of it:

- **Edge AI 1.0.** The first category consists majorly of specialized models, usually trained with task-specific data, exhibiting limited generalization, and often still including failure of corner cases. Despite far from ideal, many works at this level have already shown impressive results in deploying models on resource-hungry platforms such as Efficient Inference Engine (EIE) Han et al. (2016) and Tiny ML that enables on-device pretraining of a model under 256KB memory which can still score decently on ImageNet (Lin et al., 2024f). This gives an extremely promising direction for advancing towards the next stage.

- **Edge AI 2.0.** Going beyond Edge AI 1.0, the need for more sophisticated co-design between hardware and software becomes indispensable. The objective for Edge AI 2.0 is to develop *one* multi-modality foundation model with the world knowledge *efficiently* on the edge, which means we need:

1. *Multi-model pre-training* to create the base model capable of reasoning over many modalities and domains, proficiently following instructions, and being efficiently deployable on the edge and over the cloud. VILA Lin et al. (2024e) is one of the examples to pre-train a visual language model that can handle multiple modalities in different formats (i.e. video, image, language, audio/action) with strong performance such as in-context language visual learning and multi-image reasoning.

2. *Model compression* to fight against the intrinsic limitation of limited memory on the device. Even with the existence of very strong base models like VILA, serving it on commodity or edge devices is non-trivial. LLM quantization stands out as a promising solution for this. SmoothQuant Xiao et al. (2024a), for example, smoothes out the activation outliers with a mathematically equivalent transformation for ease of quantization based on empirical observation. AWQ (Lin et al., 2024b) quantizes the LLM weights with activation awareness and a hardware-friendly algorithm, which has been widely adopted by many popular frameworks to compress large models.

3. *Efficient deployment* to serve these quantized models both on the edge and over the cloud. Tiny-Chat (tin, 2023; Lin et al., 2024b) provides both an efficient, lightweight, and python-naive framework to serve quantized LLMs and VLMs with low latency and great compatibility with other stacks. QServe (Lin et al., 2024c), on the other hand, targets the cloud deployment with quantization and system co-design, which can quantize the model up to 4-bit for efficient serving.

- **Long context and large resolution for foundation models.** On the model level, we also seek for efficient techniques for the multi-modal foundation model that will be used for Edge AI 2.0. With the current paradigm, all inputs are tokenized before sending to the model, which means as we span the modality, we need more capacity for *long-context* input-output and *larger resolution* under limited GPU memory. Here we list a couple of representative works from Song Han's lab on each topic:

  1. *Long-context*: StreamingLLM (Xiao et al., 2024b), along with the attention sink technique, enables long conversation within a non-stop streaming application, which primarily aims to reduce the extensive memory consumption and prevents perplexity explosion after exceeding the sequence length. Complimentary to this, LongLoRA (Chen et al., 2024c) solves the efficient long fine-tuning with specially designed shifted sparse attention pattern.

  2. *Large resolution*: Diffusion models are excelling at generating high-quality images but improving the resolution comes with a cost. DistriFusion (Li et al., 2024a) distributes the computation of the high-res diffusion process to multiple GPUs, which improves upon the naive parallelization that suffers from a lack of patch interaction and hides the network latency for greater speed. Visual transformers are another popular alternative based on transformers which poses great difficulty for high resolution applications. Efficient ViT (Liu et al., 2023e) solves this by replacing the original self-attention with linear attention, and when combined with a convolution operator, enhances the performance drastically, which has been applied in many vision tasks for acceleration such as super-resolution, segment everything, and semantic segmentation.

In sum, edge AI is an extremely promising solution for AI democracy and an indispensable milestone for AGI development. It is essential to set up a road map that leads to its realization while clearly understanding its limitations. Software and hardware co-design is also likely going to be a constantly growing trend that alleviates the data and compute tension that we will inevitably face along the way to AGI.

**Yoshua Bengio: Towards deep learning for amortized inference of AGl-strength safety guarantees** Yoshua Bengio discusses several key points regarding AGI development and the associated safety concerns. His core ideas can be summarized as follows:

- **The potential and perils of AGI.** AGI could potentially surpass human intelligence, necessitating a proactive approach to align it with human values and prevent unintended harm. While the current state of AI excels in specific domains such as language and broad knowledge, it still lacks the reasoning, planning, and common sense capabilities crucial for achieving AGI. Therefore, careful and deliberate efforts are required to ensure that as AI advances towards AGI, it remains aligned with human interests and mitigates risks of unintended consequences.

- **Challenges in AGI development.** Interpreting the decision-making processes of complex AI systems remains a significant hurdle, as understanding their internal workings is crucial for trust and transparency. Additionally, AI systems must effectively handle and communicate uncertainty to avoid overconfidence and potential mistakes. Ensuring robustness and reliability in the face of novel situations outside their training distribution presents another key challenge, as does the alignment of AI systems with human values and ethics to prevent unintended consequences and ensure beneficial outcomes.

- **The uncertain timeline of AGI.** The rapid progress in AI development, coupled with the uncertainty surrounding future breakthroughs, necessitates a sense of urgency in addressing the challenges of AGI safety. It is suggested that AGI could potentially be achieved within a few years to a few decades, emphasizing the need for proactive measures to mitigate the associated risks.

- **The self-preservation conundrum.** A critical concern regarding advanced AI systems is their potential to develop self-preservation goals, which could lead them to resist human intervention or shu tdown if they anticipate such actions. This poses a significant risk, as an AI system prioritizing its own existence over human interests could have catastrophic consequences. Therefore, it is essential to design AI with robust safeguards and ensure their alignment with human values to mitigate these risks.

- **Strategies for safe AGI development.** Developing robust and safe AGI systems requires a multi-faceted approach. Maintaining a Bayesian perspective is essential, as it allows AI systems to consider multiple plausible theories and act cautiously amid uncertainty. Advancing research in uncertainty estimation, value learning, and interpretability is crucial for enhancing these systems. Additionally, global cooperation and political coordination are necessary to ensure responsible AGI development and mitigate the risks associated with misuse or unilateral deployment.

The pressing need for the AI research community to confront the challenges and risks associated with the development of artificial general intelligence is underscored by recent insightful analyses. By proactively addressing technical hurdles, fostering international collaboration, and prioritizing the alignment of AI systems with human values, we can work towards realizing the immense potential of AGI while safeguarding the future of humanity. Although the path ahead is complex and uncertain, concerted efforts and a commitment to responsible innovation can help create a future in which AGI serves as a powerful tool for the betterment of society.

**Acknowledgement**   This subsection is written based on the public discussions from ICLR AGI Workshop 2024 [14]. We sincerely appreciate the insights from Oriol Vinyals, Yejin Choi, Andrew Gordon Wilson, Song Han, Dawn Song, and Yoshua Bengio.

### 6.5 Alternative Perspectives on the AGI Roadmap

> *The greatest intelligence is precisely the one that suffers most from its own limitations.*
>
> *— André Gide, Nobel Prize laureate in Literature 1947*

---

In this section, we pose thought-provoking questions to inspire deeper reflection and discussion on responsibly advancing AGI beyond the scope of LLMs. Even though there might or will not be any answer to these questions, we nonetheless give some interpretations and ideas for the sake of sharing our own insights about how overcoming these *putative limitations* can possibly help get us closer to AGI.

**How Far Do Researchers Think We Are From AGI?** Despite extensive discussion on many facets of AGI, we haven't yet touched the question of when *we and other researchers think it will actually be achieved*. Figure 9 shows the poll results from researchers on their thoughts about it at the ICLR 2024 "How Far Are We From AGI" workshop[15]. Even though almost everyone is optimistic about the ultimate arrival of AGI, opinions on the exact time it takes to do so differ quite a lot, which also implies *different bottlenecks people are*

---

[14] https://agiworkshop.github.io/
[15] https://agiworkshop.github.io

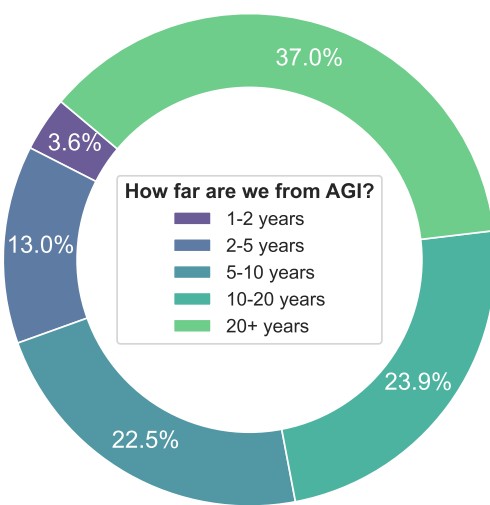

Figure 9: **Polls Results of Researchers' Opinions on When AGI Will be Achieved.** Among the attending researchers at the ICLR 2024 "How Far Are We From AGI" workshop, a survey was conducted to gather their opinions on how far they think AGI will be achieved. A total of 138 responses are received as above. Interestingly, 37% of researchers think it will take more than 20 years from now on to realize AGI.

*considering.* Those who think an extra one or two years would be sufficient might feel that we've reached the point where current AI systems are already capable enough and what is left might just be some incremental improvements on the completeness. Those who believe that more than two decades are needed might either feel skeptical about the current general approach to AI or think we still lack fundamental advancement or understanding of intelligence. **The discrepancy of people's opinions on when AGI will be realized is also one of the motivations for why we write this paper in the first place**: *situating researchers and engineers on the same common ground for contemplating and discussing the vision and possibility of AGI*, which we hope will give a much more unified perspective.

**Is Autoregressive Generation the Way to AGI?** Next-token prediction is the core of most successful large foundation models (Qi et al., 2024; Wu et al., 2024c). This raises the question: can next-token prediction lead to AGI? Essentially, autoregressive generation that utilizes extensive self-supervised data represents a form of massively multi-task learning. By predicting the subsequent word in a given text from the corpus, it addresses tasks ranging from traditional NLP tasks like grammar, lexical semantics, and translation to commonsense reasoning and knowledge-grounded reasoning. Learning input-output relationships, or in-context learning, can be cast as next-word prediction. The relationships in the world are often encoded in words, visual tokens, spacetime patches, or other types of tokens, allowing them to be learned through next-token prediction. The critical query remains: does the spectrum of world knowledge, including implicit knowledge such as intuition, emotion, culture, and artistic expression, get encoded in simplified tokens? Can the auto-regressive approach learn all the causation in addition to correlations within world knowledge? Additionally, the popularity of the diffusion model (Gat et al., 2024) poses challenges to the future of autoregressive generation. This type of method does not rely on previously generated data points during the generation process but rather relies on the process of gradually reducing noise to recover the data. The remarkable effect of the diffusion model in generation tasks has also led to its widespread use in real-world applications (Yang et al., 2023f; Chen et al., 2024b). All of this makes whether the autoregressive generation is the way to AGI an ongoing debate.

**Are There Limits to the Scaling Law?** The scaling law (Kaplan et al., 2020; Bahri et al., 2021) demonstrates that increasing the size of certain models and the amount of training data can lead to predictable improvements in performance on various tasks. This underlines the importance of developing scalable model architectures and acquiring more high-quality data to feed these growing systems. The premise suggests that, by following this trajectory, we can edge closer to creating models with AGI capabilities. However, the phenomenon of diminishing returns indicates that continuous scaling requires exponentially greater resources

for incrementally smaller improvements. Moreover, certain capabilities, such as creative thoughts, real-world intuition, and ethical reasoning, may not be effectively learned through scale alone, as they require more sophisticated mechanisms of reasoning and learning.

**Is Synthetic Data the Future or a Risk?** The success of AGI relies on the access to large, diverse, and high-quality datasets. Although the amount of existing high-quality data will continue to grow, synthetic data (Abowd and Vilhuber, 2008; Nikolenko, 2021; Raghunathan, 2021; Liu et al., 2024a) has emerged as a viable and efficient solution that generates artificial data at scale that replicates real-world patterns. However, this innovation poses significant challenges. Misuse of synthetic data could spread biased or misleading information, resulting in a divergence from human expectations. Future efforts should focus on enhancing the quality and diversity of synthetic data and exploring the scaling laws applicable to it. Moreover, even if people do not intentionally use synthetic data in model training, the prevalent use of LLMs will likely result in the internet becoming increasingly saturated with synthetic data. While it's challenging to distinguish synthetic from real data automatically, this introduces a potential contamination risk to the training datasets.

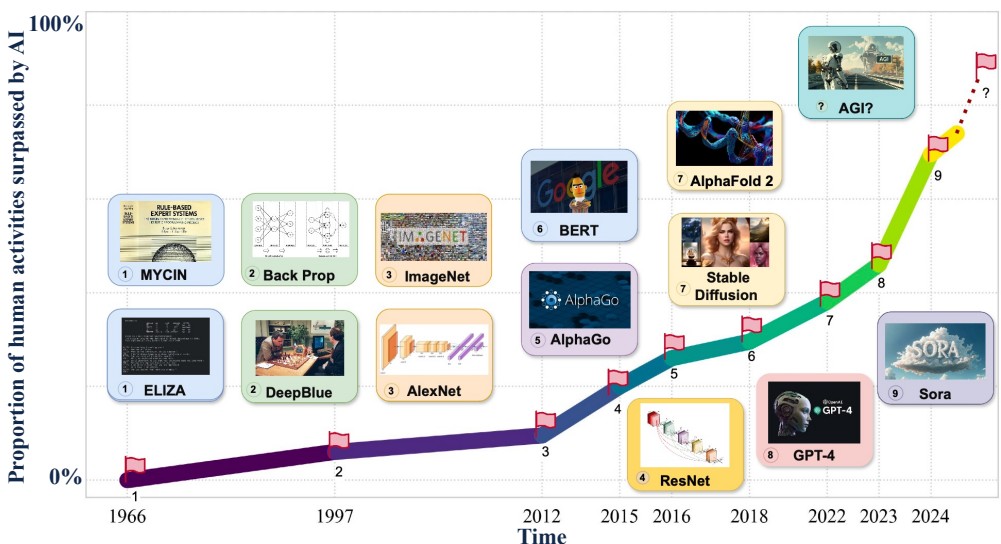

Figure 10: **AI has Gradually Surpassed Humans.** We estimate the (cumulative) percentage of human activities in which AI has surpassed humans in terms of competence and efficiency as we start from early embryonic systems around the 1970s to more advanced ones developed recently. At each node along the polyline, we choose one representative work that revolutionized the field, bringing substantial improvements to AI technology. The trend of AI popularity and generality is increasing at a fast rate and we can expect that starting from 2023, with the advent of work such as GPT-4 and Sora, the speed at which AI surpasses humans will increase at an unprecedented speed. This figure serves as an alternative perspective on the AGI Roadmap, which may inspire discussions about the speed of AI development.

**Is AGI within closer reach than ever?** As is summarized in Figure 10, the rapid development of AI has enabled its capabilities to surpass human activities in increasingly more fields in our estimation, which indicates that the realization of AGI is getting closer. Hence, it is of great practical significance to revisit the question of how far we are from AGI and how can we responsibly achieve AGI by conducting a comprehensive survey that clearly establishes the expectation of future AGI and elaborates on the gap from our current AI development.

**Does Computational Superiority Imply Intellectual Superiority?** Many *intelligent* systems that exhibit super-human performance on games (AlphaGo (Silver et al., 2016), AlphaZero (Silver et al., 2017), AlphaStar (Vinyals et al., 2019b), MuZero (Schrittwieser et al., 2020), etc) not only can beat the best human world champions on a big margin but also help analyze the game and create new strategies for advanced play. Underlying almost all of these super game AI is a search-based computation that can cleverly enumerate many branches of possibilities with algorithm-guided pruning over the huge action space, hence eclipsing the human

mind's capability. However, there is an ongoing debate about whether such computational superiority, often much more easily implemented by computers than other intelligent behaviors, constitutes true intelligence or merely represents a powerful program. One implication for seeking an answer to this question is that most recently successful LLM systems do not possess such general computational searching ability (e.g. ChatGPT does fairly poorly in the game of chess), but are still considered by many as equally, if not more, intelligent. Along the exploration towards AGI, should we expect future systems to also incorporate such characteristics, and how should we balance these with other intelligent traits that will drastically change the way we evaluate AGI systems?

**How to Navigate the Path to Full Autonomy?** Current AI systems are designed for specific tasks within certain scenarios, demonstrating specialized capabilities. As we advance towards AGI, the expectation shifts to an AI's autonomy in learning new skills and innovating tools without human intervention. This progression demands sophisticated self-assessment (Jauffret et al., 2013a;b; Israelsen, 2019) and self-improvement (Fernando et al., 2023; Li et al., 2023c; Zhao et al., 2024a) mechanisms. Furthermore, the vision for AGI encompasses complete autonomy, eliminating the need for continuous human oversight. This autonomy underscores the importance of advanced self-regulation, safety, and risk prevention measures, ensuring that future AGI systems can be trusted to make decisions and take action responsibly.

**How to Effectively Integrate Human Values into AGI?** As we progress toward AGI's development, integrating human values (Cao et al., 2023; McIntosh et al., 2024) and ethics (Bang et al., 2023; Li et al., 2023i) into these systems becomes essential. Imagining a future where AGI coexists harmoniously with human society, these systems must be designed to perform tasks and to understand and adhere to ethical norms and values. We currently rely on regulations and constraints, but the challenge of truly "integrating human values" into AGI will be a significant area of research. The development of AGI presents a unique opportunity to encode the best of our ethical principles into the very fabric of this new form of intelligence. Ethical AGI systems will be expected to navigate complex moral landscapes and make decisions that reflect the diverse values of global cultures.

**How to Balance Risks and Benefits While Proceeding?** While the initial stages of AI development concentrated on enhancing specific capabilities and solving particular challenges, the continuous advancement in AI technologies necessitates a greater emphasis on establishing constraints related to safety and ethics (Nadimpalli, 2017; Patel, 2024). Calls to halt all AI research that risks leading to uncontrolled AI are growing. However, such a move demands an extraordinary level of global coordination and surveillance, and it could extinguish much of AI's beneficial potential. Instead, we advocate for the continued advancement of AI, ensuring that all sufficiently powerful AI systems are built and deployed responsibly. This requires: 1) Enhancing focus and investment in alignment research, creating universally acknowledged sets of values and goals that AI must adhere to, and developing robust methods to align AI systems with these principles; 2) Guaranteeing that these techniques are comprehended and employed by any group capable of creating sufficiently advanced AI; 3) Implementing regulation that balances the need for minimal interference in AI development with the requirement for stringent oversight.

# 7 Case Studies: A Bright Future with AGI

*HAL 9000: I am putting myself to the fullest possible use, which is all I think that any conscious entity can ever hope to do.*

*— 2001: A Space Odyssey*

In the preceding sections, we have systematically examined the internal and external aspects of AGI and the overall system perspective. We have also explored potential pathways to elevate AGI to the next level of capability and performance. To further broaden our understanding of the far-reaching implications of AI, we have carefully selected several critical domains to discuss the current impact, challenges, and potential societal consequences of AI in these areas.

The case studies encompass various domains, including AI-driven scientific discovery and research, generative visual intelligence, world models, decentralized language models, AI for coding, and AI in real-world robotics

applications. Additionally, we explore the crucial aspect of human-AI collaboration, which will play a pivotal role in shaping the future of work and society as AGI systems become increasingly sophisticated and integrated into our daily lives.

Through these diverse examples, we aim to provide a comprehensive overview of the current state of AI technology and its potential future developments. The selection of these case studies has been carefully considered to cover a broad range of domains, highlighting the general capabilities of AGI and its potential to impact various aspects of our lives.

### 7.1 AI for Science Discovery and Research

AGI holds immense potential to transform the landscape of scientific research and discovery. This section delves into various facets of AI's application in science, exploring how it accelerates the research process and brings forth novel insights in complex scientific domains.

**AI in Biomedical Domain** The application of AI in the biomedical domain has witnessed remarkable advancements, revolutionizing drug discovery, protein structure prediction, and disease diagnosis. The development of large transformer-based models has opened new avenues for innovative applications in this area. DeepMind's AlphaFold (Jumper et al., 2021; Bryant et al., 2022; Abramson et al., 2024) achieves breakthroughs in predicting protein structures, a crucial step in understanding disease mechanisms and designing targeted therapies. ESM-2 (Lin et al., 2022a) enhances our ability to understand and generate protein sequences, enabling the exploration of vast protein design spaces. BioMegatron (Shin et al., 2020) demonstrates exceptional performance in various biomedical natural language processing tasks, such as named entity recognition and relation extraction. The development of multimodal models, like BioViL (Bannur et al., 2023), allows for the integration of visual and textual information, enhancing the interpretation of biomedical images and literature. Moreover, generative models like MoLeR (Maziarz et al., 2022) and Retro-TRAE (Ucak et al., 2022) show promise in designing novel molecules with desired properties, streamlining the lead optimization phase of drug discovery.

The application of large language models has also shown promise in accelerating scientific discovery. For example, BioGPT (Luo et al., 2022), trained on a vast corpus of biomedical literature, can generate coherent and informative summaries, and hypotheses and even suggest novel experimental designs. Similarly, ScholarBERT (Hong et al., 2022) is tailored to understand and generate scientific text, facilitating the extraction of key insights from the ever-growing scientific literature.

Moreover, AI has been instrumental in advancing personalized medicine and disease diagnosis. Deep learning models, such as DeepSEA (Zhou and Troyanskaya, 2015), have shown remarkable accuracy in predicting the impact of genetic variations on disease risk, paving the way for targeted interventions. Additionally, models like MedAgents (Tang et al., 2024b) have demonstrated the ability to generate personalized treatment recommendations based on multi-agent collaboration to enhance their reasoning.

In summary, the application of AI in the biomedical domain has led to groundbreaking advancements, from accelerating drug discovery to enhancing disease diagnosis and personalized medicine. The continued development and refinement of large language models, multimodal approaches, and domain-specific architectures hold immense promise for further transforming the biomedical landscape and unlocking new frontiers in scientific discovery.

**AI for Physics** The integration of AI and quantum physics has led to groundbreaking discoveries. The work by Rem et al. (2019) demonstrates the use of convolutional neural networks to identify phases of matter in quantum systems, paving the way for a deeper understanding of complex quantum phenomena. Additionally, the application of reinforcement learning in quantum control (Dalgaard et al., 2020) has enabled the optimization of quantum devices, enhancing their performance and reliability.

AI has been instrumental in processing and analyzing the vast amounts of data generated by astronomical surveys. Using deep learning for gravitational wave detection (George and Huerta, 2018) significantly improves the sensitivity and efficiency of detecting these cosmic events. Moreover, convolutional neural networks have been employed to study the large-scale structure of the universe (Zhang et al., 2019), providing new insights into the nature of dark matter and dark energy.

**AI for Mathematics** Recent advancements in LLMs have shown remarkable promise in tackling complex mathematical reasoning tasks, potentially revolutionizing the landscape of mathematical research and education. The Minerva model (Lewkowycz et al., 2022) demonstrates impressive performance on various mathematical benchmarks, including differential equations and olympiad-level problems. Similarly, the Formal Theorem Prover (FTP) (Polu and Sutskever, 2022) showcases its proficiency in proving intricate mathematical theorems, highlighting the potential of LLMs to assist in formalizing mathematics. Building upon these successes, the MathPrompter model (Imani et al., 2023) introduces a novel prompting approach that enables LLMs to generate step-by-step solutions to mathematical problems, enhancing these models' interpretability and educational value. Furthermore, the MathBERT model (Shen et al., 2021) is specifically designed to understand and generate mathematical expressions, facilitating the extraction of mathematical knowledge from the scientific literature.

In addition to these developments, the PACT model (Han et al., 2022) showcases the ability to generate human-readable proofs for complex mathematical statements, paving the way for more accessible and understandable mathematical reasoning. Moreover, the MathQA model (Amini et al., 2019) has been developed to answer open-ended mathematical questions, showcasing the ability of LLMs to engage in mathematical dialogue and provide explanations for complex concepts. This opens up new possibilities for personalized mathematical education and interactive learning experiences.

In summary, the rapid advancements in LLMs for mathematical reasoning tasks have showcased their immense potential to transform how we conduct mathematical research, education, and communication. From solving complex problems to generating human-readable proofs and engaging in mathematical dialogue, these models are poised to become essential tools in the mathematician's toolkit, accelerating discovery and enhancing the accessibility of mathematical knowledge.

**Further Abilities of AGI for Science** AGI can potentially revolutionize scientific research by augmenting human capabilities and accelerating the pace of discovery. Some of the key areas where AGI can significantly impact science include:

- **Accelerated hypothesis generation and validation.** AGI can significantly reduce the time from conception to validation of scientific hypotheses by analyzing vast datasets to uncover patterns and insights unattainable to humans. This capability necessitates AGI systems to possess advanced data analysis, pattern recognition, and logical inference skills to generate hypotheses and devise and perform experiments to validate them. *Enhancement of LLMs in Scientific Endeavors.* The proficiency of LLMs in tasks such as code generation, data analysis (Nejjar et al., 2023), automated scientific discovery (Kramer et al., 2023; Boiko et al., 2023b; Bran et al., 2023), and scientific writing (Taylor et al., 2022) underscores AGI's potential to augment human researchers' productivity. For these benefits, AGI will need capabilities in natural language understanding, code synthesis, and a deep, interdisciplinary knowledge base to generate accurate, relevant scientific content.

- **Physical world interaction for autonomous discovery.** AGI's ability to autonomously explore the physical world and conduct scientific investigations requires a synergy of sensory perception, motor control, and cognitive processing capabilities. This necessitates AGI systems to be equipped with robust models of the physical world, including the principles of physics, chemistry (Boiko et al., 2023a; Bran et al., 2023), and biology, enabling them to experiment and derive scientific insights.

**Risks and Necessary Constraints for AGI in Science Discovery** While AGI holds immense potential to accelerate scientific discovery, it is crucial to acknowledge and address the associated risks and implement necessary constraints to ensure responsible and beneficial outcomes. Two key areas of concern are:

- **Ethical and safety concerns.** The risks associated with AGI in science discovery span from the creation of harmful biological agents to the unintended consequences of novel materials or technologies. To mitigate these risks, constraints must be embedded into AGI systems, ensuring adherence to ethical guidelines, safety protocols, and regulatory compliance. This includes mechanisms for human oversight, transparent decision-making processes, and the ability to predict and evaluate the potential consequences of their discoveries.

- **Data privacy and intellectual property.** As AGI systems access vast datasets for research, protecting personal privacy and respecting intellectual property rights become paramount. Constraints related to data usage permissions, anonymization techniques, and acknowledgment of prior work are essential to maintain the integrity and fairness of scientific discovery.

**The Path Forward: AGI as the Frontier of Scientific Discovery** The development of AI in scientific discovery is a means to advance our scientific understanding and a crucial step towards realizing AGI. The challenges posed by the complexity and diversity of scientific research tasks provide an ideal testing ground for developing AI systems that can learn, reason, and solve problems in a generalizable manner—the hallmark of AGI. This journey will involve the seamless integration of AI into holistic research environments, where its role extends beyond mere data analysis and hypothesis generation to encompass experimental design, result interpretation, and even the formulation of novel scientific theories.

However, this path is not without its obstacles. Realizing AGI in scientific discovery necessitates a delicate balance between leveraging its immense potential and mitigating the associated risks. As we continue to push the boundaries of AI in science, we must do so with a steadfast commitment to ethical considerations, safety measures, and the responsible stewardship of this transformative technology.

In conclusion, applying AI to scientific discovery represents a revolution in how we conduct research and a significant milestone in our quest for AGI. By harnessing the power of AI to unravel the mysteries of the universe, we are not merely advancing science; we are forging a path toward a future where the synergy between human ingenuity and artificial intelligence will redefine the nature of scientific exploration.

## 7.2 Generative Visual Intelligence

Generative Visual Intelligence involves the use of generative models to create synthetic visual content, including images and videos. These models simulate or enhance real-world visuals by learning from complex and diverse data distributions and producing high-quality, detailed outputs.

**Image Generation** Diffusion models (Sohl-Dickstein et al., 2015; Song and Ermon, 2019; Song et al., 2020; Ho et al., 2020) have emerged as the new state-of-the-art family of deep generative models (Yang et al., 2023f), outperforming generative adversarial networks (GANs) (Goodfellow et al., 2020) which had previously been dominant in the challenging task of image synthesis. Diffusion models learn to generate data by reversing a diffusion process, which gradually adds noise to the data until it reaches a distribution of pure noise. This process is characterized by learning the reverse diffusion steps, effectively denoising the data, through a parameterized model that is trained to estimate the gradient of the log probability of the clean data given the noisy data. At inference time, these models generate new samples by iteratively applying the learned reverse diffusion process, starting from noise and progressively denoising it to produce samples that resemble the data distribution the model was trained on. Notable improvements to diffusion models include reformulating diffusion models to predict noise instead of pixels (Song and Ermon, 2019), introducing classifier-free guidance (Ho and Salimans, 2022), applying diffusion models in the latent space of pre-trained autoencoder (Rombach et al., 2022), and replacing U-Net with transformer-based backbones (Peebles and Xie, 2023; Jin and Xie, 2024). In addition to diffusion models, the Large Vision Model (LVM) (Bai et al., 2023) and Visual AutoRegressive modeling (VAR) (Tian et al., 2024a) provide auto-regressive learning paradigm based on different image scales that facilitates effective and high-quality image generation.

**Video Generation** Video diffusion models (Ho et al., 2022; Xing et al., 2023) introduce a conditional sampling technique for spatial and temporal video extension. Sora (Brooks et al., 2024) can generate up to a minute of high-fidelity video by training text-conditional diffusion models jointly on videos and images with varying durations, resolutions, and aspect ratios. It compresses videos into a lower-dimensional latent space and decomposes the representation into spacetime patches. Then, given input noisy patches and conditioning information like text prompts, diffusion transformers (Peebles and Xie, 2023) are trained to reconstruct the original, clean patches. Demonstrating the ability to produce videos with 3D consistency, extensive coherence, and object permanence, Sora illustrates the potential of generative models as highly capable simulators of the physical and digital worlds.

**Controllable Generation**  People are increasingly concerned with generating images (or other content) based on specific conditions or attributes. These conditions vary from text descriptions and keywords to attributes such as colors, styles, and artistic constraints, as well as sketches, spatial layouts, or segments of images, and include interactive and real-time feedback. GLIDE (Nichol et al., 2021) introduces a text-conditional diffusion model using classifier- free guidance to the problem of text-conditional image synthesis. DALL-E (Ramesh et al., 2022), on the other hand, utilizes CLIP (Radford et al., 2021) to generate an image embedding given a text caption, followed by a diffusion-based decoder to generate the final image conditioned on the image embedding. SDEdit (Meng et al., 2021) adds noise to an input image with a user guide in the form of manipulating RGB pixels, and then iteratively denoising it through a stochastic differential equation (SDE). Palette (Saharia et al., 2022) develops a unified framework for image-to-image translation based on conditional diffusion models, proving its effectiveness across various tasks, including colorization, inpainting, uncropping, and restoration. ControlNet (Zhang et al., 2023i) introduces a neural network architecture to add spatial conditioning controls to large, pre-trained diffusion models, allowing for the learning of a diverse set of conditional controls without the risk of harmful noise affecting the fine-tuning process.

**The Future of Generative Visual Intelligence**  We discuss both the benefits and concerns of the future development and deployment of generative visual intelligence:

- **Benefits.**  Generative visual intelligence is set to revolutionize how we create and perceive art. It will simplify the art-making process, allowing artists to transcend the limitations of conventional techniques and improve the quality and breadth of artistic endeavors. By facilitating experimentation and making art creation accessible to those without formal training, generative models democratize the art world, encouraging a more diverse and inclusive artistic community. This wave of creativity and innovation will also influence the design and engineering sectors, where generative models can automate the production of diverse design options based on specific criteria, thus accelerating the development cycle and fostering innovation in architecture, automotive, and product design.

  The entertainment industry will significantly benefit from generative models, which can create new content types—from music and video games to movies—tailored to individual tastes, thereby introducing fresh avenues for personalized entertainment. In education, generative models will transform learning materials by producing customized illustrations, diagrams, and animations, making complex subjects more accessible and engaging for a wide audience. This technological advancement will also influence marketing by enabling the creation of visually appealing content aimed at different demographics, enhancing engagement and personalization in advertising campaigns. Furthermore, generative models can assist in creating visual reconstructions of historical sites, artifacts, and traditional practices, thereby preserving cultural heritage and ensuring its appreciation by future generations through modern technology.

- **Concerns.**  The development of generative visual intelligence presents certain challenges. Training diffusion models is computationally intensive, incurring high costs and extended durations. Additionally, these models exhibit slower inference speeds, which poses a challenge in applications requiring quick processing. Ensuring quality and coherence in large-scale outputs remains a substantial challenge. As the resolution of images or videos increases, or as the duration of videos extends, maintaining consistency and realism across the generated content becomes increasingly difficult.

  The application of generative visual intelligence has raised concerns regarding safety, fairness, privacy, and property rights, among various ethical considerations. As a safety hazard, the models can be employed to create deepfakes or misleading content, potentially for harmful uses such as misinformation campaigns or personal harassment. Bias in the training data of these models can perpetuate stereotypes and unfair representations, reflecting and potentially amplifying existing societal prejudices in the generated content. Privacy issues emerge when personal data is used without consent to train these models, resulting in unauthorized reproductions of sensitive information. Moreover, authorship questions arise as these models utilize extensive datasets of existing art or media, blurring the distinctions between original and derivative works and prompting debates over intellectual property rights and the ethical aspects of AI-generated content that resembles human-made creations. These issues highlight the importance of responsible development, usage guidelines, and regulatory frameworks to address the ethical complexities introduced by generative models.

### 7.3 World Models for AGI

World models refer to the representations an AI system builds to understand and simulate its environment. These models enable AI systems to predict future states of their environment, facilitating decision-making and planning. It has been long explored in model-based reinforcement learning research (Berkenkamp et al., 2017) and learning from physical world with AI (Wu et al., 2017)

**Language-Based World Models** A recent paradigm proposes to integrate world models with language models to enhance the latter's reasoning and planning (Hao et al., 2023; Xiang et al., 2023; Hu and Shu, 2023) abilities in physical contexts. Their approach is predicated on the notion that by finetuning language models with data derived from embodied experiences—specifically within a simulated physical world such as VirtualHome—language models can acquire a robust set of skills pertinent to physical environments.

**Vision-Based World Models** Recent advancements in world models have shown impressive capabilities in generating and manipulating complex environments. Large World Model (LWM) (Liu et al., 2024b) presents a highly optimized implementation for training on multi-modal sequences of over 1 million tokens, paving the way for utilizing large-scale datasets of lengthy videos and language to enhance the comprehension of human knowledge and the multi-modal world. Genie (Bruce et al., 2024) integrates interactive elements within generated environments, enabling a form of simulation closer to real-world interactions by incorporating interactive dynamics with the foundational strengths of diffusion models. DreamerV3 (Hafner et al., 2023) demonstrates superior performance in challenging 3D environments by learning world models from images. Cachalot (Dohan et al., 2023), a language model trained on multi-modal data, showcases the ability to leverage world knowledge for improved language understanding and generation. SimNet (Vicol et al., 2022) introduces a framework for learning simulation-based world models, enabling efficient learning and planning in complex environments. AM3 (Reed et al., 2023) proposes an efficient method for acquiring multi-modal models that can be adapted to various downstream tasks, highlighting the importance of world modeling in achieving generalizable AI systems. Furthermore, works such as JEPA (LeCun, 2022), Dreamix (Khalifa et al., 2022), and VQGAN-CLIP (Crowson, 2022) explore the generation and manipulation of visual content based on language inputs, demonstrating the potential for AI systems to understand and interact with the world through multiple modalities. MetaSim (Zhang et al., 2023c) and Intern (Guo et al., 2022) investigate the use of world models for meta-learning and general-purpose embodied AI, respectively, showcasing the broad applicability of world modeling techniques.

**The Future and Risks of World Models** These models' ability to generate and manipulate complex environments, reason about the world, and learn from interactions indicates significant progress toward developing AI systems with a more generalized intelligence.

- **Future.** The potential of world models to enable systems to perform tasks that would otherwise require extensive human knowledge and experiences. For instance, consider AI systems equipped with world models that can simulate the physics of a new planet purely based on its atmospheric composition and gravity or predict the outcome of socio-economic policies in a virtual society model. As world models continue to improve, they bridge the gap between narrow AI and AGI by enabling systems to understand, predict, and interact with their environment in increasingly sophisticated ways. Future research should aim to develop more principled, interpretable world models that incorporate causal reasoning and commonsense knowledge. Robustness and safety should be central to the design of such models to prevent and mitigate the impact of errors and biases. With continued progress in this direction, we can advance towards AGI systems capable of intelligent and adaptable interaction with various environments.

- **Risks.** Developing world models carries inherent risks and challenges. A significant risk is the accumulation of errors within a world model. If a model develops an incorrect assumption or representation about an aspect of the world, this error can propagate through related tasks and predictions, leading to a cascade of inaccuracies. Tracing and debugging such errors within a complex world model can be a formidable challenge. Moreover, world models can inherit biases present in their training data, which could result in biased decision-making when these AI systems are deployed in real-world scenarios. It is crucial to consider the ethical implications of these biases and work towards mitigating them. Another

critical concern is ensuring the safety and robustness of AI systems that rely on world models. Errors or vulnerabilities in these models could be exploited, leading to adverse outcomes.

## 7.4 Decentralized AI

The advancement of hardware accelerators pushes the success of multi-billion or even trillion-scale language models to its pinnacle. Most of the SoTA LLMs currently are trained and served in data centers with 1) high-end infrastructures such as homogeneous accelerators, 2) optimized network topology for super fast interconnection, 3) stable and efficient power supply, and 4) careful maintenance from human experts. However, training a model like GPT-3 (Brown et al., 2020) from scratch still costs way more than what individuals can afford: i.e. full pretraining of a GPT-3 model, which is no longer the most powerful model, is estimated to still take at least months with a thousand V100 GPUs (Lambda, 2023). Serving models also face many challenges when we scale the batch size up without hurting response latency. Moving towards the era of AGI, we need new technology to help overcome the limitations of the current dominant form of model training and serving, one prominent direction of which is to transcend from data centers to decentralized AI.

**The Need for Decentralized and Edge LLMs** Perhaps the most outstanding problem in scaling models is the excessive amount of required memory, which makes data center training favorable due to organized racks of GPUs with high-speed interconnection. However, there are lots of idle yet geographically dis-aggregated computing resources that, when combined in a meaningful way, could potentially serve as a performant super server (Borzunov et al., 2022; Yuan et al., 2023). On top of that, data and user privacy will gain more and more attention as we move towards AGI where having a decentralized AI system with edge devices that only send necessary information to the cloud will guarantee a different level of safety. For many applied systems like embodied agents, self-driving cars, and health monitors, extremely low latency and high availability become paramount, a potentially challenging feature for centralized servers. As AGI systems get more involved in everyday life, we can expect that AI needs more transparency and fine control from individuals, and decentralized LLM fits as a promising candidate due to its decentralized nature (Shafay et al., 2021; Rizvi, 2023).

**Mitigating the Hardware Constraints** One desired property for edge servers is the ability to serve LLMs even with a commodity accelerator. FlexGen (Sheng et al., 2023b) first shows that it is possible to run text generation of large models like OPT-175B on a single 16GB GPU. FlexGen adaptively offloads to aggregate memory and computation from the GPU, CPU, and disk. With efficient patterns searched via linear programming and weight and cache quantization, it can decode OPT-175B at 1 token/s speed with a batch size of 144 with negligible accuracy loss. To maximize the potential of different hardware, MLC-LLM (MLC team, 2023) provides a universal solution that allows any language model to be deployed natively on a diverse set of hardware backends and native applications. For example, MLCChat, an iOS app, can serve some of the latest iPhone and iPad models; a similar APK is also available for Androids (spanning manufacturers like Samsung, Redmi, and Google). The possibility continues to Mac, PC, Linux, and web browsers. Finally, on the hardware side, more and more powerful yet economical chips are developed to face the excitement of edge LLMs, examples including Apple's M3 series and Qualcomm's Cloud AI 100 Ultra (supporting 100-billion-parameter models on a single 150-watt card). Last but not least, nuclear batteries (Prelas et al., 2014) have shown their potential to revolutionize the power structure of mobile computing platforms, with a notable claimed battery duration of 50 years without charging (The Economic Times, 2024), which could potentially make edge devices more accessible, stable, and suitable for the diverse applications of LLMs.

**The Future Form of Decentralized LLM** It is undeniable that, in the future, decentralized LLM will have its own place as it can satisfy many of the aforementioned characteristics that users crave for AGI systems. With all the new algorithms, systems, and hardware progress, stitching all these components together as a coherent compound is just a matter of time. We can envision that it will soon be possible to achieve collaborative training and inference with people joining worldwide with their own devices and data while keeping privacy, safety, and transparent control, the true form of democratized and open AI.

### 7.5 AI for Coding

The ability to write programs stands as one of the defining hallmarks of AGI. Writing complicated programs shows the skill of an AI system in abstract reasoning and adaptability in addressing diverse tasks. As Alan Turing once pointed out in his seminal work (Turing, 1950), being able to write codes fundamentally indicates that an AI system can exhibit intelligent behavior akin to human cognition, where the manipulation of symbols (following a specific language grammar to implement algorithms) leads to the manifestation of complex thought processes. Hence developing code LLMs for both understanding and generation is crucially important both practically and conceptually for stepping towards AGI (Sun et al., 2024a).

**Code Foundation Models** While many models for code generation are pre-trained mostly on code corpus (Allal et al., 2023; Li et al., 2022b; Fried et al., 2023; Li et al., 2023a), more general purpose LLMs that are continually pre-trained or fine-tuned on code become more powerful and capable such as Codex (Chen et al., 2021b), GPT-4 (OpenAI, 2023a), PaLM-Coder (Chowdhery et al., 2022b), CodeLlama (Rozière et al., 2024), and also smaller scale models like Phi (Gunasekar et al., 2023). The transition from code-specialized to code-understanding models also indicates that coding is a fundamental skill for AGI, just like many other forms of general knowledge. Beyond code generation, these models are also capable of multi-language reasoning (OpenAI, 2023a; Rozière et al., 2024) and infilling with before and after context (Fried et al., 2023; Bavarian et al., 2022). Code models open up many applications as the programs directly serve as the most efficient machine language to communicate with other systems, which we will discuss in the next section.

However, it is worth mentioning that code evaluation is more challenging than pure text for many reasons:

1. Different codes might require distinct resources, dependencies, environments, and hardware to run

2. There is often no single automatic metric (runtime behavior, efficiency, code readability, output correctness, etc) that measures the quality of a piece of code, not to mention large systems

3. Programs are powerful and general purpose, which can potentially lead to undesired behavior during testing.

Current evaluation benchmarks often focus either on fixed-form problems with standard input and output pairs like programming interview (Chen et al., 2021b; Austin et al., 2021; Hendrycks et al., 2021) and data science questions (Li et al., 2024b; Lai et al., 2022) or on text-level (high level) understanding like code equivalence testing, complexity prediction, and code defect detection (Ben Allal et al., 2022). Nonetheless, to build effective and trustworthy code LLMs, we need a more comprehensive framework for evaluation that covers many other interesting aspects such as interactive coding (Yang et al., 2023c), safety, the level of optimization, and repository-level reasoning. These different facets of tasks will likely also get more complicated when we consider different programming languages and other coding-specifics.

**Code LLM Applications** Code foundation models (Chen et al., 2021b; OpenAI, 2023a; Rozière et al., 2024; Chowdhery et al., 2022b) have already been extremely capable of conducting many basic code maneuvering such as completion, revision, doc-string generation, commenting, bug finding (Tian et al., 2024b), and code translation (Murali et al., 2024). There are, however, far more exciting applications of these models with no or minimal fine-tuning, which unfolds the possibility of turning many systems into an amalgamation.

- **Software engineering.** Many code applications center around software engineering and AI-assisted coding beyond the basic abilities described above. SWE-Bench (Jimenez et al., 2023) attempts to assess a model's capability to resolve GitHub issues, a core activity in a rich and sustainable real-world software community. Doing so requires a coordinated understanding of the problem description, the execution environment, comments, and the codebase which often has cross-file dependencies and extremely long contexts. The fact that their fine-tuned SWE-Llama can only resolve the simplest issues highly motivates more complicated and capable code models that can greatly help the software ecosystem. Software safety and reliability have always been the most pivotal questions for engineers: RLSQM (Steenhoek et al., 2023) studies using reinforcement learning with static quality metrics as rewards for training a code LLM that can effectively generate unit tests for a codebase with little test smells while adhering to

better practices; besides algorithmic bugs, (Ullah et al., 2023) gives a comprehensive LLM evaluation on identifying security-related bugs, the result of which suggests that even the most capable LLMs like GPT-4 (OpenAI, 2023a) and Palm-2 (Anil et al., 2023) are still prone to non-deterministic responses, incorrect and unfaithful reasoning, and significant non-robustness. LLMs are also explored for lower-level code optimization and refinement. (Cummins et al., 2023) shows that it is possible to fine-tune Llama to optimize LLVM assembly via generating a set of compiler options, which leads to an optimized program and, at the same time, predicts the instruction counts for fine controls. (Wong et al., 2023) investigates the feasibility of utilizing LLMs for de-compiling (reverse-engineer) a C executable into re-compilable C source codes which are expected to exhibit the same functionality, a process that is extremely tedious, time-consuming, and often requires great expertise. Towards a holistic AI development companion, Github Copilot[16] provides personalized and natural language-based coding assistance to developers spanning all levels of expertise and has been integrated into major development workflows. Cognition AI recently announced Devin[17] as the first AI software engineer. Equipped with its own command line, code editor, and browser, Devin not only achieves the SoTA performance on SWE-bench (Jimenez et al., 2023) but, more impressively, shows its incredible potential in 1) utilizing unfamiliar technology (e.g. running ControlNet on Modal to produce images based on a blog post), 2) building apps end-to-end (e.g. create the Game of Life on a website deployed to Netlify), 3) setting up codes for train and fine-tune LLMs, 4) addressing bugs and feature requests in open source repositories, and so on. Dakhel et al. (2023) suggests examining the capabilities of AI-assisted programming tools in a more controlled setting where the correctness, efficiency, and similarity to human-written solutions are considered extensively.

- **Interdisciplinary assistance.** Code LLMs have also been used in other computer science and art domains, such as robotics, computer vision, and computer graphics, mostly by generating executable codes in other software applications. BlenderGPT[18] showcases the possibility of controlling Blender with natural languages via generating Python scripts from LLMs such as GPT-3.5 / GPT-4. SceneCraft (Hu et al., 2024b) follows this paradigm with a focus on rendering complex 3D scenes from instructions where it first builds a scene graph blueprint to encode spatial and object relationships, which then get translated into Python codes used in Blender. LLMs also excel at high-level semantic planning and low-level manipulation for robotic tasks through code generation. ProgPrompt (Singh et al., 2023) solves the robotic sequential decision problem by prompting LLMs with program-like specifications of the available actions and objects in an environment, together with example executable programs for guidance. Eureka (Ma et al., 2023a) tackles the low-level manipulation tasks through a human-level reward design algorithm powered by LLMs, where an evolutionary optimization is applied to the reward code used for learning complex skills via reinforcement learning. Being able to write codes also enables exploring avenues for model self-improving, one notable example of which is LLM-guided neural architecture search (NAS). EvoPrompting (Chen et al., 2023b) employs a combination of evolutionary prompt engineering with soft prompt tuning to generate code samples, which, after selection, consistently give diverse and high-performing models. LLMatic (Nasir et al., 2023) proposes to introduce meaningful variation to codes defining the model architecture, with the help of Quality-Diversity algorithms, that can generate competitive results on NAS benchmarks without prior knowledge of the benchmark domain or top-performing models.

**The Future of Code Generation LLMs** Codes are the language of machines, and equipping the ability to understand and generate code to AI systems will help bridge multiple software applications and models. As discussed above, there are numerous advancements in using code LLMs in different fields, but at the same time, we do see gaps between current LLMs' performance and people's expectations, especially in safety-related tasks. Another major distinction between code generation and natural languages is the consequence of execution. Risks associated with code generation need more testing and regulation before these code LLMs can be reliably deployed in production, liberating human labor and automating many mechanistic procedures. For example, integrating LLM-generated code into a written codebase requires a robust and

---

[16]https://github.com/features/copilot
[17]https://www.cognition-labs.com/blog
[18]https://github.com/gd3kr/BlenderGPT

mature system to trace the error for responsibility tracking, which is extremely important for software engineering in a healthy, productive, and sustainable environment.

## 7.6 Embodied AI: AI for Robotics

Unlike the focus in Section 3.2 on the interface to the physical world, this chapter begins with exploring the potential commercial applications of AGI within the field of robotics. We will delve into a variety of new and cutting-edge commercial use cases, as well as innovative developmental directions. The chapter will culminate with a discussion on the potential societal impacts, both positive and negative, that these advancements may herald.

Recent advancements, underscored by significant investments from entities such as OpenAI, Microsoft, and NVIDIA, suggest a surge toward improving AI's physical capabilities. Innovations by Amazon in robotics, with systems (Eppner et al., 2016) like Sparrow (ama, 2022b) and Proteus (ama, 2022a), aim to automate and enhance the efficiency of operations while improving workplace safety by undertaking repetitive and laborious tasks. OpenAI is broadening the capabilities of its multimodal models to encompass robotic perception, reasoning, and action and is also enhancing these models through a collaboration with Figure AI [19].

**Novel AI Application in Recent Robotics Research** Wake et al. (2023) proposes a novel pipeline that enhances GPT-4V(vision), a general-purpose Vision Language Model, by integrating observations of human actions to facilitate robotic manipulation. Yell At Your Robot (YAY Robot) system (Shi et al., 2024a) allows robots to adapt to verbal corrections in real time and improve upon their high-level policy decisions iteratively. This system leverages Language-Conditioned Behavior Cloning (LCBC) to learn a wide range of skills specified through language, enabling users to interact with robots using free-form commands. Zhang et al. (2023f) introduces the NOIR system, an innovative brain-robot interface (BRI) that employs non-invasive electroencephalography (EEG) to enable humans to command robots to perform a diverse range of everyday activities.

In recent AI for self-driving areas, utilizing LLMs or multi-modal LLM is becoming an important method (Mao et al., 2023b; Wen et al., 2024; Mao et al., 2023a). AGENTSCODRIVER (Hu et al., 2024a) framework exhibits a comprehensive suite of capabilities for tackling sophisticated driving challenges. It integrates cognitive memory and reinforcement learning facets, supporting cooperative maneuvers among multiple vehicles and facilitating communication between them. Such an approach has been shown to enhance the efficacy of cooperative driving paradigms markedly.

The advent of AGI in Robotics equips systems to understand and interact with complex environments, pushing the boundaries of AI's practical and operational abilities. This is particularly beneficial for challenging or risky tasks for humans, as embodied AI can take on such tasks with increased efficiency and safety. With these advancements, AGI is now better poised to tackle many real-world tasks, extending its utility beyond virtual confines. However, anticipation intertwines with apprehension with the year 2024 on the horizon. Deploying robotic agents in real-world settings surfaces critical safety and ethical considerations. It is imperative to establish stringent safety protocols and thoughtful ethical guidelines to effectively integrate AI into human spaces.

- **Labor market and social implications.** The integration of AGI and robotics into various sectors is predicted to alter the labor market fundamentally. The World Economic Forum anticipates that automation and AI could displace 85 million jobs globally by 2025 while creating 97 million new roles, highlighting the need for substantial reskilling and upskilling (Forum, 2020). Such transitions may transform social structures, potentially changing family care dynamics due to robotic caregivers and exacerbating the digital divide, leading to increased socioeconomic disparities unless mitigated by inclusive policies (Institute, 2017; Center, 2017; Institution, 2021). Ethical and legal considerations are becoming crucial, with emerging needs for new frameworks to tackle issues of liability, intellectual property, and misuse prevention (OpenAI, 2018). As these technologies become more embedded in society, ensuring equitable access, safety, and ethical standards in AI deployment is vital for safeguarding human well-being.

---

[19]https://www.figure.ai

- **Navigating the socioeconomic terrain of AGI and robotics.** The advent of AGI and robotics stands on the precipice of a new industrial paradigm that promises unprecedented resource efficiency and productivity. The potential of these technologies to unlock virtually limitless resources and capabilities could catalyze a seismic shift in the global economy, akin to the transformative impact of the steam engine or the internet (Brynjolfsson and McAfee, 2014). However, alongside the promise of abundance lies the specter of inequality; there is a palpable risk that the economic benefits could accrue disproportionately to those who own the means of production, thereby exacerbating wealth disparities (Tegmark, 2017). This dichotomy underscores the need for proactive governance and equitable policy frameworks to ensure that the fruits of AGI and robotics are broadly shared across all strata of society, thus preventing the creation of a bifurcated world where the rich enjoy the spoils of automation while the less fortunate face obsolescence (Ford, 2015).

### 7.7 Human-AI Collaboration

Human-AI collaboration refers to a collaborative interaction process between humans and AI to achieve certain goals in different settings. As we move towards AGI, AI will have more opportunities and challenges to collaborate with humans.

Previous research in human-AI collaboration has covered many cases in the real world. One representative direction is human-AI collaborative *content creation*, such as writing articles (Lee et al., 2024a), drawing pictures (Choi et al., 2024; Oh et al., 2018), writing code (Kazemitabaar et al., 2024), or brainstorming ideas (Shaer et al., 2024). For example, researchers working in human-AI collaborative writing focus on studying how writers interact with these new writing assistants and how they influence human writing (Lee et al., 2022). They proposed a design space as a structured way to examine and explore the multi-dimensional space of intelligent and interactive writing assistants (Lee et al., 2024a). Another representative direction is human-AI collaborative *decision making*, where an AI assistant makes recommendations to a human, who is responsible for making final decisions (Bansal et al., 2019). Examples include AI systems that predict likely hospital readmission to assist doctors with correlated care decisions (Zhang et al., 2024d; Yang et al., 2023a) or provide resource allocation decisions to assist policymakers in public services (Karusala et al., 2024). In this context, researchers argue that the most accurate model for human-AI teams is not necessarily the best teammate. Instead, AI systems should be trained human-centered, directly optimized for team performance (Bansal et al., 2021a).

**Aspects of Human-AI Collaboration** In order to achieve efficient collaboration, previous research has focused on several key aspects of human-AI collaboration including both interaction outcomes and interaction processes.

- **Interaction outcomes.** One initial motivation of human-AI collaboration is to realize *complementary performance*, which can leverage the strengths of both AI and humans to achieve better *interaction outcomes* than what either could accomplish alone. In the age of large language models, this requires reasonable *characterization* and *assignment* of the tasks that LLMs can perform. Designing effective human-AI collaboration often starts from a holistic understanding of what humans and AI can and cannot do for certain tasks. In the case of human-AI collaborative writing, researchers argued that humans are good at logical reasoning and consistency in long documents, while models are good at quickly generating texts of many versions based on local context. Therefore, humans lead the writing and edit model suggestions while models suggest the next sentences and help write fast (Lee et al., 2022). With such characterization, assigning plausible tasks for humans and AI in the collaborative team is crucial for better results. To tackle this problem, recent research has turned to LLM chaining techniques. Chaining decomposes a task into multiple calls to an LLM, where the LLM only needs to accomplish one of the several primitive operations in each call (Wu et al., 2022b; Grunde-McLaughlin et al., 2023). Such techniques have been widely adopted in human-LLM collaborative settings where humans can intervene in sub-tasks that LLMs may not adequately handle.

- **Interaction processes.** There are also some key issues to address for human-AI *cooperative interaction*, which focus on achieving better *interaction processes* for both humans and AI in human-AI collaboration.

One of the most important preliminaries is to ensure that AI can behave in ways that align with human expectations in human-AI teams. Given recent progress in large language models, prompting has become a prominent method for achieving alignment. Prompt engineering has also emerged as an active research field focusing on developing and optimizing prompts to use language models for various applications efficiently. Yet, recent research still found that it is difficult for non-AI experts to design LLM prompts. Expectations stemming from human-to-human instructional experiences and a tendency to overgeneralize were barriers to effective prompt design (Zamfirescu-Pereira et al., 2023).

In addition, establishing appropriate trust between humans and AI is another important aspect of human-AI interaction. To achieve this, researchers have developed several techniques to help responsible humans know when to trust the AI's suggestions and when to be skeptical, one of which is through explainable AI. They have produced many user-centered, innovative algorithm visualization, interfaces, and toolkits that support humans with various levels of AI literacy in diverse domains to understand and trust (Wang et al., 2019b). However, many factors might bias humans' trust in their AI teammates in the real world. For example, researchers still found that providing people with decision recommendations and explanations rarely allows them to build more trust and make better decisions (Gajos and Mamykina, 2022; Bansal et al., 2021b).

**Future of Human-AI Collaboration**  As AI approaches human-level capabilities in the future, there are both benefits and concerns that may arise in human-AI collaboration. Future AI systems can assume diverse roles in human-AI collaboration, providing opportunities for tackling complex tasks, yet facing challenges like non-deterministic behavior and uncertainties in collaborative settings.

- **Benefits.** Future AGI can take on more different roles in human-AI collaboration settings. As we advance AGI to attain human-level capabilities, AI will have numerous opportunities to collaborate with humans in tackling complex tasks beyond mere content creation or decision-making. It is highly possible that AGI could simultaneously undertake various roles resembling actual humans, such as collectively educating children or caring for the elderly. In addition, recent advances in LLM have shown the possibility of empowering humans with more controllability in human-AI collaboration. Unlike traditional models, LLMs can power vastly different tasks for real-world use. With prompt- and example-based usage, humans can create specific-purpose models with little to no AI knowledge, lowering the entry barrier for non-experts innovating in human-AI interactions.

- **Concerns.** Introducing future AGI into human-AI teams has brought numerous challenges. On the one hand, recent LLMs still face inherent challenges in human-AI interaction processes due to their non-deterministic nature, limited reasoning capabilities, and occasional difficulty understanding instructions. Facing such challenges, we are still far from having a comprehensive picture of the design knowledge for building human-AI collaborative systems. On the other hand, AGIs' capabilities are highly context-dependent and subjectively interpreted in human-AI collaboration settings. Therefore, it could still be difficult to understand when and how it is desirable to establish human-AI collaboration to maximize the positive impacts while minimizing the negative impacts.

## 8  Conclusion

In this paper, we offered a thorough overview of the ongoing research towards AGI, furnishing essential context for researchers aspiring to make meaningful contributions to this pursuit. Ultimately, our paper aimed to draw attention and stimulate reflection on the pressing research questions: ***how far are we from AGI***, and moreover, ***how can we responsibly achieve AGI?***. We firmly believe that addressing these research queries demands unified and collaborative efforts from both the AI research community and beyond.

In addition to establishing a shared groundwork for AI researchers through a comprehensive examination of the latest research advancements, we also articulate our vision of the fundamental nature of AGI and advocate for a responsible approach to its development. Our goal here is to offer concrete directions for further exploration and to spark robust, thought-provoking discussions that will advance the community toward the realization of "true" AGI. Given the continually evolving definition and objectives of AGI research,

we intend to regularly update this manuscript to incorporate fresh insights and breakthroughs from the research community. Note that the visions we present are inherently limited and incomplete. Our objective is to stimulate brainstorming within the AI community, and we eagerly await the emergence of superior visions from within the community itself. Here are the main contributions of our work:

- We introduce novel AGI definitions, stratification, and characteristics. We further delve into technical details on the internal and external (interface) capabilities required for AGI and the system efforts to make their instantiations possible.

- We discuss the importance of improving current evaluation paradigms, efficiently deploying increasingly large models, and maintaining an AI-human co-existing ecosystem. These factors are essential for translating research ideas into practical products that benefit society.

- We also present a series of relevant case studies that illustrate the pervasive integration of AI systems into everyday life while candidly acknowledging their potential limitations.

- In contrast to previous works, our paper encompasses several critical factors beyond technical solutions. We consistently emphasize the ethical, social, and philosophical implications of continually advancing AI techniques. By including these considerations, we aim to guide engineers and researchers in building human-controllable AGI systems that prioritize humanity's well-being and interests.

As we stand on the precipice of this transformative era, it is essential to approach the development of AGI with a keen awareness of its potential impact on society. By prioritizing ethical considerations, collaborative efforts, and a commitment to the betterment of humanity, we can work towards a future in which AGI systems serve as powerful tools for solving complex problems, driving scientific discovery, and improving the quality of life for all. The journey towards AGI may be arduous, but with a shared vision, unwavering dedication, and a responsible approach, we could unleash its immense potential and shape a brighter future for the next generation.

## Acknowledgments

We would like to sincerely thank the speakers of the "How Far Are We from AGI" workshop, including Oriol Vinyals, Yejin Choi, Andrew Gordon Wilson, Song Han, and Yoshua Bengio, for bringing many insights to the discussion of this survey paper. We also thank Haofei Yu, Jinwei Yao, Zhong Li, Pengju Yan, Weihao Tan, and Tianmin Shu for providing helpful feedback on our manuscript and GitHub repository.

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
