# OpenReview forum: "How Far Are We From AGI: Are LLMs All We Need?"
_TMLR — Accepted by TMLR_

### Review · Reviewer_XGZH · 2024-08-02

**Summary Of Contributions:**

This paper proposes to review the current state of AGI research and proposes directions for future research. The paper highlights 5 major topics: Internals (memory, reasoning, perception, metacognition), Interfaces, Systems and Alignment. It concludes with a proposed roadmap to AGI as well as case studies.

The main contribution of this paper is that it is (to the best of my knowledge) the first and only paper articulating the state of AGI research (and milestones missing) in an holistic manner.

**Audience:**

Yes

**Broader Impact Concerns:**

n/a (survey paper)

**Claims And Evidence:**

Yes

**Requested Changes:**

# 1- Introduction

On figure 1: how is the fraction of activities where AI is superhuman "heuristically estimated"?
In particular, how do authors come to the fact that current AI systems outperform humans on more than 1% of tasks?
Cite(s) could be useful here, along with some basic explanation.

### Brief History of AI

Authors could set the historical context better by mentioning the Dartmouth conference with Minsky, which coined the term Artificial Intelligence, along with foundational papers [1 - 5].

The citation for "ChatGPT" should be InstructGPT [6] instead.. (at least the first citation).
The second citation seems a bit unnecessary.

Authors could replace "ChatGPT" with large language models? No need to put emphasis on a single instantiation (even if currently most prominent).

### Craving for General Purpose AI

Authors could mention more papers that discuss AGI définition (e.g. "Levels of AI", which is mentioned later on).

Also the related work (if it is one) is relatively thin: it would be great to setup the context in which this work is established.

"AGI is Within Closer Reach than Ever" -> please choose a better title.

# 2 - AGI Internal: Unveiling the Mind of AGI

Separate agency from reasoning? Seems that some humans have good reasoning but lack agency to fulfill their plans (example of a poor climber with good problem solving skills)

Unclear how general the framework (memory / reasoning / metacog / perception) is? Authors could discuss pros and cons of such a framework.

## 2.1 - AI Perception

### Current state of AI Perception

Authors mention that multimodal LLMs are trained to comprehend other modalities than text from existing LLMs -> in principle they could cover multimodality from scratch

### External Connection of Modalities

Text is not that clear. A figure could help distinguish the different families of methods.

### AGI level Perception

No citation for Gemini (appears later on but could be placed here).

## 2.2 AI Reasoning

Missing citations for PoT [7], zero-shot CoT [8], tool-integrated reasoning [9], more agents is all you need [10].

The description of Maieutic Prompting is not clear as is.

### AGI Reasoning

Authors could discuss the literature on causal understanding in LLMs - some works in that direction are [44 - 45].

"AGI should enhance social reasoning" -> change to "should get better at social reasoning" or similar

On explainability, recent works on Sparse Autoencoders should be mentioned: [11-13], and this whole line of research acknowledged in the text

"that allows reasoning planning" -> sentence missing an "and"

On dynamic reasoning, seems like it is a vibrant area of research (see Thought Rollback [14], REFT [15] and others) which could be mentioned

## 2.3 AI memory

Authors could mention model merging as a way to integrate knowledge in model weights

On long-context and memory utilization, I think things like Ring Attention [16] and LongRoPE [17] could be mentioned.

## 2.4 AI metacognition

On self-awareness in AGI, several notions are introduced (self-immersion etc) and could be attributed to previous works and / or explained.

# 3 - AGI Interface

"Interface to Digital World provides an accessible, practical, and scalable testbed for studying advanced cognitive skills in agents, promoting their development through language-based inputs and the ease of developing digital tools" -> this sentence is strange and feels out of context?

## 3.2 AI Interfaces to Physical World

Google's PaLM -> PaLM (unnecessary company attribution in the context of a scientific paper).

Missing, up-to-date citations on robotic control (Palm-E [19] / RT2 [18] / Mobile Aloha [20]).

"The ability to interpret and adapt to dynamic, real-world environments in a manner akin to human cognition." -> this is not a proper sentence

# 4 AGI Systems

## 4.2 Scalable Model Architectures

on self-attention mechanisms, not sure about the link between attention sparsity and the lottery ticket hypothesis?

On model compression, online distillation (Agarwal et al) should be mentioned.

"Beyond Transformers" -> Griffin / Hawk and subsequent integrations (i.e. RecurrentGemma) could be mentioned!

## 4.3 Large-scale Training

The description of LLAMA-Adapter is not particularly clear from the text.

“linearly probing” -> “linear probing”

Decentralization:
* DiLoCo (Douillard et al) could be mentioned

Data economy:
* Shapley values are mentioned but never explained
* this part is not super clear at the moment

## 4.6 Computing Platforms
* Though it is nice to see where we stand in terms of potential for computation, I am not sure listing precise accelerator names is necessary? (e.g. NVIDIA DGX GH200, Qualcomm Cloud AI 100 Ultra).

In Figure 6, the legend is a bit misleading: the arrows seem to indicate that “Model Copy” implies “Strong Connection”.

# 5 - AGI Alignment

My assessment is that this section lacks a bit in clarity and in organization of the existing literature as well.

## 5.1 Expectations of AGI Alignment

I find the choice of approaches to AI alignment peculiar (in “To achieve this goal, researchers have proposed various approaches to AI alignment, such as value learning
(Soares, 2016), inverse reinforcement learning (Hadfield-Menell et al., 2016), and cooperative inverse reinforcement learning (Hadfield-Menell et al., 2016)”): learning from preferences (and RLHF) should be mentioned given these are the default approaches used today. Also, this is not consistent with the rest of the section where mostly RLHF approaches are indeed presented.

## 5.2 Current Alignment Techniques

“These methods can generate online human supervision using techniques such as reinforcement learning” -> reinforcement learning (RL) does not generate online supervision, it optimizes online supervision (for instance given by a reward model trained on preference data). Please rephrase to cleanly separate the contribution of RL from that of reward modeling.

“Online human supervision” seems to imply that there are humans-in-the-loop, which is not the case. “Online Alignment from (Approximate) Human Feedback” could be an alternative title.

I am not sure to agree with the three main directions proposed by the authors (i.e. rule optimization, data utilization and theoretical justification).
* rule optimization seems like an important but also a minor aspect of the literature
* optimal data utilization (which we would call sample efficiency in the RL literature) is important but the works chosen to illustrate are not very representative
* on theoretical justification: most methods are theoretically motivated and are flavors of KL-regularized RL (both RL and direct preference optimization methods)

I would argue in favor of four main topics instead: reward modeling, optimization, data and self-improvement.

Here are some works I would cite for each of these topics (which are not cited in the current state):
* reward modeling: ODIN [25], WARM [26], Herding or Helping? [27], Reward model ensembles [28], West-of-N [29]
* optimization: NashLLM [32], ReMax [33], RFT [34], online iterated RLHF [35], WARP [36]
* data: OAIF [38]
* self-improvement (a lot of these are already mentioned)

Currently the works on online human supervision are not highly representative of the state of the literature.

Authors would do well discussing the gap between offline and online methods, and can rely on the following paper discussing the drawback of purely offline methods [39].

Text-based feedback signals: missing reference to SteerLM [40] and CLP [37].

Ranking-based feedback signals: missing reference to vBoN [30], BoNBoN [43] and BOND [31] (all flavors of distribution matching to the best-of-N distribution).

On Aligning with Interactive Supervisions:
* title could be superalignment ?? or weak-to-strong supervision ? or scalable oversight)
* “SPIN (Chen et al., 2024a) decomposes the training objective into progressively evolved goals and leverages the iterative self-improvement to improve strong models with a weak one” -> I do not agree with that sentence: SPIN is essentially about self-improvement (by increasing the log-likelihood of demonstrations and decreasing the log-likelihood of model completions), and its objective does not change along training. I would remove the mention of SPIN in the “Through task decomposition” paragraph.

## 5.3 How to Approach AGI Alignment

I would put the paragraph on “Interaction with tools and APIs” after “Interaction with humans”.

Interactions with tools and APIs
* “AGI cannot use APIs or tools to cause crimes during the interaction process (Zhang et al., 2024c; Yao et al., 2024; Chen et al., 2023a)” -> I think AGI should also keep to an absolute minimum potential damage overall (which includes mental harm done to humans, general harm to other species and even valuable goods – this is not mentioned in the current state).

In the “vision of the future in alignment techniques”, additionally to the topics mentioned I would put emphasis on safety and weak-to-strong alignment (i.e. scalable oversight).

# 6 AGI Roadmap

## 6.1 AGI Levels

I would consider presenting these much earlier in the paper.

## 6.2 AGI Evaluation

In “Super-evaluation”, I would mention the existence of formal proof systems (e.g. Lean) that could be used to verify superhuman mathematical outputs.

## 6.3 How to Get to Next AGI Level

The entire paragraph on “advanced code machines” (“Advanced code machine brings the ultimate AGI within easy reach”) is unintelligible to me. What such an advanced code machine is should be explained and the whole paragraph clarified.

Authors mention VAEs and GANs as potential world modeling tools, but could also mention diffusion models, and also Transformer-based world models such as in [41].

## 6.4 Further considerations

“In 1951, Shannon et al. proposed 3-gram to point to ninety-nine point six billion dollars from
two hundred four oh six three percent of the rates of interest stores as Mexico and Brazil on market conditions.” -> this sentence is quite puzzling to me.

Missing citation for WebText.

“In Shane’s 2009 prediction” -> please add a corresponding citation.

In “Is Autoregressive Generation the Way to AGI?”, authors could discuss the emergence of discrete diffusion methods (e.g. [42]).

# References

[1] - The perceptron: A probabilistic model for information storage and organization in the brain, Rosenblatt, 1958

[2] - Steps Toward Artificial Intelligence, Minsky, 1961

[3] - ImageNet Classification with Deep Convolutional Neural Networks, Krizhevsky et al, 2012

[4] - Deep Q-Networks, Mnih et al, 2013

[5] - Attention is all you Need, Vaswani et al, 2017

[6] - Training language models to follow instructions with human feedback, Ouyang et al, 2022

[7] - Program of Thoughts Prompting, Chen et al, 2022

[8] - Large Language Models are Zero-Shot Reasoners, Kojima et al, 2022

[9] - ToRA: A Tool-Integrated Reasoning Agent for Mathematical Problem Solving, Gou et al,, 2023

[10] - More Agents Is All You Need, Li et al, 2024

[11] - Sparse Autoencoders Find Highly Interpretable Features in Language Models, Cunningham et al, 2023

[12] - Scaling and evaluating sparse autoencoders, Gao et al, 2024

[13] - Scaling Monosemanticity: Extracting Interpretable Features from Claude 3 Sonnet, Templeton et al, 2024

[14] - Toward Adaptive Reasoning in Large Language Models with Thought Rollback, Chen and Li, 2024

[15] - ReFT: Representation Finetuning for Language Models, Wu et al, 2024

[16] - Ring Attention with Blockwise Transformers for Near-Infinite Context, Liu et al, 2023

[17] - LongRoPE: Extending LLM Context Window Beyond 2 Million Tokens, Ding et al, 2024

[18] - Rt-2: Vision-language-action models transfer web knowledge to robotic control, Brohan et al 2023

[19] - Palm-e: An embodied multimodal language model, Driess et al, 2023

[20] - Mobile ALOHA: Learning Bimanual Mobile Manipulation with Low-Cost Whole-Body Teleoperation, Fu et al, 2024

[21] - On-Policy Distillation of Language Models: Learning from Self-Generated Mistakes, Agarwal et al, 2023

[22] - Griffin: Mixing Gated Linear Recurrences with Local Attention for Efficient Language Models, De at al, 2024

[23] - RecurrentGemma: Moving Past Transformers for Efficient Open Language Models, Botev et al, 2024

[24] - DiLoCo: Distributed Low-Communication Training of Language Models, Douillard et al, 2023

[25] - ODIN: Disentangled Reward Mitigates Hacking in RLHF, Chen et al, 2024

[26] - WARM: On the Benefits of Weight Averaged Reward Models, Rame et al, 2024

[27] - Reward Model Ensembles Help Mitigate Overoptimization, Coste et al, 2023

[28] - Helping or Herding? Reward Model Ensembles Mitigate but do not Eliminate Reward Hacking, Eisenstein et al, 2023

[29] - West-of-N: Synthetic Preference Generation for Improved Reward Modeling, Pace et al, 2024

[30] - Variational Best-of-N Alignment, Amini et al, 2024

[31] - BOND: Aligning LLMs with Best-of-N Distillation, Sessa et al, 2024

[32] - Nash Learning from Human Feedback, Munos et al, 2023

[33] - ReMax: A Simple, Effective, and Efficient Reinforcement Learning Method for Aligning Large Language Models, Li et al, 2023

[34] - Scaling Relationship on Learning Mathematical Reasoning with Large Language Models, Yuan et al, 2023

[35] - Llama 2: Open foundation and fine-tuned chat models, Touvron et al, 2023

[36] - WARP: On the Benefits of Weight Averaged Rewarded Policies, Rame et al, 2024

[37] - Conditioned Language Policy: A General Framework for Steerable Multi-Objective Finetuning, Wang et al, 2024

[38] - Direct Language Model Alignment from Online AI Feedback, Guo et al, 2024

[39] - Understanding the performance gap between online and offline alignment algorithms, Tang et al, 2024

[40] - SteerLM: Attribute Conditioned SFT as an (User-Steerable) Alternative to RLHF, Dong et al, 2023

[41] - Transformers are Sample-Efficient World Models, Micheli et al, 2022

[42] - Discrete Flow Matching, Gat et al, 2024

[43] - BoNBoN Alignment for Large Language Models and the Sweetness of Best-of-n Sampling, Gui et al, 2024

[44] - CLadder: Assessing Causal Reasoning in Language Models, Jin et al, 2023

[45] - Causal Parrots: Large Language Models May Talk Causality But Are Not Causal, Zecevic et al, 2023

**Strengths And Weaknesses:**

I focused my technical attention on section 5, which is the one I am most familiar with.

## Strengths
* holistic approach: covers a lot of ground on the multiple aspects of AGI research
* content on AGI systems: I liked the part on AGI systems and large scale training in general, lots of works referenced (the section on parallel computing is very useful)
* figures: Figures are quite useful and well-made
* recall: a lot of relevant work being cited (though some missing as raised)

## Weaknesses

I think writing is lacking currently: there is a lot of room for improvement on the clarity and organization of the ideas being presented. Section 5 is not in a great state yet.

### Structure

I think authors could share their definition / assessment of the different levels of AGI as one of the first points of the paper.

### Writing
* I think there are unnecessary/low value sentences throughout the paper (see examples above).
* Writing is a bit too grandiose sometimes (e.g. “AGI inference systems need to ensure user responsiveness, availability, and efficiency, which helps unleash the ultimate potential of large models from the training phase and revolutionize how users interact with the system.” in section 4.4)
* It feels like authors could refactor many parts to make them crisper (see examples above).
* Many citet / citep issues in the text, please fix.
* I think many parts are vague (see examples above) and would benefit from clearer statements.

### Content
* Section 5 is not satisfactory in the current state, and would clearly benefit from addressing the issues and comments I raised.
* There is a large variance in the number of citations among subtopics.

---

> ### Author Response · Authors · 2024-08-31
> **Response to Reviewer XGZH (1/7)**
>
> **Q1. [AI Perception: Current state of AI Perception]: Authors mention that multimodal LLMs are trained to comprehend other modalities than text from existing LLMs -> in principle they could cover multimodality from scratch.**
>
> **Response:** Thank you for your suggestions. In Section 2.1, "Perception," we've focused primarily on LLM-based AI perception, as this is the most prevalent direction today. Addressing multimodality from the ground up is a broader topic compared to evaluating the extent of AGI perception. At present, our main emphasis is on the current scope of multimodality, given that LLMs and related AI models are the most adept at both generation and understanding tasks.
>
> **Q2. [AI Perception: External Connection of Modalities]: Text is not that clear. A figure could help distinguish the different families of methods.**
>
> **Response:** Thank you for your feedback. We have added a figure (Figure 4 in the revised version) to illustrate this subsection more clearly.
>
> **Q3. [AI Perception: AGI level Perception]: No citation for Gemini (appears later on but could be placed here).**
>
> **Response:** Thanks for the reviewer’s discerning suggestion. We have added the citations in the paper.
>
> **Q4. [AI Reasoning]:  Missing citations for PoT [7], zero-shot CoT [8], tool-integrated reasoning [9], more agents is all you need [10]. The description of Maieutic Prompting is not clear as is. Authors could discuss the literature on causal understanding in LLMs - some works in that direction are [44 - 45]. "AGI should enhance social reasoning" -> change to "should get better at social reasoning" or similar. On explainability, recent works on Sparse Autoencoders should be mentioned: [11-13], and this whole line of research acknowledged in the text. "that allows reasoning planning" -> sentence missing an "and". On dynamic reasoning, seems like it is a vibrant area of research (see Thought Rollback [14], REFT [15] and others) which could be mentioned.**
>
> **Response:** Thank you for your suggestions. While we aim to provide an overview of the current state of reasoning to highlight the gap between existing capabilities and AGI, it’s quite challenging to cover all the components and perspectives. Nevertheless, we have followed the reviewer’s direction and incorporated all the suggested references into our paper.
>
> **Q5. [AI memory]: Authors could mention model merging as a way to integrate knowledge in model weights; On long-context and memory utilization, I think things like Ring Attention [16] and LongRoPE [17] could be mentioned.**
>
> **Response:** Thanks for the reviewer’s insightful comments. We address each concern as follows:
>
> **(1) Model merging:** We agree with the reviewer that model merging can be regarded as a way to integrate knowledge in model weights. Indeed, we have mentioned that other techniques like supervised fine-tuning or knowledge editing can integrate knowledge in model weights in our [Textual memory vs. parametric memory of Sec 2.3]. Following the reviewer’s suggestion, we have added a discussion on model merging on this basis, specifically:
>
> Techniques like supervised fine-tuning (Hu et al., 2021), knowledge editing (De Cao et al., 2021; Mitchell et al., 2021) and model merging (Du et al., 2024; Yang et al., 2024; Lu et al., 2024; Goddard et al., 2024) can integrate domain-specific knowledge into model parameters.
>
> **(2) Long-context and memory utilization:** Following the reviewer’s suggestion, we add the discussions about Ring Attention [16] and LongRoPE [17] in our [Long-context LLMs of Sec 2.3], specifically:
>
>  Works like Ring Attention (Liu et al., 2023i) and LongRoPE (Ding et al., 2024b) greatly reduce the time and cost of long context inference by improving the operation mechanism and storage method of attention. More powerful GPUs with enhanced memory capabilities and further breakthroughs in memory-efficient attention mechanisms
> (Dao et al., 2022; Tay et al., 2022), allowing the context window for pre-trained LLMs to increase from 1024 tokens in GPT-2 (Radford et al., 2019), to 8192 in GPT-4 (Achiam et al., 2023), and now exceeding 16K tokens.

---

> ### Author Response · Authors · 2024-08-31
> **Response to Reviewer XGZH (2/7)**
>
> **Q6. [AI metacognition]: On self-awareness in AGI, several notions are introduced (self-immersion etc) and could be attributed to previous works and / or explained.**
>
> **Response:**
> We appreciate the reviewer's insightful comment on the discussion of self-awareness in AGI. We have taken the following steps to address this feedback:
>
> **(1)** We have clarified and provided additional references for key concepts related to self-awareness in AGI In section 2.4 especially the ‘current state of the metacognition part’.
>
> **(2)** We have removed some less critical concepts, such as self-immersion, and emotion resilience of AI, to focus on the discussion's most relevant aspects of metacognition.
>
> **(3)** We have added several important references to strengthen the theoretical foundation of our work. These include:
>
> References:
>
> [1] Kross, E., & Ayduk, O. (2017). Self-distancing: Theory, research, and current directions. Advances in experimental social psychology, 55, 81-136.
>
> [2] Scassellati, B. (2002). Theory of mind for a humanoid robot. Autonomous Robots, 12, 13-24.
>
> [3] Dehaene, S., Lau, H., & Kouider, S. (2021). What is consciousness, and could machines have it? In Robotics, AI, and humanity: Science, ethics, and policy (pp. 43-56). Springer International Publishing.
>
> [4] Chella, A., Pipitone, A., Morin, A., & Racy, F. (2020). Developing self-awareness in robots via inner speech. Frontiers in Robotics and AI, 7, 16.
>
> [5] Subagdja, B., Tay, H. Y., & Tan, A. H. (2021). Who am I?: Towards social self-awareness for intelligent agents. International Joint Conferences on Artificial Intelligence.
>
> [6] Floreano, D., Mondada, F., Perez-Uribe, A., & Roggen, D. (2004). Machine self-evolution. YLEM journal, 24(12), 4-10.
>
> [7] Tao, Z., Lin, T. E., Chen, X., Li, H., Wu, Y., Li, Y., ... & Zhou, J. (2024). A survey on self-evolution of large language models. arXiv preprint arXiv:2404.14387.
>
> [8] Cuzzolin, F., Morelli, A., Cirstea, B., & Sahakian, B. J. (2020). Knowing me, knowing you: theory of mind in AI. Psychological medicine, 50(7), 1057-1061.
>
> [9] Shinn, N., Cassano, F., Gopinath, A., Narasimhan, K., & Yao, S. (2024). Reflexion: Language agents with verbal reinforcement learning. Advances in Neural Information Processing Systems, 36.
>
> [10] Langdon, A., Botvinick, M., Nakahara, H., Tanaka, K., Matsumoto, M., & Kanai, R. (2022). Meta-learning, social cognition and consciousness in brains and machines. Neural Networks, 145, 80-89.
>
> These additions provide a more comprehensive theoretical background for our discussion on self-awareness in AGI, addressing the reviewer's concern and strengthening the overall quality of our work.
>
>
>
>
>
> **Q7. [AGI Interface]: "Interface to Digital World provides an accessible, practical, and scalable testbed for studying advanced cognitive skills in agents, promoting their development through language-based inputs and the ease of developing digital tools" -> this sentence is strange and feels out of context?**
>
> **Response:** We sincerely appreciate the reviewer's insightful comment regarding the discussion of the AGI interface. We agree that the original sentence could be improved in specific ways. In response to this valuable feedback, we have thoroughly revised this section to better emphasize the role and necessity of the AGI interface to the digital world.
>
> In our revision, we have:
>
> **(1)** Removed the specific mention of "advanced cognitive skills" and "language-based inputs" to focus more on the broader implications of the interface.
>
> **(2)** Restructured the paragraph to highlight the interface's crucial role in grounding AGI in real-world scenarios.
>
> **(3)** Emphasized how the interface serves as an essential bridge between AGI and complex digital environments, facilitating more realistic development and evaluation.
>
> **(4)** Clarified the importance of this interface in accelerating AGI progress towards versatility and robustness across various domains.
>
> The revised paragraph now reads:
>
> "The AGI interface to the digital world significantly expands AGI development by enabling interaction with diverse digital environments such as the Internet, databases, and APIs . This interface serves as a crucial bridge for grounding AGI in complex, real-world scenarios, providing an indispensable platform for simulating and interacting with the multifaceted nature of human knowledge and experience. By facilitating AGI's engagement with real-world information structures and problem-solving contexts, this digital world interface accelerates the development of more versatile and robust artificial general intelligence capable of operating effectively across various domains."
>
> We believe this revision addresses the reviewer's concern by providing a more coherent and contextually relevant description of the AGI interface's role and significance in AGI development.

---

> ### Author Response · Authors · 2024-08-31
> **Response to Reviewer XGZH (3/7)**
>
> **Q8. [AI Interfaces to Physical World]: Google's PaLM -> PaLM (unnecessary company attribution in the context of a scientific paper). Missing, up-to-date citations on robotic control (Palm-E [19] / RT2 [18] / Mobile Aloha [20]). "The ability to interpret and adapt to dynamic, real-world environments in a manner akin to human cognition." -> this is not a proper sentence.**
>
> **Response:**  We sincerely appreciate the reviewer's careful examination and valuable feedback. We have addressed each point as follows:
>
>  **(1)** We have removed the unnecessary company attribution for PaLM.
>
>  **(2)** We have updated our citations to include the latest versions of PaLM-E [19], RT-2 [18], and Mobile ALOHA [20], ensuring our paper reflects the most current research in robotic control and embodied AI.
>
>  **(3)** We have revised the problematic sentence to read:
>
>  "AGI needs to develop the capacity to interpret and adapt to dynamic, real-world environments with a level of sophistication at least comparable to human cognition. This advanced environmental analysis and perception capability is essential for AGI to effectively navigate and interact with complex, ever-changing physical surroundings, mirroring the adaptability and intuition that humans exhibit in diverse situations."
>
> We believe these changes address the reviewer's concerns and strengthen the overall quality and accuracy of our paper. We are grateful for the opportunity to improve our work based on this valuable feedback.
>
> **Q9. [Scalable Model Architectures]: On self-attention mechanisms, not sure about the link between attention sparsity and the lottery ticket hypothesis? On model compression, online distillation (Agarwal et al) should be mentioned. "Beyond Transformers" -> Griffin / Hawk and subsequent integrations (i.e. RecurrentGemma) could be mentioned!**
>
> **Response:** We thank the reviewer for mentioning potentially unclear points discussed in the paper. Here are the explanations:
>
> **(1) “link between attention sparsity and the lottery ticket hypothesis”:** ( We think both lottery ticket hypothesis and sparsity in transformers (self-attention modules) are similarly motivated despite having quite different approaches to the same problem. Both of them aim to identify potential sparsity patterns that can improve both memory efficiency and computations without significant quality degradation. We added more detailed explanations in the revision.
>
> **(2) Recommended works:** We appreciate the mentioning of all the recommended works and we added them into the paper with proper focus and level of details.
>
> **Q10. [Large-scale Training]:  The description of LLAMA-Adapter is not particularly clear from the text. “linearly probing” -> “linear probing”; Decentralization:  DiLoCo (Douillard et al) could be mentioned; Data economy: Shapley values are mentioned but never explained, this part is not super clear at the moment.**
>
> **Response:** We appreciate reviewer’s feedback on this section. We made the following revisions:
>
>  **(1)** We introduced the concept of LLAMA-Adapter in more detail in the main paper section.
>
> **(2)** Fixed the typo of “linearly probing” -> “linear probing”
>
> **(3)** We believe Douillard et al is a great and orthogonal work to what we have mentioned and hence added that to the paper.
>
> **(4)** Similar to LLAMA-Adapter, we added more context for Shapley value in the paper.
>
> Our sincere gratitude for the detailed and constructive reviews.
>
> **Q11. [Computing Platforms]: Though it is nice to see where we stand in terms of potential for computation, I am not sure listing precise accelerator names is necessary? (e.g. NVIDIA DGX GH200, Qualcomm Cloud AI 100 Ultra). In Figure 6, the legend is a bit misleading: the arrows seem to indicate that “Model Copy” implies “Strong Connection”.**
>
> **Response:** Thanks for pointing out the potential room for improvement:
>
> **(1)** Although it could be more concise not to list these accelerators, we believe that as a survey, review, and position paper, it can still provide useful information for those who are less familiar with the topic. These accelerators are having very special properties and technological improvements that no other variants possess. For example, NVIDIA DGX GH200 creates a homogeneous memory space larger than 100TB, which allows for vastly different approaches to modern deep learning. Qualcomm Cloud AI 100 Ultra, on the other hand, aims to serve large models with extremely low-power consumption.
>
> **(2)**  Sorry for the misleading legends. We adjusted the legends to remove potential confusion.

---

> ### Author Response · Authors · 2024-08-31
> **Response to Reviewer XGZH (4/7)**
>
> **Q12. [Expectations of AGI Alignment]: I find the choice of approaches to AI alignment peculiar (in “To achieve this goal, researchers have proposed various approaches to AI alignment, such as value learning (Soares, 2016), inverse reinforcement learning (Hadfield-Menell et al., 2016), and cooperative inverse reinforcement learning (Hadfield-Menell et al., 2016)”): learning from preferences (and RLHF) should be mentioned given these are the default approaches used today. Also, this is not consistent with the rest of the section where mostly RLHF approaches are indeed presented.**
>
> **Response:** Thank you for your feedback. According to your suggestions, we added learning from preferences and RLHF into the introduction of the AGI alignment section in the section 5.1:
> To achieve this goal, researchers have proposed various approaches to AI alignment, such as value learning, inverse reinforcement learning, cooperative inverse reinforcement learning, and the most prevalent RLHF related strategies.
>
> **Q13. [Current Alignment Techniques]: “These methods can generate online human supervision using techniques such as reinforcement learning” -> reinforcement learning (RL) does not generate online supervision, it optimizes online supervision (for instance given by a reward model trained on preference data). Please rephrase to cleanly separate the contribution of RL from that of reward modeling.**
>
> **Response:** We apologize for making the confusion and thanks for the reviewer’s constructive feedback. We rephrase this sentence in our paper as: These methods can help LLMs align with online human feedback using techniques such as reinforcement learning or only inquiring about human supervision offline.
>
> **Q14. [Current Alignment Techniques]: “Online human supervision” seems to imply that there are humans-in-the-loop, which is not the case. “Online Alignment from (Approximate) Human Feedback” could be an alternative title.**
>
> **Response:**  Thanks for the insightful comments. We agree with the reviewer’s opinion and revise the corresponding text title and figure 8 in Sec 5.2.

---

> ### Author Response · Authors · 2024-08-31
> **Response to Reviewer XGZH (5/7)**
>
> **Q15. [Current Alignment Techniques]: I am not sure to agree with the three main directions proposed by the authors (i.e. rule optimization, data utilization and theoretical justification): 1. rule optimization seems like an important but also a minor aspect of the literature; 2. optimal data utilization (which we would call sample efficiency in the RL literature) is important but the works chosen to illustrate are not very representative; 3. on theoretical justification: most methods are theoretically motivated and are flavors of KL-regularized RL (both RL and direct preference optimization methods). I would argue in favor of four main topics instead: reward modeling, optimization, data and self-improvement.  Here are some works I would cite for each of these topics (which are not cited in the current state): 1. reward modeling: ODIN [25], WARM [26], Herding or Helping? [27], Reward model ensembles [28], West-of-N [29]; 2. optimization: NashLLM [32], ReMax [33], RFT [34], online iterated RLHF [35], WARP [36] data: OAIF [38]; 3. self-improvement (a lot of these are already mentioned). Currently the works on online human supervision are not highly representative of the state of the literature. Authors would do well discussing the gap between offline and online methods, and can rely on the following paper discussing the drawback of purely offline methods [39]. Text-based feedback signals: missing reference to SteerLM [40] and CLP [37]. Ranking-based feedback signals: missing reference to vBoN [30], BoNBoN [43] and BOND [31] (all flavors of distribution matching to the best-of-N distribution).**
>
> **Response:** Thank you for your feedback and suggestions. Our initial plan didn’t separate the reward modeling and optimization sections, but we agree that these two areas could be more clearly delineated. We have incorporated these essential references into our manuscript. Thanks to the valuable feedback from our reviewers, the revised section on alignment techniques now presents a more thorough and accurate reflection of the current literature in Section 5.2.
>
> **(1) Reward Modeling:** As the main supervision in the alignment process, the reward modeling is a crucial way to improve the alignment techniques. Sparrow incorporates adversarial probing and language-based rules into RLHF rewarding models. Bai et al. investigate using pure RL to provide online human-level supervision for LLMs training and provide detailed explorations of the tradeoffs between output helpfulness and harmlessness. Other techniques that unify both the reward and policy models have also emerged, which broadens directions for aligning AI models. Another direction focusing on mitigating reward hacking or overoptimization issues, by updated evaluation protocols, assembling multiple reward models, and refining the reward policy with synthetic data.
>
> **(2) Optimization:** How to incorporate the supervision, either the online or offline one is an open question worth exploring. For example, Cheng et al. optimize the reward model with the policy model simultaneously using min-max optimization. Recent works are exploring several alternatives for the RLHF method. DPO discards the reward model and optimizes for the final goal using the labelled preference in data. Muldrew et al. propose an improvement based on DPO with active learning strategy. NashLLM employs pairwise human feedback to train the policy model using Nash learning. ReMax removes the value model in conventional reinforcement algorithms and introduces a novel variance reduction technique for stabilizing the optimization.}
>
> **(3) Data:** Sensi tries to embed human value judgments into each step of language generation via a language model for both reward assignment (as the critic) and generation control (as the actor) during the generation. Baheti et al. focus on augmenting current training data by empowering different instances with varied weight for tuning models, taking the best advantage of the data w.r.t. their contribution to the language models. To ensure continuous, high-quality data, AI-generated instances are utilized for adapting to RLHF and, more recently, to DPO.
>
> **(4) Self-Improvement:** Strong AI models should learn how to improve themselves with or without external supervision. One of the most recent progress is the weak-to-strong generalization. For improving current RLHF-related methods, f-DPG is framed as a generalization of RLHF to use any f-divergence to approximate any target distribution that can be evaluated, which differentiates it from previous method that can only fit KL divergence in the process. Zhu et al. connect RLHF with max-entropy IRL and propose a unified paradigm for such a process with a sample complex bound for both situations.

---

> ### Author Response · Authors · 2024-08-31
> **Response to Reviewer XGZH (6/7)**
>
> **Q16. [Current Alignment Techniques]: On the topic of “Aligning with Interactive Supervisions”: 1. Title could be superalignment ?? or weak-to-strong supervision ? or scalable oversight) 2. “SPIN (Chen et al., 2024a) decomposes the training objective into progressively evolved goals and leverages the iterative self-improvement to improve strong models with a weak one” -> I do not agree with that sentence: SPIN is essentially about self-improvement (by increasing the log-likelihood of demonstrations and decreasing the log-likelihood of model completions), and its objective does not change along training. I would remove the mention of SPIN in the “Through task decomposition” paragraph.**
>
> **Response:** Thank you for your suggestions. We have updated the title to ‘scalable oversight’ according to your advice. In addition, we thank your suggestion for revising the SPIN paper, and we have removed it from the ‘task decomposition’ section.
>
> **Q17. [How to Approach AGI Alignment]: I would put the paragraph on “Interaction with tools and APIs” after “Interaction with humans”. Interactions with tools and APIs: “AGI cannot use APIs or tools to cause crimes during the interaction process (Zhang et al., 2024c; Yao et al., 2024; Chen et al., 2023a)” -> I think AGI should also keep to an absolute minimum potential damage overall (which includes mental harm done to humans, general harm to other species and even valuable goods – this is not mentioned in the current state). In the “vision of the future in alignment techniques”, additionally to the topics mentioned I would put emphasis on safety and weak-to-strong alignment (i.e. scalable oversight).**
>
> **Response:** We have revised the related sections by changing the order of these paragraphs. We would like to thank the reviewer for pointing out that the damage of AGI should be minimized as much as possible. We have emphasized this point in our current version. Meanwhile, we have also included more alignment-related topics other than RLHF in this section, such as scalable oversight to make our paper more comprehensive and representative of alignment research.
>
> **Q18. [Current Alignment Techniques]: Include missing references for: Text-based feedback signals: Add SteerLM and CLP; Ranking-based feedback signals: Include vBoN, BoNBoN, and BOND.**
>
> **Response:** Following the reviewer’s suggestions, we have added these missing references in the text-based feedback signal and the ranking-based feedback signal parts, which made our paper more comprehensive.
>
> **Q19. [AGI Levels]: I would consider presenting these much earlier in the paper.**
>
> **Response:** We appreciate the reviewer's insightful suggestion regarding the placement of AGI levels in our paper. We agree that introducing these concepts earlier can provide readers with a clearer framework for understanding the subsequent discussions. In response to this valuable feedback, we have added a brief mention of the AGI levels in the last paragraph of the Introduction section.
>
> **Q20. [AGI Evaluation]: In “Super-evaluation”, I would mention the existence of formal proof systems (e.g. Lean) that could be used to verify superhuman mathematical outputs.**
>
> **Response:** Thank you for your insightful comment regarding the "Super-evaluation". We acknowledge that systems like Lean are capable of rigorously verifying mathematical theorems, even when they require extensive computations. This capability is crucial when AI models surpass human expertise, especially in domains such as theorem proving.
>
> We have revised the manuscript to include a discussion on formal proof systems, emphasizing how they could be utilized to verify the outputs of AI models that achieve superhuman performance. This addition will enhance the paragraph of the "Super-evaluation" framework by considering tools like Lean that can independently confirm the correctness of AI-generated mathematical proofs. For your convenience, we have put the revised text as below:
>
> “
> Super-evaluation. Similar to super alignment, … Fortunately, formal proof systems such as Lean(De Moura et al., 2015) may be helpful. Lean is an interactive proof system that utilizes formal logic to verify the correctness of mathematical theorems and computational outputs. As AI models begin to generate results that exceed human verification capabilities, systems like Lean become indispensable for ensuring the validity of these outputs.
> ”
>
> [De Moura et al., 2015]: Leonardo De Moura, Soonho Kong, Jeremy Avigad, Floris Van Doorn, and Jakob von Raumer. 2015. The Lean theorem prover (system description). In Automated Deduction-CADE-25: 25th International Conference on Automated Deduction, Berlin, Germany, August 1-7, 2015, Proceedings 25. Springer, 378–388.

---

> > ### Author Response · Authors · 2024-08-31
> > **Response to Reviewer XGZH (7/7)**
> >
> > **Q21. [How to Get to Next AGI Level]: The entire paragraph on “advanced code machines” (“Advanced code machine brings the ultimate AGI within easy reach”) is unintelligible to me. What such an advanced code machine is should be explained and the whole paragraph clarified. Authors mention VAEs and GANs as potential world modeling tools, but could also mention diffusion models, and also Transformer-based world models such as in [41].**
> >
> > **Response:** I understand your confusion regarding the "advanced code machine" paragraph. We've made some changes to clarify the content:
> >
> > **(1)** We've adjusted the title to make it less sensational while still conveying the key idea. The new title is: "Automated Coding AI: Bridging the Gap to the Ultimate Artificial Intelligence". We've replaced "code machine" with "coding AI" for better clarity. A coding AI refers to an AI system capable of automatically planning and using code to complete tasks. For a more comprehensive explanation of coding AI, please refer to Section 7.5 of the paper, where you'll find detailed information on this technology. These changes should make the paragraph more intelligible and provide a clearer context for the concept of coding AI. The revised content emphasizes the role of automated coding AI as a significant step towards more advanced artificial intelligence, without overstating its immediate impact.
> >
> > **(2)** We appreciate the reviewer's insightful suggestion about world modeling methods and have updated our paper accordingly. We have added mentions of both diffusion models and Transformer-based world models (citing [41]) in the relevant section, providing a more comprehensive overview of current world modeling techniques in AI.
> >
> > **Q22. [Further considerations]: In 1951, Shannon et al. proposed 3-gram to point to ninety-nine point six billion dollars from two hundred four oh six three percent of the rates of interest stores as Mexico and Brazil on market conditions.” -> this sentence is quite puzzling to me; Missing citation for WebText; “In Shane’s 2009 prediction” -> please add a corresponding citation; In “Is Autoregressive Generation the Way to AGI?”, authors could discuss the emergence of discrete diffusion methods (e.g. [42]).**
> >
> > **Response:** We sincerely appreciate the reviewer's valuable feedback. We have addressed each point as follows:
> >
> > **(1) Quite puzzling sentence, missing citation for WebText and adding a corresponding citation:** Thank you for your valuable insights. We appreciate the points raised and have attempted to address each thoroughly. Unfortunately, regarding the specific issue mentioned, we note that it does not directly appear in our submission or relate to the content discussed in our manuscript. Could you please provide us with further comments on it?
> >
> > **(2) Discussions on emergence of discrete diffusion methods:** We add a discussion about discrete diffusion methods at the end of this section to more fully discuss the expectations and concerns of autoregressive generation as a path to AGI. The detail is as follows:
> >
> > Additionally, the popularity of the diffusion model (Gat et al., 2024) poses challenges to the future of autoregressive generation. This type of method does not rely on previously generated data points during the generation process, but rather relies on the process of gradually reducing noise to recover the data. The remarkable effect of the diffusion model in generation tasks has also led to its widespread use in real-world applications (Yang et al., 2023f; Chen et al., 2024b). All of this makes whether autoregressive generation is the way to AGI an ongoing debate.

---

### Review · Reviewer_wAgx · 2024-08-08

**Summary Of Contributions:**

The paper "How Far Are We From AGI?" offers a comprehensive review of the current state of LLMs and potential future trajectory towards AGI.

The paper provides a detailed capability framework for AGI, integrating internal, interface, and system dimensions.

A significant contribution is the discussion on AGI alignment technologies. The authors highlight the necessity of aligning AGI systems with human values and safety constraints.

The paper introduces a structured approach to evaluating AGI progress, defining levels of AGI progression and proposing an evaluation framework.

The inclusion of case studies on diverse applications such as science discovery, generative visual intelligence, world models, decentralized AI, AI for coding, embodied AI, and human-AI collaboration provides tangible insights into the current challenges and potential pathways towards AGI in various domains.

The paper emphasizes the importance of responsible AGI development, advocating for ethical considerations and broader public discussions among researchers and practitioners. This focus on responsibility ensures that the pursuit of AGI considers the broader societal implications.

**Audience:**

Yes

**Claims And Evidence:**

Yes

**Requested Changes:**

1. As a minor problem, the paragraph “Future AGI will efficiently ...” is repeated twice. It might be an editing error. (Critical)
2. Give more arguments on Figure 1. The quantification should not be too arbitrary. (Critical)
3. Give more words to the definitions of AGI, and claim what definition(s) the authors adopt for the following survey/discussion. (Critical)
4. Survey other works/projects towards AGI other than LLMs. Otherwise, the title is more appropriate to be something like "How Far Are We From AGI: From the Perspective of LLMs". Nevertheless, I understand this paper can represent a group of peoples views. So it's fine to keep the current scope of discussion. If the authors add other AGI works (except LLMs) to the paper, the evaluation part should also be extended to cover more previous works on evaluating AGI. (Simply strengthen the work)
5. Add an explaination about the meaning of "reasoning", and keep it coherent through out the paper. (Simply strengthen the work)

**Strengths And Weaknesses:**

The paper encompasses a comprehensive survey of Large Language Models (LLMs), majorly in Sections 2 through 5; in each of the sections, the authors organize literatures elaborately, which are mostly related to LLMs, and then imagine the future directions along existing routes. There is no doubt that this paper is a great guide for recent progress. I am quite impressed by the breadth of the author's knowledge.

In Section 6, “‘How Far Are We from AGI’ Workshop Discussions” is valuable, because it presents the understandings of the term AGI from well-known AI researchers. Those understandings are representative, though not fully correct from my view.

With my respect for authors and admiration for their efforts, there are some points that I disagree with. In spite of my personal perspectives on AGI, I will focus on the points that are directly related to the flaws of this paper.

First, Figure 1 is too subjective and lacks enough arguments. Figure 1 estimates the progress of AI, giving each phenomenal work a percentage, indicating “the proportion of human activities surpassed by AI”. The numbers make sense only when the quantification method is provided. In addition, one famous metaphor is that monkeys climbing treetops will do little to help the moon landings. To what extent the current achievements in narrow AI facilitate AGI is controversial. Thus, I do not think it is a strong argument for “the realization of AGI is getting closer”.

Second, what AGI means, or the definition of AGI, is actually the foundation of all discussions of AGI, and the same goes for this paper. The paper discusses this most fundamental and critical issue with some short paragraphs. To answer the question of “how far are we from AGI”, the question “what is meant by AGI” should be answered or discussed seriously. At least, the authors need to make it very clear what definition(s) of AGI they adopt as the basis of later discussion and survey, and why they adopt it/them.

Third, I do not deny that LLM is one potential route to AGI, however, there are also plenty of other works and projects published in AGI conferences or other related conferences and journals, some of which directly claim that they aim for AGI, and some (especially those from psychology, neuroscience, brain science, etc) do not. I strongly believe those works are worthwhile and necessary to support such a big title “How far are we from AGI”.

Fourth, the AGI evaluation in Sec. 6.2 seems to almost refer to the evaluation of LLMs. AGI evaluation is a long-standing issue that has not been solved, and one reason is that evaluation depends on definition, while the disagreements among definitions cannot be settled in the near future. Nonetheless, there are many previous works on evaluating AGI, though not in the sense of LLMs (For example, [1-6]). I would like to remind the authors of these works and think over the title of Sec. 6.2.

Finally, in Sec. 2.2, the authors seem to use the concept of reasoning as “the process of thinking logically and systematically, drawing on evidence and past experiences to form conclusions or make decisions”. However, in the following description, the term “reasoning” is not used strictly in that way. For example, LLMs do not do logical reasoning, but it does statistical reasoning, thus, “thinking logically” in this definition does not apply to them. I would suggest the authors make a note or explanation about it.

BTW, an interesting observation is that so many famous people had predicted when the “real” AI (whatever it is called) would occur, but no one succeeded. Nevertheless, learning from existing attempts/works/research is an effective and efficient strategy of human intelligence to solve problems, even for the problem of AGI.

I hope these comments can help the authors to improve the paper and obtain a broader view of AGI.

---

References

[1] Adams, S.S., et al.: I-athlon: towards a multidimensional turing test. AI Mag. 37(1), 78–84 (2016)

[2] Goertzel, B., Bugaj, S.V.: AGI preschool: a framework for evaluating early-stage human-like AGIs. In: Proceedings of AGI, vol. 9, pp. 31–36 (2009)

[3] Wang, P.: The evaluation of AGI systems. In: Proceedings of the Third Conference on Artificial General Intelligence, vol. 11, pp. 164–169. Citeseer (2010)

[4] Wray, R., Lebiere, C.: Metrics for cognitive architecture evaluation. In: Proceedings of the AAAI-07 Workshop on Evaluating Architectures for Intelligence, pp. 60–66 (2007)

[5] Xu, B., & Ren, Q. (2022, August). Artificial Open World for Evaluating AGI: A Conceptual Design. In International Conference on Artificial General Intelligence (pp. 452-463). Cham: Springer International Publishing.

[6] Peng, Y., et al. (2024). The tong test: Evaluating artificial general intelligence through dynamic embodied physical and social interactions. Engineering, 34, 12-22.

---

> ### Author Response · Authors · 2024-08-31
> **Response to Reviewer wAgx (1/3)**
>
> **Q1. The AGI evaluation in Sec. 6.2 seems to almost refer to the evaluation of LLMs. AGI evaluation is a long-standing issue that has not been solved, and one reason is that evaluation depends on definition, while the disagreements among definitions cannot be settled in the near future. Nonetheless, there are many previous works on evaluating AGI, though not in the sense of LLMs (For example, [1-6]). I would like to remind the authors of these works and think over the title of Sec. 6.2.**
>
> **Response:** We appreciate the your insightful feedback regarding the AGI evaluation in Section 6.2. We would like to clarify that our paper does not solely focus on the evaluation of LLMs. For instance, our Interface-level evaluation discusses multiple modalities, including vision, language, and sensory data. While we have emphasized LLMs due to their prominence in recent AGI developments, we recognize the importance of a broader perspective on AGI evaluation.
>
> Thanks for your useful suggestion, we have added the recommended references to our existing discussion. These include works that address the limitations of the Turing test and propose more comprehensive evaluation frameworks, such as the I-athlon [1] for multidimensional testing of AI systems, the AGI Preschool framework [2] for early-stage AGI, and other significant contributions on evaluating AGI systems [3], as well as works focusing on cognitive architecture [4], Open World [5], and dynamic embodied physical and social interactions [6]. These additions ensure that our discussion now reflects a more holistic view of AGI evaluation, considering both LLMs and broader aspects of AGI research and development. For your convenience, we have put the added discussion about limitation of Turing Test as below.
>
> “
> …
>
> This level of AGI usually performs better or on par with humans at specific benchmark tasks[2]. Level-1 AGI represents the current state-of-the-art AI systems.
>
> …
>
> The concept of evaluating AGI traces back to the famous Turing Test proposed by Alan Turing in … However, the Turing test has several drawbacks, such as its reliance on deception, subjective evaluation, and narrow focus on language use. To address these issues, a more comprehensive approach called the I-athlon[1] has been proposed, which evaluates machine intelligence across multiple dimensions and aims to provide a more objective and practical method for assessing progress in general-purpose AI.
> …
>
> These tests should encompass a broader range of cognitive abilities[4] and evaluate the AI's performance in real-world scenarios.
>
> …
>
> Examine the AI's overall behavior and its ability to pursue long-term goals, collaborate with humans and other AI systems[3], and make ethical decisions. This could involve scenarios that assess the AI's alignment with human values, its transparency and explainability, and its robustness and reliability in uncertain and adversarial situations.
> …
>
> Here we list a couple of challenges associated with AGI evaluation[5] design,
> …
>
> Often, the challenge associated with these tasks also comes with the difficulty of applying a single metric for comparison and can usually encounter many physical constraints[6] that are hard to overcome.
> …
>
> ”
>
> [1] Adams, S.S., et al.: I-athlon: towards a multidimensional turing test. AI Mag. 37(1), 78–84 (2016).
>
> [2] Goertzel, B., Bugaj, S.V.: AGI preschool: a framework for evaluating early-stage human-like AGIs. In: Proceedings of AGI, vol. 9, pp. 31–36 (2009).
>
> [3] Wang, P.: The evaluation of AGI systems. In: Proceedings of the Third Conference on Artificial General Intelligence, vol. 11, pp. 164–169. Citeseer (2010)
>
> [4] Wray, R., Lebiere, C.: Metrics for cognitive architecture evaluation. In: Proceedings of the AAAI-07 Workshop on Evaluating Architectures for Intelligence, pp. 60–66 (2007)
>
> [5] Xu, B., & Ren, Q. (2022, August). Artificial Open World for Evaluating AGI: A Conceptual Design. In International Conference on Artificial General Intelligence (pp. 452-463). Cham: Springer International Publishing.
> [6] Peng, Y., et al. (2024). The tong test: Evaluating artificial general intelligence through dynamic embodied physical and social interactions. Engineering, 34, 12-22.

---

> > ### Author Response · Authors · 2024-08-31
> > **Response to Reviewer wAgx (2/3)**
> >
> > **Q2. In Sec. 2.2, the authors seem to use the concept of reasoning as “the process of thinking logically and systematically, drawing on evidence and past experiences to form conclusions or make decisions”. However, in the following description, the term “reasoning” is not used strictly in that way. For example, LLMs do not do logical reasoning, but it does statistical reasoning, thus, “thinking logically” in this definition does not apply to them. I would suggest the authors make a note or explanation about it.
> > Add an explaination about the meaning of "reasoning", and keep it coherent through out the paper. (Simply strengthen the work)**
> >
> > **Response:**  Thank you for pointing out that the reasoning in AI models does not align strictly with traditional logical reasoning; instead, it refers to the statistical reasoning, which relies more on data-driven insights rather than logical deduction. We have accordingly expanded the definition of reasoning in the paper and keep it coherent through out the paper. Additionally, we have noted in the paper that AI reasoning refers to the ability of AI systems to simulate the human reasoning process, enabling machines to understand situations, infer conclusions, and make decisions in a way that mimics human reasoning.
> >
> > **Q3. As a minor problem, the paragraph “Future AGI will efficiently ...” is repeated twice. It might be an editing error. (Critical)**
> >
> > **Response:** Thanks for the reviewer’s insightful suggestion. We have revised this in our current version.
> >
> > **Q4. Figure 1 is too subjective and lacks enough arguments. Figure 1 estimates the progress of AI, giving each phenomenal work a percentage, indicating “the proportion of human activities surpassed by AI”. The numbers make sense only when the quantification method is provided. In addition, one famous metaphor is that monkeys climbing treetops will do little to help the moon landings. To what extent the current achievements in narrow AI facilitate AGI is controversial. Thus, I do not think it is a strong argument for “the realization of AGI is getting closer”. Give more arguments on Figure 1. The quantification should not be too arbitrary. (Critical)**
> >
> >
> > **Response:**  Thank you for your insightful feedback. We have already detailed our quantification method in the caption of Figure 1: We estimate the (cumulative) percentage (on a logarithmic scale) of human activities where AI has surpassed human competence and efficiency, tracing developments from early embryonic systems in the 1970s to more sophisticated recent advancements. We acknowledge the challenge you highlighted in developing precise metrics to measure AI's proximity to AGI. Current literature [Wang et al. (2018), Voss and Jovanovic (2023), Bugaj and Goertzel (2007)] tends to evaluate AGI development using subjective and qualitative methods. In contrast, our work tries to introduce a unique quantitative perspective to assess the AGI process. We hope this approach and the new perspective it offers will serve as valuable references for future research in the field.
> >
> > **Q5. What AGI means, or the definition of AGI, is actually the foundation of all discussions of AGI, and the same goes for this paper. The paper discusses this most fundamental and critical issue with some short paragraphs. To answer the question of “how far are we from AGI”, the question “what is meant by AGI” should be answered or discussed seriously. At least, the authors need to make it very clear what definition(s) of AGI they adopt as the basis of later discussion and survey, and why they adopt it/them. Give more words to the definitions of AGI, and claim what definition(s) the authors adopt for the following survey/discussion. (Critical)**
> >
> > **Response:** Thanks for the reviewer’s insightful comment. In fact, we have discussed the existing work on the definition of AGI in the "Craving for General-purpose AI" section of the introduction. Due to the somewhat vague and broad nature of the existing definitions, we provided specific definitions and criteria for different developmental stages of AGI in Section 6.1. Our paper further discusses the current progress of AGI and its distance from the future based on the definitions we proposed.

---

> > > ### Author Response · Authors · 2024-08-31
> > > **Response to Reviewer wAgx (3/3)**
> > >
> > > **Q6. I do not deny that LLM is one potential route to AGI, however, there are also plenty of other works and projects published in AGI conferences or other related conferences and journals, some of which directly claim that they aim for AGI, and some (especially those from psychology, neuroscience, brain science, etc) do not. I strongly believe those works are worthwhile and necessary to support such a big title “How far are we from AGI”. Survey other works/projects towards AGI other than LLMs. Otherwise, the title is more appropriate to be something like "How Far Are We From AGI: From the Perspective of LLMs". Nevertheless, I understand this paper can represent a group of peoples views. So it's fine to keep the current scope of discussion. If the authors add other AGI works (except LLMs) to the paper, the evaluation part should also be extended to cover more previous works on evaluating AGI. (Simply strengthen the work)**
> > >
> > > **Response:**  Thank you for your valuable feedback. We would like to clarify that our paper does not solely focus on the evaluation of LLMs. In Section 6.2, for example, (1) our Interface-level evaluation covers multiple modalities, such as vision, language, and sensory data, beyond just LLMs; (2) we start the story with Turing test; (3) We also talk about alpha-Go at super-evaluation and etc. Indeed, we emphasize LLMs due to their prominence in recent AGI developments and the significant progress observed in this area. We fully acknowledge the importance of including a broader perspective on AGI evaluation.
> > > In response to your suggestion, we have strengthened our discussion by incorporating more works related to AGI beyond LLMs. These additions include references to works that critique the Turing test and propose more comprehensive evaluation frameworks, such as the I-athlon [1], which suggests a multidimensional approach to testing AI systems, and the AGI Preschool framework [2], which focuses on evaluating early-stage AGI development. Additionally, we have included other notable works on AGI evaluation that address cognitive architectures [3], dynamic embodied physical and social interactions [4], and more general AGI system evaluation methodologies [5].
> > > We believe these additions provide a more balanced view of the various approaches and projects aimed at achieving AGI. These works illustrate the diverse methodologies and perspectives across fields like psychology, neuroscience, and brain science that contribute to the AGI discourse. This expanded scope aligns with your suggestion and helps support the title of our paper, "How Far Are We From AGI," by incorporating a wider range of approaches and evaluations in the field.
> > >
> > > [1] Adams, S.S., et al.: I-athlon: towards a multidimensional turing test. AI Mag. 37(1), 78–84 (2016).
> > >
> > > [2] Goertzel, B., Bugaj, S.V.: AGI preschool: a framework for evaluating early-stage human-like AGIs. In: Proceedings of AGI, vol. 9, pp. 31–36 (2009).
> > >
> > > [3] Wang, P.: The evaluation of AGI systems. In: Proceedings of the Third Conference on Artificial General Intelligence, vol. 11, pp. 164–169. Citeseer (2010)
> > >
> > > [4] Wray, R., Lebiere, C.: Metrics for cognitive architecture evaluation. In: Proceedings of the AAAI-07 Workshop on Evaluating Architectures for Intelligence, pp. 60–66 (2007)
> > >
> > > [5] Peng, Y., et al. (2024). The tong test: Evaluating artificial general intelligence through dynamic embodied physical and social interactions. Engineering, 34, 12-22.

---

> ### Comment · Reviewer_wAgx · 2024-09-19
>
> I don't think the the quantification in Figure 1 is clear enough. To me, the descriptions "we estimate the (cumulative) percentage (on a logarithmic scale) of human activities where AI has surpassed human competence and efficiency" and "The statistics presented here are calculated based on our heuristic estimations" are confusing, providing few information on how they are calculated. I knew the previous works did not provide regid arguments on the quantification, but that is not the reason to continue this casual manner of scientific research. Before publishing, I would suggest the authors to make it clearer.

---

> ### Comment · Reviewer_wAgx · 2024-09-19
>
> Thanks for the authors' responses. They address most of my concerns.

---

### Review · Reviewer_HX9c · 2024-08-17

**Summary Of Contributions:**

This paper explores the current state and possible future trajectory of AGI, listing key questions about how close we are to achieving AGI.  The paper outlines essential capability frameworks, discusses necessary alignment issues, and discuss multiple case studies.

**Audience:**

Yes

**Claims And Evidence:**

No

**Requested Changes:**

See above.

**Strengths And Weaknesses:**

The authors have done an excellent job of providing a detailed summary of large language models (LLMs). The paper reads more like exploring how far LLMs are from achieving Artificial General Intelligence (AGI) rather than a discussion on generalized models. The authors hold strong positions on various aspects of reasoning, cognition, and metacognition, and I thoroughly enjoyed reading these perspectives. However, I find myself not fully aligned with some of their views. I might have been more persuaded if the paper included technical discussions on how to achieve the positions the authors advocate for, as these discussions seem heavily missing.

I understand the position that LLMs could potentially lead to AGI, but it would be beneficial to also discuss other fields and how they fit into this context, given the plethora of works aiming for AGI.

Here are my comments, categorized by section:

## Perception:
The authors do a great job in outlining all the existing literature related to AGI perception. However, the potential future directions seem somewhat weak, as the paper mainly lists existing contributions without presenting strong positions from the authors' viewpoints. It would be helpful and interesting if the authors could provide detailed steps or suggestions for diversification, robustness, or explainability in AGI perception.

## Reasoning:
The reasoning section covers many methods in NLP and provides an in-depth discussion. Since the authors discuss Chain of Thought (CoT), Graph of Thought (GoT), self-consistency, and dynamic reasoning only from the LLM perspective, I believe that including a discussion on these properties from the viewpoint of causal and neurosymbolic models would be an interesting addition. Additionally, while the authors address the challenges in achieving AGI-level reasoning, a paragraph on potential ways to tackle these challenges would be useful.

## Memory:
The statement "Future AGI will enhance memory utilization through the integration of retrieval and advanced reasoning" is intriguing, but could the authors clarify what "advanced reasoning" corresponds to in this context? In the discussion on memory updates with safety constraints, it would be helpful to include a discussion on how to design these safety constraints and how they can be implemented in practice.

## Interface:
I appreciate the inclusion of Figure 4. The AGI interface is discussed from multiple modalities, and the section also explores various value propositions and integrations of AGI.

## Alignment:
The alignment section focuses solely on Reinforcement Learning from Human Feedback (RLHF) methods. Including methods such as contestable AI, debates, and argumentative exchanges could provide a more comprehensive view.

## Conclusion:
In conclusion, the paper breaks down the authors' notion of AGI with some case studies but primarily focuses on surveying LLMs and listing challenges that exist in achieving more intelligent machines. However, the paper does not provide any technical insights into modelling and addressing these challenges, making it feel somewhat superficial.

---

> ### Author Response · Authors · 2024-08-31
> **Response to Reviewer HX9c (1/2)**
>
> **Q1. [Perception]: The authors do a great job in outlining all the existing literature related to AGI perception. However, the potential future directions seem somewhat weak, as the paper mainly lists existing contributions without presenting strong positions from the authors' viewpoints. It would be helpful and interesting if the authors could provide detailed steps or suggestions for diversification, robustness, or explainability in AGI perception.**
>
> **Response:**  Thank you for your suggestions. In the original section 2.1, built the future direction framework by listing three directions — diversification, robustness, and explainability. As for detailed steps for future development, we have added paragraphs to look into future research avenues. We provide the detailed solutions below:
>
> **For diversification:** Potential methods for incorporating other modal perceptions: a unified modal representation tool like ImageBind, LangaugeBind could bridge the modal gap and lessen the burden of learning from other modalities. Existing models that incorporate these tools have shown promising results in efficiency and task performance
>
> **For robustness:** Future researches could benefit from incorporating adversarial examples into training or involving increased diversity of training data instruction formats
>
> **For explainability in AGI perception:** For example, future research avenues could explore strict controlled experiments for training AI models to decompose each part or probing model components to find the most effective module.
>
> **Q2. [Reasoning]: The reasoning section covers many methods in NLP and provides an in-depth discussion. Since the authors discuss Chain of Thought (CoT), Graph of Thought (GoT), self-consistency, and dynamic reasoning only from the LLM perspective, I believe that including a discussion on these properties from the viewpoint of causal and neurosymbolic models would be an interesting addition. Additionally, while the authors address the challenges in achieving AGI-level reasoning, a paragraph on potential ways to tackle these challenges would be useful.**
>
> **Response:**
> Thank you for your suggestions. We have incorporated discussions on causal and neurosymbolic models. While addressing the challenges in achieving AGI-level reasoning, we have also included notes on potential solutions directly following the paragraphs for each challenge.
>
> **Q3. [Memory]: The statement "Future AGI will enhance memory utilization through the integration of retrieval and advanced reasoning" is intriguing, but could the authors clarify what "advanced reasoning" corresponds to in this context? In the discussion on memory updates with safety constraints, it would be helpful to include a discussion on how to design these safety constraints and how they can be implemented in practice.**
>
> **Response:** Thank you for your suggestions. We propose that instead of treating the memory retrieval process and reasoning module as separate components, they can be more effectively integrated into a unified system that enhances both efficiency and performance. "Advanced reasoning" refers to a sophisticated integration of retrieval processes with reasoning that strategically synthesizes and applies information in ways that are contextually appropriate. We have refined our explanation to make this clearer, and we discuss the expectations of advanced reasoning in more detail in Section 2.2.
> Since our paper primarily addresses the question, “How far are we from AGI?”, we focus on discussing the current state and the gap between existing capabilities and AGI. While we include some predictions on future trends, detailed future design and implementation is somewhat speculative and beyond the scope of this paper. However, in line with the reviewer's suggestion, we have added an outline on how to design these safety constraints and implement them in practice. Designing safety constraints for autonomous AGIs will involve creating robust validation protocols that assess the truthfulness, relevance, and impact of new information before integration. These could include the implementation of expert systems for periodic review of updates, anomaly detection to flag outliers and potentially harmful data, and additional methods to enforce these safety constraints.
>
> **Q4. [Interface]: I appreciate the inclusion of Figure 4. The AGI interface is discussed from multiple modalities, and the section also explores various value propositions and integrations of AGI.**
>
> **Response:**  We appreciate your insightful feedback and are pleased that the discussion on the AGI interface resonated with you.

---

> > ### Author Response · Authors · 2024-08-31
> > **Response to Reviewer HX9c (2/2)**
> >
> > **Q5. [Alignment]: The alignment section focuses solely on Reinforcement Learning from Human Feedback (RLHF) methods. Including methods such as contestable AI, debates, and argumentative exchanges could provide a more comprehensive view.**
> >
> > **Response:** Thank you for your meticulous review. In Section 5.2, we have included constitutional AI (Through human-written principles subsection) debate (Through model interactions subsection) in the paper, which consists of the whole alignment technique framework along with RLHF-related strategies. Also note that, RLHF as one of most prominent methods for aligning AI models nowadays, has way more related works. That’s why we emphasize such a strategy in the section.
> >
> > **Q6. I understand the position that LLMs could potentially lead to AGI, but it would be beneficial to also discuss other fields and how they fit into this context, given the plethora of works aiming for AGI. In conclusion, the paper breaks down the authors' notion of AGI with some case studies but primarily focuses on surveying LLMs and listing challenges that exist in achieving more intelligent machines. However, the paper does not provide any technical insights into modelling and addressing these challenges, making it feel somewhat superficial.**
> >
> > **Response:** Thank you for your insightful feedback. We address your concerns one by one:
> >
> > **(1) Discussions beyond LLM:** We wish to clarify that our paper extends beyond merely discussing LLMs. In fact, in our submission, we discussed advances in many other areas of AI and their potential impact on AGI. For instance, in Section 6.2, our evaluation at the interface level encompasses various modalities, including vision, language, and sensory data, not limited to LLMs alone. Besides, in Sec 6.4 and 7, we discussed the future development of AGI and some specific examples from many fields of AI. We have highlighted LLMs due to their salience in recent advancements toward AGI and the notable progress achieved in this domain. Additionally, following the reviewer's suggestions, we have expanded our discussion of various other AGI domains in the revised paper. For details, please refer to the responses to questions Q1 and Q6 of  Reviewer wAgx.
> >
> > **(2) Discussion on solving challenges:** In fact, we have detailed some possible approaches to addressing AGI challenges at different stages in Section 6.3 of our paper. Additionally, in Section 6.4, we propose solutions and plans for some specific challenges. However, how to tackle these challenges and achieve AGI is not the main focus of our article. As human society continues to evolve, AI technology also changes over time, making the realization of AGI a topic that is both progressive and difficult to answer definitively. In our paper, we primarily focus on the current state of AGI development and envision future scenarios for AGI. This helps provide references for the ongoing development of AI and outlines directions, which is more pragmatically significant.

---

### Author Response · Authors · 2024-08-31
**General Response**

We are grateful for the detailed and thoughtful feedback provided by the reviewers, which acknowledges the breadth and depth of our exploration into the current state and potential future trajectory of Artificial General Intelligence (AGI). They appreciated our comprehensive review and the establishment of a capability framework that integrates internal, interface, and system dimensions to assess the progression towards AGI. The reviewers also highlighted the significance of our discussions on AGI alignment technologies, emphasizing the importance of aligning AGI systems with human values and safety constraints. Additionally, our paper's structured approach to evaluating AGI progress and the inclusion of diverse case studies were noted as strengths that contribute valuable insights into overcoming current challenges and forging pathways toward AGI. Such recognition reinforces the potential of our work to advance the understanding and development of AGI, encouraging broader discussion and ethical consideration in the field.

In response to the concerns raised, we have revised the manuscript (with revisions marked in blue) to more clearly differentiate our research from prior AGI studies, **highlighting our unique contribution**: offering a comprehensive analysis of the current state of AGI research and evaluating the remaining distance to future AGI achievements in a holistic manner. We have clarified, through examples in the text, that **our paper discusses various aspects of AGI development beyond large language models (LLMs)**. Following the reviewers' suggestions, **we have included extensive discussions of related work**. Additionally, we have **refined the framework and logical structure of the paper** and further **enhanced the detailed descriptions of the figures and tables**.

We respectfully request that the reviewers reconsider the enhancements made in light of their constructive feedback. We believe these revisions not only strengthen the paper but also provide substantial insights to the community. We are hopeful that these changes will merit an increase in the evaluations of our submission. Thank you for your thoughtful consideration.

---

### Decision · Action_Editor_bSYm · 2024-09-22

**Recommendation:** Accept with minor revision

**Comment:**

While reviewers certainly do not fully agree with all of the opinions, predictions, and positions presented in the paper, putting them all together they are leaning more towards recommending acceptance (potential for disagreement in and of itself is fine for this type of position paper).

I suspect that the changes I requested above (in my comments on Claims and Evidence) may be relatively minor in terms of the amount of text that requires changing, but I view them as crucial for the paper to be considered acceptable. Based on this, I recommend **Accept with minor revision**, with the acceptance being conditional on the changes as requested above.

Considering the extensive literature review that the paper draws its insights from, and the recommendations from the reviewers, I propose that a Survey Certification may be appropriate.

**Audience:**

I see no reason to doubt that this paper could be of interest to at least a part of the TMLR audience.

**Claims And Evidence:**

While the paper does not bring new evidence in the form of, e.g., experiments or theoretical proofs, it does draw insights from a very extensive literature review, and builds upon these insights and references to motivate their predictions and proposed research agenda. I view this as an adequate form of evidence for most parts of the paper.

However, there are two parts of the paper where the evidence (or discussion surrounding it) is lacking, and it is crucial for these concerns to be addressed before I see the paper being fit for acceptance:

1. Figure 1 is one of the only places in the paper where claims are made involving rather specific numbers: percentages on the y-axis indicating the "proportion of human activities surpassed by AI". It is unclear what evidence these numbers are based on. The caption mentions *"The statistics presented here are calculated based on our heuristic estimations, which, despite being empirical, can serve as a rudimentary guidance for understanding the speed at which AI develops."*
    - "based on our heuristic estimations" is not a sufficiently clear description. If you present precise numbers, it should be possible to precisely describe the equation/procedure/algorithm that produced them. There should be no ambiguity. Any reader should be able to repeat the same process and arrive at exactly the same numbers.
    - While I am not sure, I suspect that these numbers are very much your personal ballpark estimates. I find it difficult to imagine how anyone could get anything more precise than that. But, if this is the case, it should be clearly presented as such. It would probably be better even to have no numbers at all on the $y$-axis, and only clearly indicate that your personal estimation is that this would probably be a log scale.
    - The caption says that the trend of AI popularity and generality is increasing in an exponential rate, and implies that we may expect similar growth continuing forwards. Even if we take the numbers for granted (ignoring my concerns described above), the current discussion fails to acknowledge the alternative explanation that we might be on a sigmoid curve rather than an exponential curve. If you zoom in on only the initial part of a sigmoid, it will very much look like an exponential, but if you zoom out to include the future (for which we do not yet have data), it will plateau rather than continuing to accelerate. It is important that the paper covers all plausible viewpoints, not just one.

2. Close to the end of the paper, Jensen Huang, Sam Altman, and Elon Musk are cited as "prominent figures in the AI community" who have expressed their confidence in the eventual realization of AGI.
    - While I think it can be okay to cite them, it is important to consider that all three of these people are people who have substantial financial interests (given their positions in various companies) in hyping up the field. They have significant conflicts of interest when making statements about exactly this topic. It is important that the paper acknowledges this context.
    - The choice of just these three people as examples of prominent figures in the AI community is questionable. For the topic of this paper, Gary Marcus is an example of a prominent figure who immediately jumps to mind, who in some ways has very different opinions related to AGI (still being confident in an eventual realization of AGI, albeit on a very different time scale). I am sure there are plenty of other prominent figures who have expressed their opinions too. It won't be possible to cite every single person, but it is important to give a balanced overview of the different perspectives that exist in the community.

---

> ### Author Response · Authors · 2024-10-28
> **Revision Notes for the Camera-Ready Version and Acknowledgment of the Editors' and Reviewers' Hard Work**
>
> We sincerely appreciate the hard work of all of the editors and reviewers. Your valuable suggestions have greatly improved our paper. To illustrate the changes made in our camera-ready version, we have summarized them as follows:
>
> **Regarding the controversial issue surrounding Figure 1**, we carefully considered the suggestions from the editor and reviewers and decided to present them in Section 6.5, "Alternative Perspectives on the AGI Roadmap," on page 62. We framed the illustrations of Figure 1 as the authors' estimation of one possible perspective on AGI development, aiming to inspire further reflection from readers. Additionally, we provided a brief overview of the estimation process in the figure’s caption.
>
> **Concerning the citation and discussion of the views expressed by Jensen Huang, Sam Altman, and Elon Musk towards the end of the paper**, we decided to remove this. Following the editor’s advice, we introduced Section 6.4, "'How Far Are We from AGI' Workshop Discussions," to offer a balanced overview of the various perspectives present within the community.
>
> **We also revised the title based on the reviewers’ suggestions** to better reflect the relationship between our paper and LLMs. Specifically, we emphasized that, in addition to LLMs, we discussed other potential pathways to AGI. Correspondingly, we added clarifications at the beginning of both the abstract and introduction.
>
> Once again, we sincerely thank all the editors and reviewers for their dedicated efforts and valuable feedback, which have significantly enhanced the quality of our paper.

---

> > ### Comment · Action_Editor_bSYm · 2024-10-29
> >
> > Thanks for the revision.
> >
> > I continue to have reservations about Figure 10 (which used to be Figure 1 before), and this still requires fixing as per my original decision and post above. The $y$-axis of the figure still contains specific numbers, with the axis label still simply describing these as a "proportion of human activities". There is no suggestion of these numbers being essentially made up there. The caption also still describes these numbers as "statistics" (a word that implies they come from actual data), being based on "heuristic estimations", and being "empirical".
> >
> > - If the numbers are actual statistics supported by data, we need to know what that data is and how the statistics were derived from the data.
> > - If a heuristic was used, we need to know exactly what the heuristic was (e.g., some sort of equation or algorithm), such that we could compute the same numbers accordingly.
> > - The word "empirical" again suggests somehow running experiments/having data. If any experiments/data were involved in producing these numbers, they need to be described.
> > - If the numbers were produced by any kind of other procedure, this should be described clearly and unambiguously, and some of the other words I highlighted above may not be appropriate and should be changed.
> > - If the numbers were really just completely made up, it might be better not to have numbers at all.
> >
> > I am happy with the other changes.

---

> > > ### Author Response · Authors · 2024-10-30
> > > **Further Revision Notes for Figure 10**
> > >
> > > Thank you for your valuable suggestions. We fully recognize that obtaining precise numerical estimates through experiments or computational calculations is very challenging or currently unlikely for Figure 10. We have taken note of this and followed your advice by removing the numerical labels from the y-axis of Figure 10 and have explained this in the caption. We hope that Figure 10 and the corresponding analysis will inspire readers to think about the progress of AI development from the perspective of the proportion of human activities surpassed by AI, and how far we are from achieving AGI. Once again, thank you for your hard work and patient responses, all of which have greatly enhanced our paper.